# Genome-resolved metagenomics reveals site-specific diversity of episymbiotic CPR bacteria and DPANN archaea in groundwater ecosystems

Christine He [1], Ray Keren [2], Michael L. Whittaker[3,4], Ibrahim F. Farag [1], Jennifer A. Doudna [1,5,6,7,8], Jamie H. D. Cate [1,5,6,7] and Jillian F. Banfield [1,4,9] ✉

**Candidate phyla radiation (CPR) bacteria and DPANN archaea are unisolated, small-celled symbionts that are often detected in groundwater. The effects of groundwater geochemistry on the abundance, distribution, taxonomic diversity and host association of CPR bacteria and DPANN archaea has not been studied. Here, we performed genome-resolved metagenomic analysis of one agricultural and seven pristine groundwater microbial communities and recovered 746 CPR and DPANN genomes in total. The pristine sites, which serve as local sources of drinking water, contained up to 31% CPR bacteria and 4% DPANN archaea. We observed little species-level overlap of metagenome-assembled genomes (MAGs) across the groundwater sites, indicating that CPR and DPANN communities may be differentiated according to physicochemical conditions and host populations. Cryogenic transmission electron microscopy imaging and genomic analyses enabled us to identify CPR and DPANN lineages that reproducibly attach to host cells and showed that the growth of CPR bacteria seems to be stimulated by attachment to host-cell surfaces. Our analysis reveals site-specific diversity of CPR bacteria and DPANN archaea that coexist with diverse hosts in groundwater aquifers. Given that CPR and DPANN organisms have been identified in human microbiomes and their presence is correlated with diseases such as periodontitis, our findings are relevant to considerations of drinking water quality and human health.**

Metagenome-enabled phylogenomic analyses have led to the classification of two groups of organisms that lack pure culture representatives—the CPR bacteria and DPANN archaea[1–4]. Although diverse, CPR and DPANN organisms share conserved traits that are indicative of a symbiotic lifestyle, being ultrasmall in size with small genomes and minimal biosynthetic capabilities[5–9]. Episymbiosis (surface attachment) with bacterial or archaeal hosts has been observed in co-cultures of Saccharibacteria with Actinobacteria[10,11], Nanoarchaeota with Crenarchaeota[12–14], and Nanohaloarchaeota and archaeal Richmond Mine acidophilic nanoorganisms (ARMAN) with Euryarchaeota[15–17], and one case of endosymbiosis has been reported in which a member of the CPR superphylum Parcubacteria lives inside a protist[18]. CPR and DPANN organisms are ubiquitous and can be abundant in groundwater, in which they are predicted to contribute to biogeochemical cycling[2,4,8,9,19–24]. CPR bacteria can persist in drinking water through multiple treatment methods[25–27], posing the question of whether groundwater is a source of CPR[10,11,28–30] and DPANN[31] organisms detected in human microbiomes.

The variation in the abundance and distribution of CPR and DPANN organisms in groundwater environments, their roles and their relationships with host organisms are not well characterized. Subsurface environments such as groundwater are difficult to sample and are poorly characterized compared with surface environments, despite harbouring an estimated 90% of all bacterial biomass[32]. CPR/DPANN organism abundance is likely to have been underestimated in genomic surveys because they are small enough to pass through 0.2 μm filters, which are widely used to collect cells. Furthermore, 'universal' primers to divergent or intron-containing 16S rRNA genes[2,12,15] are unlikely to detect many members of both groups. Most of the available near-complete CPR and DPANN genomes are from just two aquifers[2,19,22]. In this Article, to investigate the roles that CPR and DPANN organisms may have in groundwater ecosystems, we applied genome-resolved metagenomics to analyse eight groundwater communities in Northern California, and cryogenic transmission electron microscopy (cryo-TEM) to image the community with the highest abundance of CPR/DPANN organisms.

## Results

**Metagenome sampling and MAG assembly.** The planktonic fractions of eight groundwater communities in Northern California were sampled during 2017–2019 (Fig. 1a) using bulk filtration (0.1 μm filter). Some sites were also sampled using serial size filtration (2.5 μm, 0.65 μm, 0.2 μm and 0.1 μm filters) in parallel. This enterprise required pumping 400–1,200 l of groundwater from each site through a purpose-built sequential filtration apparatus to recover sufficient biomass for deep sequencing of each size fraction (Methods). One site (Ag) is an agriculturally impacted,

[1]Innovative Genomics Institute, University of California, Berkeley, CA, USA. [2]Department of Civil and Environmental Engineering, University of California, Berkeley, CA, USA. [3]Energy Geoscience Division, Lawrence Berkeley National Laboratory, Berkeley, CA, USA. [4]Department of Earth and Planetary Sciences, University of California, Berkeley, CA, USA. [5]Department of Molecular and Cell Biology, University of California, Berkeley, CA, USA. [6]Department of Chemistry, University of California, Berkeley, CA, USA. [7]Molecular Biophysics and Integrated Bioimaging Division, Lawrence Berkeley National Laboratory, Berkeley, CA, USA. [8]Howard Hughes Medical Institute, University of California Berkeley, Berkeley, CA, USA. [9]Department of Plant and Microbial Biology, University of California, Berkeley, CA, USA. ✉e-mail: jbanfield@berkeley.edu

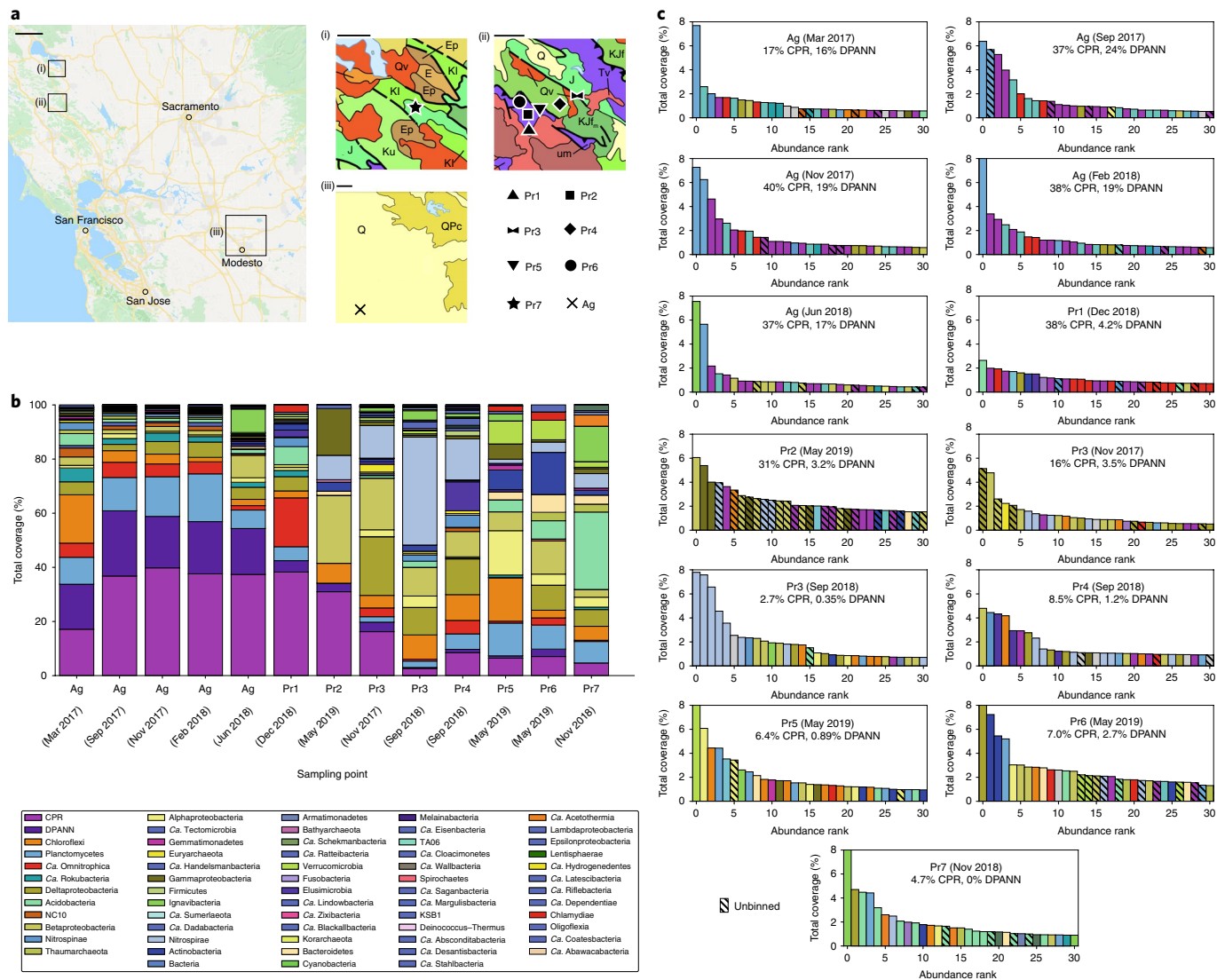

**Fig. 1 | Sampling and overview of groundwater communities. a**, Map of eight Northern California groundwater sites sampled in this study (image from Google Maps). Insets: geological maps (i)–(iii) show the sampled areas (black boxes). J, marine sedimentary and metasedimentary rocks (Jurassic); KJf$_m$, marine sedimentary and metasedimentary rocks (Cretaceous–Jurassic); Q, marine and non-marine (continental) sedimentary rocks (Pleistocene–Holocene); Qv, marine sedimentary and metasedimentary rocks (Cretaceous–Jurassic); um, plutonic (Mesozoic); K, marine sedimentary rocks (Pliocene); Kl, marine sedimentary and metasedimentary rocks (Lower Cretaceous); Ep, marine sedimentary rocks (Paleocene); E, marine sedimentary rocks (Eocene); QPc, non-marine (continental) sedimentary rocks (Pliocene–Pleistocene); Tv, volcanic rocks (Tertiary). Scale bars, 23.3 km (large map), 3.2 km (i and ii) and 9.7 km (iii). **b**, Phylum-level breakdown (with the exception of CPR and DPANN superphyla) of *rpS3* genes detected in each site. The sampling dates for each site are indicated. **c**, Rank abundance curves showing the 30 *rpS3* genes with highest relative coverage identified for each site. The hatched bars indicate an unbinned *rpS3* gene.

river sediment-hosted aquifer and the remaining sites are pristine groundwater aquifers hosted in a mixture of sedimentary and volcanic rocks (Pr1–Pr7, numbered in decreasing order of total CPR and DPANN organism abundance). On the basis of the high abundance and diversity of CPR/DPANN organisms found at the Ag site in a previous metagenomics study[33], we sampled five time points over 15 months.

Binning of bacterial and archaeal genomes from metagenomic data was performed using four different binning algorithms/techniques on the basis of GC content, coverage, the presence/copies of ribosomal proteins and single-copy genes, tetranucleotide frequencies and patterns of coverage across samples (Methods). The highest quality bins were chosen using DASTool[34] and then manually curated. All bins for a given site were dereplicated at 99% average nucleotide identity (ANI)[35], resulting in a dereplicated set of 2,007

genomes across all sites (≥70% completeness and ≤10% contamination). The median genome completeness was >90% and up to 58% of metagenomic reads mapped to each site's dereplicated genome set (assembly and binning statistics are provided in Supplementary Tables 1 and 2). Of this dereplicated set, 540 and 206 genomes were classified as CPR bacteria and DPANN archaea, respectively.

**Abundance and diversity of CPR/DPANN organisms.** We first sought to characterize and compare compositions of the eight groundwater communities, with a particular focus on CPR and DPANN organisms. To broadly survey microbial community composition, we used the ribosomal protein uS3 (encoded by *rpS3*) as a single-copy marker gene due to its strong phylogenetic signal[36]. A comparison of *rpS3* genes against recovered genomes indicated that, with the exception of the Pr2 site, the majority of the most abundant

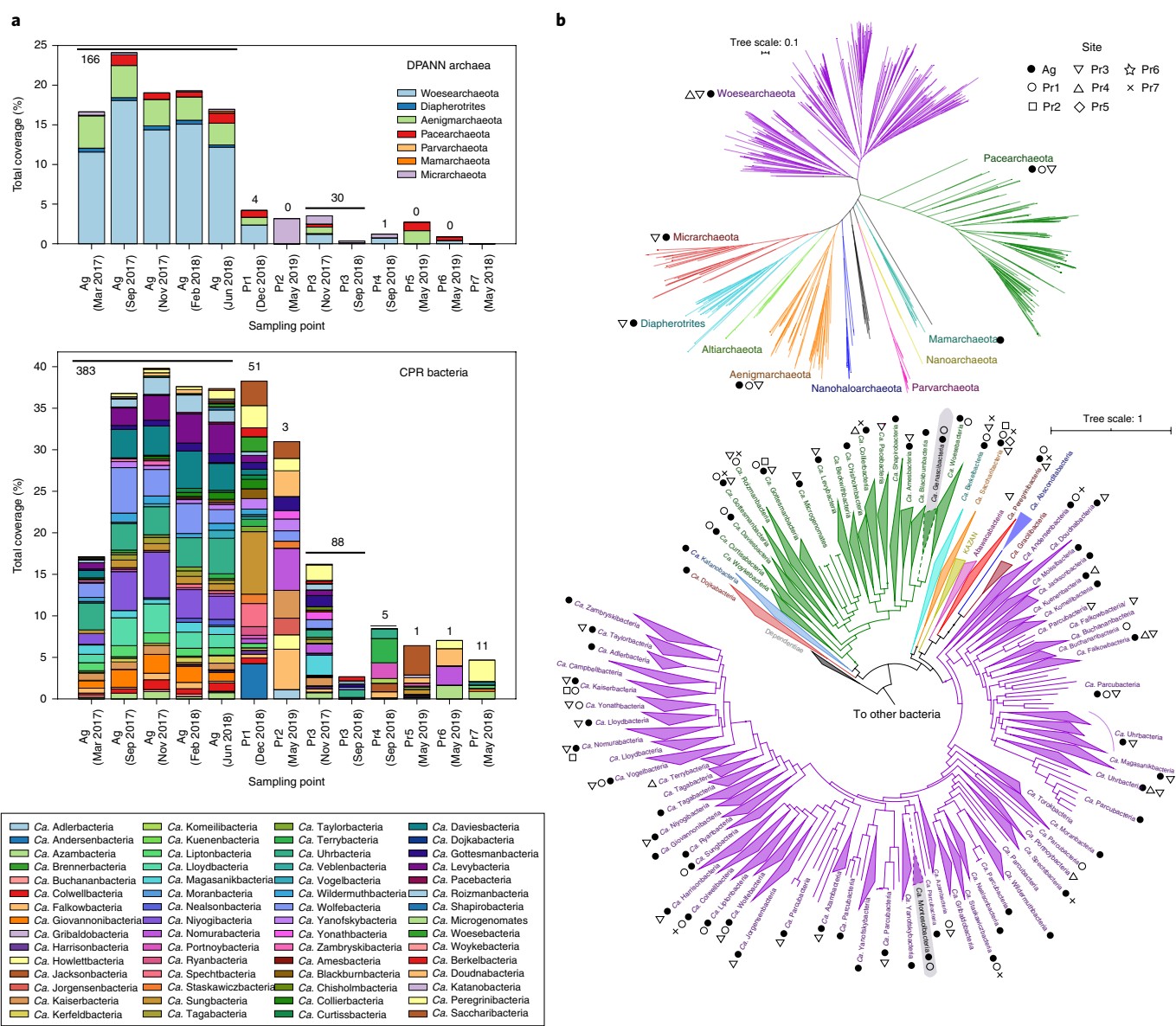

**Fig. 2 | Distribution of CPR and DPANN organisms across groundwater sites. a**, Abundances (relative coverage of scaffolds containing *rpS3* marker genes) of phylum-level lineages within DPANN (above) and CPR (below) in all sites. The numbers above the bars indicate the number of draft-quality (>70% complete, <10% contamination) dereplicated DPANN or CPR genomes that were recovered from metagenomic reads. **b**, Maximum likelihood phylogenetic tree of the DPANN radiation (top), based on 14 concatenated ribosomal proteins, and of the CPR (bottom), based on 15 concatenated ribosome proteins. Phylum-level lineages within the CPR (as previously defined[2]) are collapsed. Markers next to each lineage indicate the groundwater sites where at least one representative genome from that lineage was recovered. New CPR lineages '*Candidatus* Genascibacteria' (within the Microgenomates superphylum in green) and '*Candidatus* Montesolbacteria' (within the Parcubacteria superphylum in purple) are highlighted in grey.

organisms at each site are represented by genome bins (Fig. 1c, the hatched bars indicate unbinned *rpS3* genes). We found that all of the groundwater communities are distinct in phylum-level composition (Fig. 1b,c), with a strong divide between the Ag site and the pristine sites on the basis of principal component analysis (Extended Data Fig. 1). Change over time of the Ag groundwater community is examined in further detail in the 'An agriculturally impacted groundwater site rich in CPR/DPANN' section.

Specifically, the populations of CPR and DPANN organisms are quite distinct between sites (Fig. 2a), although a few CPR and DPANN lineages are fairly ubiquitous across sites (Extended Data Fig. 2). Across all of the sites, CPR and DPANN organisms represent 3–40% and 0–24% of the communities (measured by bulk filtration onto a 0.1 μm filter), respectively. The abundance of DPANN

archaea in Ag groundwater (10–24%) is much higher compared to the pristine sites, where DPANN organisms comprise <5% of the community. Across all of the sites, genomes were recovered from 58 out of 73 currently identified phylum-level lineages within the CPR[4] and from 6 out of 10 currently identified phylum-level lineages within the DPANN radiation (Fig. 2b). In particular, recovered CPR genomes from Ag groundwater span most of the diversity within the CPR (Fig. 2b, filled black circles). On the basis of the criteria for 16S rRNA gene sequence identity (<76% for phylum-level[21,37]) and concatenated ribosomal protein phylogenetic placement[2], we defined two new phylum-level lineages within the CPR, each consisting of sequences from Ag and Pr1 groundwater (Supplementary Table 3). We propose the names '*Candidatus* Genascibacteria' and '*Candidatus* Montesolbacteria' for these new phylum-level lineages

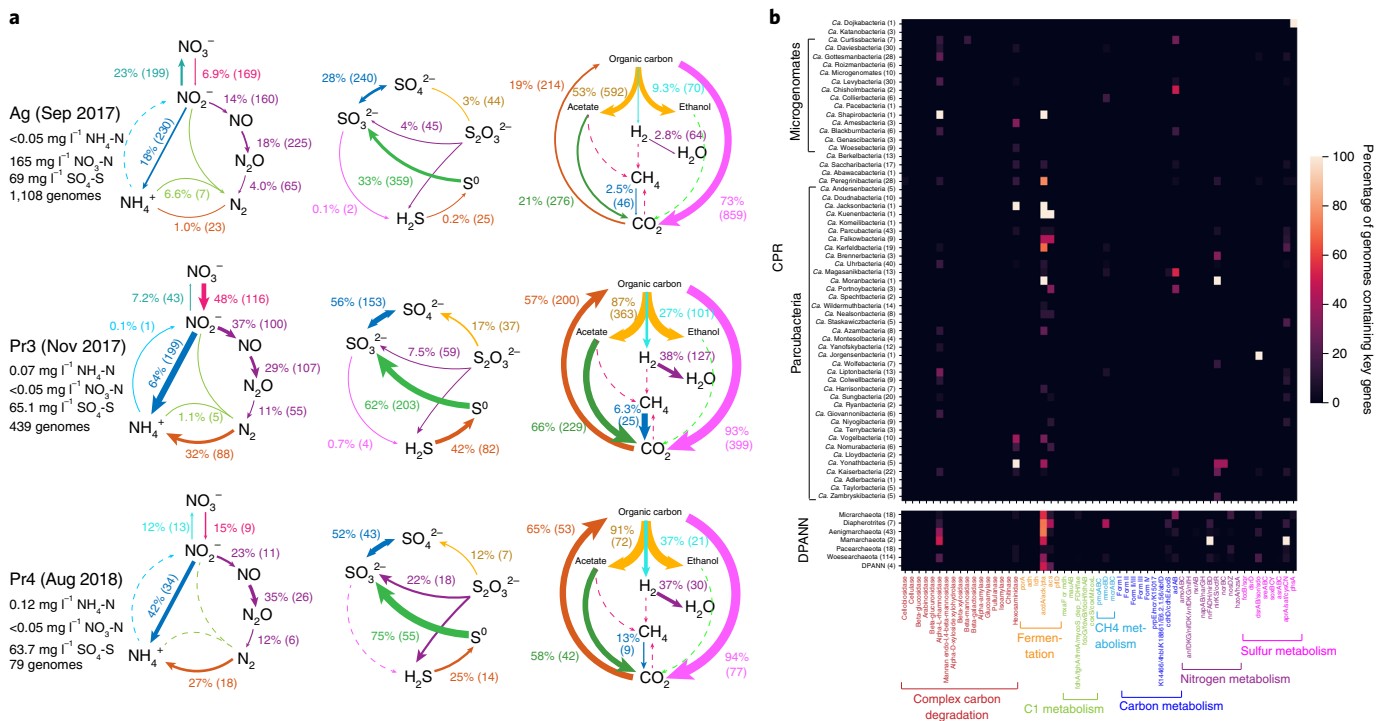

**Fig. 3 | Metabolic profile of groundwater communities. a**, Biogeochemical cycling diagrams profiling the community-level metabolic potential of three groundwater sites sampled in this study (all of the sites are shown in Extended Data Fig. 4). The total relative abundance of all genomes capable of carrying out the step, as well as the number of genomes containing the capacity for that step, are listed next to each metabolic step. Arrow sizes are drawn proportional to the total relative abundance of genomes capable of carrying out the metabolic step. **b**, Heat map of 746 CPR and DPANN genomes from this study (rows are phylum-level lineages, and the numbers in parentheses are the number of genomes recovered), showing the percentage of genomes containing key genes required for various metabolic and biosynthetic functions (columns).

(Fig. 2b, highlighted in grey) on the basis of the two sites at which the representative sequences were found.

To assess groundwater community similarity at the genome level, we used ANI to cluster[35] the 2,007 genomes from this study with 3,044 genomes from previous studies of two groundwater sites rich in CPR/DPANN organisms: Crystal Geyser in Utah[22,38,39] and an aquifer adjacent to the Colorado River in Rifle, Colorado[2,19]. At the strain level (>99% ANI), there is very little similarity between genomes of the analysed sites; most pairs of sites share one or no strains despite the fact that sites Pr1 to Pr6 are located in close proximity (~1 km between neighbouring sites) and multiple sites are hosted in plutonic rock. The sole pair of sites that share more than a few strains (>99% ANI) is Pr1 and Pr7, which share 44 strains, including 7 CPR bacterial strains (Extended Data Fig. 3). It is unlikely that the aquifers of these two sites are connected as they lie on separate sides of Putah Creek, a major hydrological feature. Furthermore, we do not attribute this observed genome similarity to index hopping during sequencing (Methods). Even at the species level (>95% ANI[40]), most pairs of analysed sites share no more than one species in common (Extended Data Fig. 3). The overall lack of genomic similarity between these ten groundwater communities—at the phylum, species and strain levels—indicates that there is a high degree of specialization based on local hydrogeochemical conditions.

**Roles of CPR/DPANN organisms in biogeochemical cycling.** Next, given the abundance of CPR and DPANN organisms, we sought to investigate the potential metabolic roles these organisms have in these eight groundwater communities. As most CPR/DPANN organisms are predicted to be symbionts, it is probable that their metabolic roles within a community vary with the metabolic capacities of their host organisms. To investigate this relationship, we profiled all recovered

genomes against a curated set of protein hidden Markov models (HMMs)[41] (Methods) and utilized genome relative coverage values to compare metabolic profiles of whole communities (Fig. 3a and Extended Data Fig. 4). The metabolic profile of Ag groundwater is clearly differentiated from that of the pristine sites[33] (Extended Data Fig. 5). In Ag groundwater, which receives heavy nitrogen input from neighbouring agricultural activity, ammonia is oxidized by seven Planctomycetes that are capable of anammox (comprising 8% of the community), resulting in low levels of ammonia in Ag groundwater (Fig. 3a). The Ag community encodes greater capacity for nitrite oxidation than nitrate reduction, consistent measurements of high nitrate (165 mg l$^{-1}$ NO$_3$-N) and low nitrite (<0.05 mg l$^{-1}$ NO$_2$-N) levels (Fig. 3a). Most of the groundwater communities sampled have an incomplete capacity for denitrification (Extended Data Fig. 4), with far fewer genomes encoding the required genes for the final step of nitrous oxide reduction compared with the previous steps. Pr3 and Pr4 are two sites with a greater capacity for nitrous oxide reduction compared with the other groundwater communities, in addition to nitrogen fixation, thiosulfate disproportionation, sulfide oxidation and carbon fixation (Fig. 3a and Extended Data Fig. 4). Although Pr3 and Pr4 have little species-level overlap, their similarity in community-level metabolic capacities may reflect their proximity (<1 km) and similar groundwater chemistry (levels of NH$_4$-N, NO$_3$-N, NO$_2$-N and SO$_4$-S).

We specifically examined key metabolic marker genes in CPR and DPANN genomes to assess what metabolic roles that they may have (Fig. 3b). The presence of the nitrite reductase *nirK* in 19 CPR and 4 DPANN genomes as well as the presence of *nosD* in 11 DPANN genomes across sites suggest a complementary or accessory role of many CPR/DPANN lineages in denitrification (consistent with previous identification of *nirK* genes in Parcubacteria[21,42]). Furthermore, 13 DPANN genomes in Ag groundwater encode the small subunit

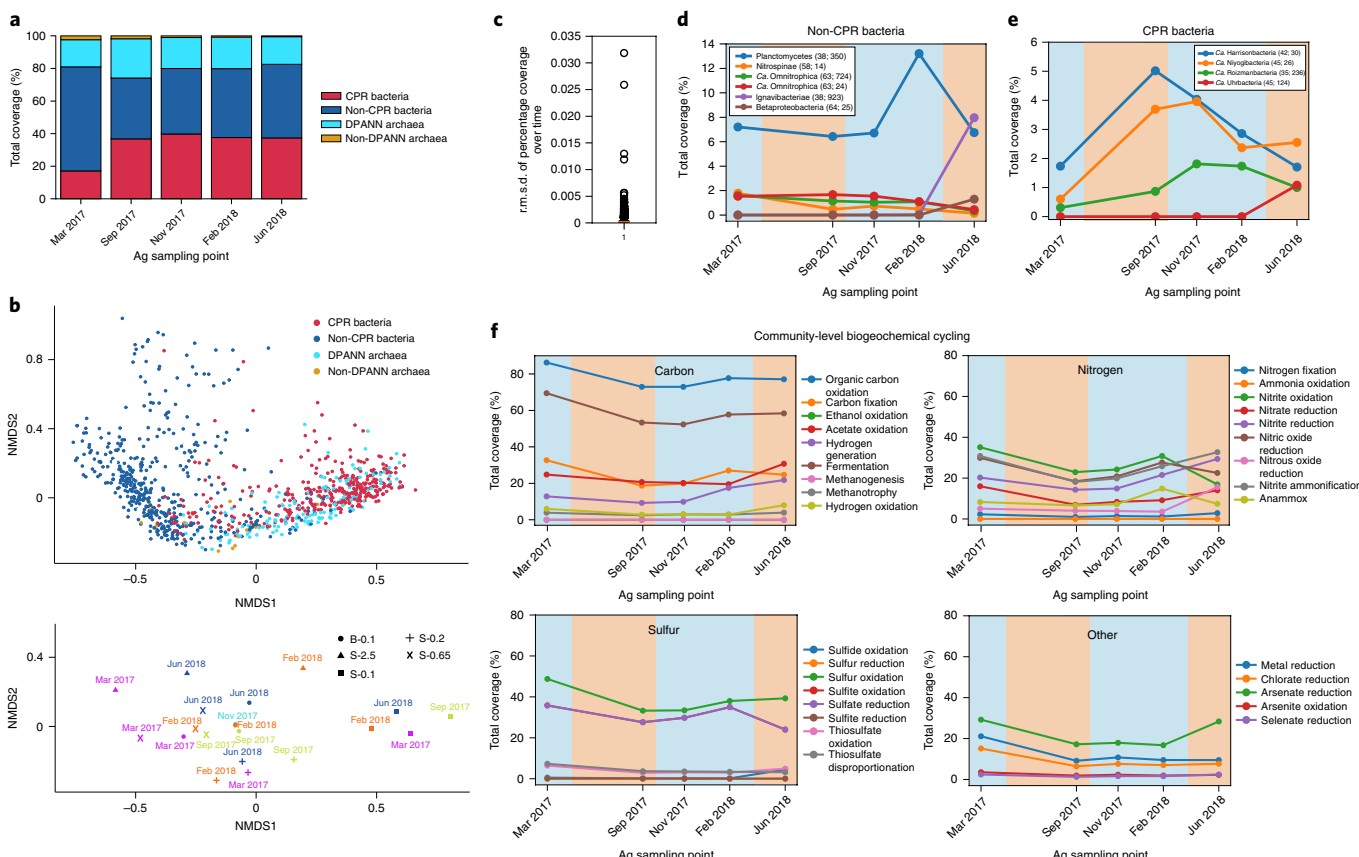

**Fig. 4 | Ag groundwater microbial community over time. a**, Relative abundances of non-CPR bacteria, CPR bacteria, DPANN archaea and non-DPANN archaea genomes (1,103 in total) in Ag over time. **b**, NMDS analysis of 1,103 Ag genome relative abundances in all size fractions over all time points. The positions of genomes in ordination space are shown in the top graph, while the positions of the samples in ordination space are shown in the bottom graph. In the bottom graph, B-0.1 refers to bulk filtration on a 0.1 µm filter (circles), S-2.5 refers to 2.5+ µm size fractions (triangles), S-0.65 refers to 0.65–2.5 µm size fractions (crosses), S-0.2 refers to 0.2–0.65 µm size fractions (plus signs), and S-0.1 refers to 0.1–0.2 µm size fractions (closed squares). **c**, Box plot showing the r.m.s.d. of the relative abundance of all of the genomes in the bulk filter (whole community on a 0.1 µm filter) over time. The median r.m.s.d. (orange line) is <0.001, indicating that there is little variation in relative abundance over time for individual genomes in Ag. **d,e**, The relative abundance over time for non-CPR bacteria (**d**) and CPR bacteria (**e**) that have an r.m.s.d. > 0.004. Genomes are identified in the legend by phylum, percentage GC and coverage in the original time point that the representative genome was derived from (the latter two in parentheses). **f**, The variation in Ag community-level capacity (total relative abundance of all genomes capable of a broad metabolic function) for carbon, nitrogen, sulfur and miscellaneous element cycling over time. For **d–f**, the blue and orange backgrounds indicate the rainy and dry seasons in Northern California, respectively.

of nitrite reductase (*nirD*) but lack the catalytic large subunit *nirB*, suggesting that DPANN organisms have an accessory role in nitrite reduction to ammonia. At Ag, Pr1 and Pr3, we found that 30 DPANN genomes and 3 CPR genomes encode sulfur dioxygenase *sdo*, while dozens of diverse CPR and DPANN genomes encode *sat, cysC* and *cysN*, which are involved in sulfate reduction, suggesting a potential role of CPR and DPANN organisms in transformations to sulfite.

**An agriculturally impacted groundwater site rich in CPR/DPANN organisms.** After establishing the prevalence and metabolic roles of CPR and DPANN organisms in groundwater communities, we performed temporal and size filtration sampling of Ag groundwater (Fig. 4a) to investigate how these characteristics change with time and environmental factors. Non-metric multidimensional scaling (NMDS) ordination (Methods) shows that, as expected, most CPR and DPANN genomes cluster together and away from other bacteria and archaea, distinguished by prevalence in the 0.1–0.2 µm fraction (Fig. 4b). There is no observable clustering of genomes by sampling time in ordination space and the median root mean square deviation (r.m.s.d.) of genome relative abundances over time is ~0.002 (Fig. 4c), indicating a very stable community at the strain level (genomes dereplicated

at 99% ANI). Inspection of abundance patterns in individual genomes with an r.m.s.d. > 0.004 (Fig. 4d,e) show a coabundance pattern between a Planctomycetes organism and several CPR bacteria that, although certainly not conclusive, may result from a parasitic CPR–host relationship. The co-occurrence of two Ignavibacteria and Betaproteobacteria organisms with an Uhrbacteria organism (Fig. 4d) may be an indication of a commensal or mutualistic CPR–host relationship. These observed temporal trends merit further investigation to determine whether they reflect symbiotic relationships.

Examination of the changes in metabolic cycling capacities in Ag groundwater over time indicate that there is a higher community capacity (5–10% relative abundance) for organic carbon oxidation, carbon fixation, fermentation, nitrite oxidation, nitric oxide reduction and sulfate reduction during the rainy season (Fig. 4f, blue background) compared with the dry season (Fig. 4f, orange background). Furthermore, we see a greater increase in these metabolic capacities during the 2016–2017 rainy season compared with the 2017–2018 rainy season (Fig. 4f), which may reflect a major difference in rainfall (more than 25 cm more in 2016–2017 versus 2017–2018)[43]. Overall, we found that Ag groundwater is an extremely stable incubator for high abundance and diversity of both CPR and

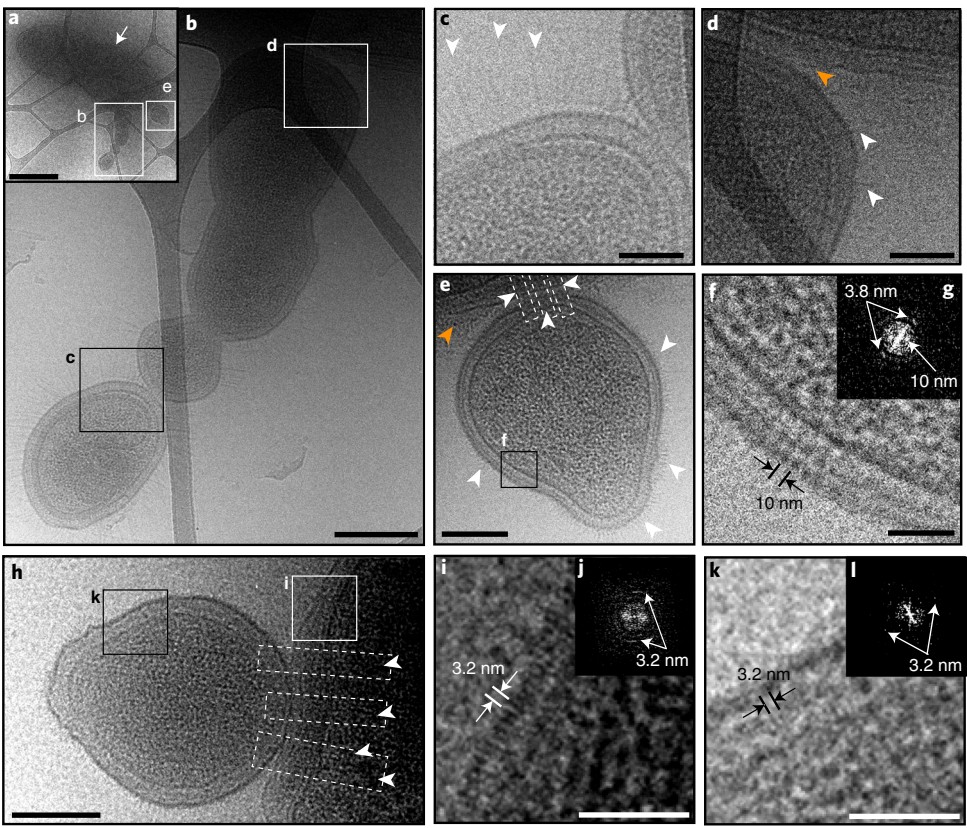

**Fig. 5 | Imaging of Ag groundwater cells concentrated with tangential flow filtration. a**, Image of larger host cell (white arrow) with multiple ultrasmall cells (magnified in **b** and **e**) attached. **b**, Magnification of the indicated area in **a** (white box) showing a chain of ultrasmall cells attached to the surface of the larger host cell. **c**, Magnification of the indicated area in **b** (black box) showing the contact region between two ultrasmall cells and the pili-like appendages decorating their surfaces (white arrowheads). **d**, Magnification of the indicated area in **b** (white box) showing the contact region between an ultrasmall cell and host cell. Pili-like appendages are indicated by white arrowheads. **e**, Magnification of the indicated area in **a** (white box) showing a single ultrasmall cell decorated by pili-like appendages (white arrowheads) attached to a host cell. Attachment may be mediated by pili-like appendages that extend from the ultrasmall cell into the host cell (dashed white boxes). **f,g**, The membrane from the ultrasmall cell in **e** (black box), with the membrane structure showing a clear periodicity measured to be 10 nm (**f**) as well as a periodicity of 3.8 nm that is evident in Fourier space (**g**; white arrows indicate repeating structure spacings). **h**, Image of a host cell with a single ultrasmall cell attached. Several pili-like appendages extend from the ultrasmall cell into the host cell (dashed white boxes and arrowheads). **i,j**, Magnification of the indicated area in **h** (white box) showing the host cell envelope, the outer layer of which exhibits a periodicity of 3.2 nm; a Fourier-transformed image is shown (**j**; white arrows indicate repeating structure spacings). **k,l**, Magnification of the indicated area in **h** (black box) showing the ultrasmall cell envelope, which also exhibits a periodicity of 3.2 nm in the outer layer; a Fourier-transformed image is shown (**l**; white arrows indicate repeating structure spacings). For **d** and **e**, the orange arrowheads indicate lines of high density observed at the contact interface. Scale bars, 1 μm (**a**), 200 nm (**b**, **e** and **h**) and 50 nm (**c**, **d**, **f**, **i** and **k**).

DPANN organisms, but the microbial community is not stagnant in its metabolic capacities, which vary between rainy and dry seasons.

**Pili-mediated episymbiotic interactions between ultrasmall cells and hosts.** Fundamental to understanding the wider role of CPR and DPANN organisms in groundwater communities is characterizing their relationships with hosts. With few exceptions, host organisms have not been conclusively identified and only a handful of studies have performed high-resolution microscopy to directly image the physical associations between CPR/DPANN organisms and hosts in natural environments[15,44,45]. To observe CPR/DPANN–host interactions in the Ag groundwater community in a near-native state, we used tangential flow filtration (TFF) to gently concentrate cells from groundwater and preserved them by cryo-plunging them in liquid ethane on-site for later characterization by cryo-TEM (Methods).

Many ultrasmall cells (longest dimension <500 nm) were observed attached to the surface of larger host cells (Fig. 5a). The ultrasmall cells have cell envelopes that are decorated by pili (Fig. 5, white arrows; a magnified view is shown in Extended Data

Fig. 6), some of which extend into the corresponding host cell, potentially mediating episymbiont–host interaction (Fig. 5, white dashed boxes). At the ultrasmall cell–host contact region shown in Fig. 5e,h, the host cell envelope appears to be thickened, whereas the episymbiont cell envelope is thinned. For multiple pairs of ultrasmall cells and hosts, a line of higher density is observed at the cell interface (Fig. 5d,e,i, orange arrows), similar to what has previously been observed at tight interfaces between archaeal ARMAN (DPANN) cells and their Thermoplasmatales hosts[44]. The host in Fig. 5a has multiple ultrasmall cells directly attached to its cell envelope that appear to be in the process of dividing (Fig. 5b,e,f), raising the possibility that CPR/DPANN replication is correlated with host attachment (discussed in the next section). Overall, cryo-TEM imaging of TFF-concentrated groundwater shows that some ultrasmall cells in Ag groundwater—which are likely to be CPR or DPANN organisms on the basis of size—are episymbionts of prokaryotic hosts, attaching through pili-like structures.

An important question regarding the biology of CPR bacteria relates to the nature of their cell envelope and the degree to which

it resembles that of their host cells. Genomic analysis indicates that CPR bacteria cannot de novo synthesize fatty acids[4] but do possess fatty-acid-based membrane lipids[46], raising the possibility that CPR bacteria receive lipids or lipid building blocks from host organisms. The ultrasmall cell in Fig. 5e has a surface layer with a periodicity of 3.8 nm and 10 nm, but lacks an outer membrane expected for Gram-negative bacteria (Fig. 5f,g), consistent with previous cryo-TEM images of groundwater CPR bacteria[45]. Meanwhile, the host's cell envelope appears to have two lipid layers, suggesting a Gram-negative structure (Fig. 5d,e). In Fig. 5h, from a different ultrasmall cell–host pair, we also resolve two lipid layers in the host cell envelope and no outer membrane in the ultrasmall cell. Interestingly, in this case, a periodicity of 3.2 nm is detected in the outermost layers of both the host cell (Fig. 5i) and the attached ultrasmall cell (Fig. 5k), but it is unclear whether the outer layers of the two cells have the same structure. On the basis of an apparent lack of an outer membrane, the ultrasmall cells observed in Ag groundwater have cell envelopes that do not resemble those of Gram-negative bacteria, but seemingly can attach to Gram-negative hosts.

**Host attachment and replication of CPR/DPANN cells.** Imaging of likely CPR/DPANN organisms directly attached to host cells led us to investigate how widespread physical attachment is across the diversity of both radiations. We analysed the distribution of CPR/DPANN organisms among size fractions across five sites, which should reflect two factors—cell size and attachment to host cells. Most microorganisms outside the CPR and DPANN groups are present in the 0.65–2.5 μm or 2.5+ μm fractions (Fig. 4b). Owing to their small cell size (average ~0.2 μm diameter[47]), a CPR/DPANN cell present in the 2.5+ μm or 0.65–2.5 μm fraction is probably attached to a larger organism, whereas a CPR/DPANN cell present in the 0.1–0.2 μm fraction is probably unattached. Substantial coverage of CPR/DPANN genomes in the 2.5+ μm and 0.65–2.5 μm fractions indicate that a fair number of CPR/DPANN cells retain host attachment throughout the filtration process, and we consider it probable that pili penetrating from ultrasmall cells into the host (Fig. 5) are strong enough to resist disruption. We therefore consider the distribution of CPR/DPANN organisms among size fractions as indicative of the degree of host attachment.

To assess the distribution of organisms among size fractions, the absolute number of cells represented by each genome was estimated from the genome relative abundance and the mass of DNA extracted (Methods). We observed high cell counts (>$10^{28}$ cells) of CPR and DPANN genomes in 2.5+ μm and 0.65–2.5 μm fractions (Fig. 6a), representing a diverse range of lineages (Supplementary Table 7). In the case of Ag on March 2017 and September 2017, cell counts of diverse CPR and DPANN genomes (Extended Data Fig. 7) were several orders of magnitude higher in the 0.65–2.5 μm fraction than in the 0.1–0.2 μm fractions (Fig. 6a). These cell count distributions suggest that a host-attached lifestyle is common across diverse CPR and DPANN lineages and across groundwater sites.

For most CPR and DPANN lineages, estimated cell counts were significantly higher in the 0.1–0.2 μm fractions compared with the 2.5+ μm fractions, whereas the other bacterial and archaeal lineages exhibited the reverse trend (paired t-test; Fig. 6b). Two notable exceptions are the CPR lineage 'Candidatus Kerfeldbacteria' and the DPANN lineage 'Candidatus Pacearchaeota', which were enriched in the 2.5+ μm fraction relative to the 0.1–0.2 μm fraction (paired t-test, P = 0.027 and 0.021; Fig. 6b), indicating that a high fraction of these populations is host-attached and/or the attachment is more resistant to the disruptive effects of filtration compared with other CPR/DPANN lineages. Ca. Pacearchaeota genomes encode especially minimal metabolic capacities among DPANN lineages (Fig. 3b), suggesting a heavy dependence on host resources[48]. An additional CPR lineage, 'Candidatus Woesebacteria', was found to

have significantly higher cell counts in the 2.5+ μm fraction versus the 0.2–0.65 μm fraction (Extended Data Fig. 8).

Cryo-EM images of dividing, host-attached ultrasmall cells (Fig. 5a,b) suggest that attachment to a host may stimulate CPR/DPANN cell division. To investigate this hypothesis, we calculated instantaneous replication rates (iRep values[49]) for CPR genomes in Ag groundwater (archaeal genomes were excluded as archaeal replication is not generally bidirectional). For reference, iRep = 1.0 indicates that, on average, no cells represented by a genome are actively replicating, whereas iRep = 2.0 indicates that, on average, every cell represented by a genome is creating one copy of its genome. We found that at three Ag sampling points—March 2017, February 2018 and June 2017—CPR organisms exhibit significantly higher replication rates in the 0.65–2.5 μm fraction than the 0.1–0.2 μm fraction (Fig. 6c), suggesting that host-attached CPR bacteria consistently exhibit a higher replication rate than non-host-attached CPR bacteria. We found that CPR bacteria as a whole (measured in the bulk filtered community) exhibited higher replication rates during the height of the 2016–2017 rainy season (March 2017) and the beginning of the next rainy season (September 2018) compared with during the height of the 2018 dry season (June 2018; Fig. 6d). Significant differences in bulk filtration replication rates were not observed between any point and the height of the 2017–18 rainy season (February 2018; Fig. 6d), which may be explained by the >25 cm more rainfall during the 2016–2017 rainy season compared with during the 2017–2018 rainy season[43]. Together, these findings support the deduction that CPR cell replication is stimulated by host attachment and may be more prevalent during the rainy season compared with the dry season.

## Discussion

We sampled one agricultural and seven pristine groundwater sites in Northern California that are situated in a range of rock types and sourced from multiple aquifers. We recovered a total of 746 draft quality CPR and DPANN genomes that derive from most of the major lineages within both radiations and from two apparently new phylum-level lineages within the CPR, hereafter named 'Candidatus Genascibacteria' and 'Candidatus Montesolbacteria'. To our knowledge, only two previous studies have recovered and compared CPR bacterial genomes across multiple groundwater sites[21,50], and neither reported DPANN genomes. Very little species-level overlap (defined as >95% ANI) exists between genomes recovered from this study and previous studies of Crystal Geyser[22,39] and Rifle[2,19] aquifers, a finding that may reflect a combination of species adaptation to different geochemical conditions of the groundwater system[51–53], bottleneck effects and/or founder effects. Our findings suggest that characterization of microbiomes of additional groundwater sites—using 0.1 μm filters rather than 0.22 μm filters and binning of MAGs to capture maximum CPR/DPANN diversity—is likely to reveal further diversity in the CPR and DPANN radiations.

The pristine sites that we sampled serve as sources of local drinking water. Notably, at the time of sampling, the Pr2 site (Rattlesnake Spring), which has been a popular source of public drinking water for over a century, contained more than 30% CPR bacteria and 3% DPANN archaea, raising the possibility that CPR/DPANN organisms in groundwater are the source for human-associated members. CPR bacteria have been detected in multiple human body sites and correlated with inflammatory bowel disease[28], vaginosis[54], periodontitis[55–57] and herpes viral titres[30,58], and DPANN archaea have been detected in lung fluids[31]. However, few genomes of human-associated CPR or DPANN organisms exist, giving limited information about their role in human microbiomes and their relationship with environmental counterparts[31]. One recent study found remarkably low variation and high synteny between human-associated and groundwater Saccharibacteria[59], suggesting the possibility that drinking water is a source of CPR bacteria in

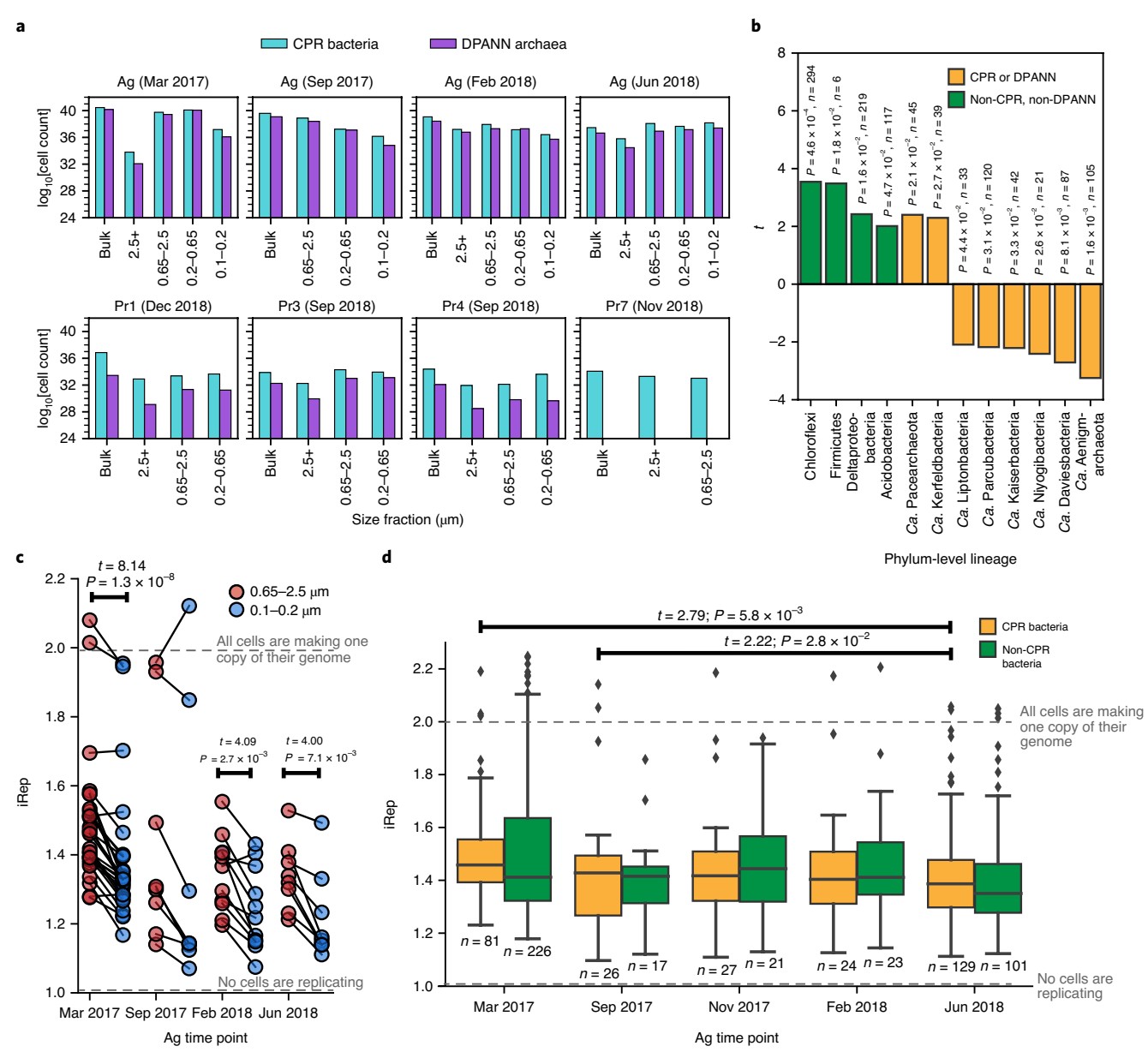

**Fig. 6 | Analysis of host attachment and growth rates of CPR/DPANN organisms. a**, Estimated cell counts (log transformed) for all size fraction data collected in this study. Each size fraction shown corresponds to a single sampling event. It was logistically infeasible to perform size filtration at some sites, and some filters collected did not contain enough biomass for DNA sequencing. **b**, Results from a two-sided paired *t*-test on estimated cell counts of genomes in the largest (2.5+ μm) and smallest (0.1–0.2 μm) size fractions after serial size filtration of Ag groundwater. A positive *t* statistic indicates enrichment of cells in the 2.5+ μm compared with the 0.1–0.2 μm fraction. Values listed above each bar are the calculated *P* value and sample size (*n*, number of genomes tested) for each phylum-level lineage. **c**, Calculated iRep values for CPR bacteria genomes in the 0.65–2.5 μm fraction versus the 0.1–0.2 μm fraction, across all Ag sampling points. *n* = 28 (March 2017), *n* = 8 (September 2017), *n* = 11 (February 2018) and *n* = 8 (June 2018) genomes tested. Note that iRep values represent the average replication state of the cell population represented by a genome. An iRep value of 1.0 indicates that, on average, no cells in the population are actively replicating, whereas an iRep value of 2.0 indicates that, on average, every cell is actively creating one copy of its genome. The statistically significant results (*P* < 0.05) of a two-sided paired *t*-test on iRep values between the two size fractions are shown above the box plots. Note that the November 2017 time point was excluded because only bulk filtration (no size filtration) was performed. **d**, Calculated iRep values for Ag bacteria caught in the bulk 0.1 μm filter (whole-community filtration). The statistically significant results (*P* < 0.05) of independent two-sided *t*-tests on iRep values of CPR bacteria between all possible pairs of sampling points are shown above the box plots. For the box plot, the centre line is the median; the top and bottom lines are the first and third quartiles, respectively; and the whiskers show 1.5× the interquartile range; individual dots are outliers; *n* values (number of genomes tested) are indicated on the plot.

the human oral cavity[25–27]. However, all of these human-associated Saccharibacteria genomes derive from people who use tap water (although CPR bacteria can persist in drinking water after treatment[25–27]) rather than groundwater as a drinking source. To investigate whether groundwater is a source of human-associated CPR

bacteria, it will be necessary to sequence groundwater sites together with the microbiomes of specific humans who use the groundwater versus tap water as their primary source of drinking water.

The Ag groundwater microbial community, which includes organisms from most CPR and DPANN lineages, is extremely stable

at the strain level (>99% ANI). This stability may be due to consistent, heavy input of carbon and nitrogen from agricultural waste (cow manure) collected on-site in lagoons, dried piles and used to fertilize the on-site corn field[33]. Although the Ag community composition is stable, increases in community metabolic capacities and CPR bacterial replication rates occur during rainy seasons. Several factors may contribute to these changes during the rainy season: the onset of more anoxic conditions in the groundwater, greater runoff from agricultural waste piles, increased volume of and changes in microbial composition of the cow manure after calves are born in the spring, and soil changes associated with the adjacent corn field that supplies much of the recharge to the sampled Ag well[33]. Analysis of coabundance patterns over time indicated potential parasitic as well as commensal/mutualistic relationships between several CPR lineages and Planctomycetes, Ignavibacteria and Betaproteobacteria hosts, although more investigation is required to directly connect these observations to symbiotic relationships. These observations provide a starting point for targeted cultivation of CPR and DPANN organisms based on conditions favourable to growth of putative hosts. The recovery of diverse DPANN but few non-DPANN archaeal genomes from Ag groundwater poses the intriguing question of whether bacteria may serve as hosts for DPANN archaea.

An important aspect of our study was the use of cryo-TEM, a technique that has only rarely been applied to study environmental communities in a near-native state[15,44,45], to observe physical attachment between the ultrasmall cells and hosts in Ag groundwater. Combined with genomic analysis of CPR and DPANN cell counts in serial size fractions, our data suggest that physical attachment to host organisms is a common lifestyle in both radiations, with the lineages *Ca.* Kerfeldbacteria in the CPR and *Ca.* Pacearchaeota within DPANN exhibiting particularly strong physical attachment to hosts relative to other CPR and DPANN organisms. On the basis of replication-rate analysis and with cryo-TEM imaging of dividing host-attached ultrasmall cells, higher CPR instantaneous replication rates are associated with physical attachment to hosts, suggesting that the availability of host-supplied resources may stimulate replication of CPR organisms. One recent study instead concluded that there is no widespread attachment of CPR bacteria to hosts on the basis of failure to detect co-occurring CPR and host genomic signatures in SAGs[47]. However, we believe the incompleteness of the reported SAGs and the small absolute number of organisms analysed per site render the results inconclusive. Our study highlights the need for high-quality MAGs and high-resolution microscopy to assess interactions among community members in a robust fashion.

## Methods

**Groundwater sampling, chemistry measurements and surface geology determination.** All groundwater sites were sampled at shallow depths (<100 m below the surface). Groundwater was pumped from each well using a submersible pump (Geotech Environmental Equipment) into a sterile container, and then pumped using a peristaltic pump into an apparatus that was custom built for filtering high volumes of water (Harrington Industrial Plastics) at a rate of 3.8–7.6 l min⁻¹. Before filtration, at least 100 l of water was pumped to purge the well volume and to flush the system. Polyethersulfone membrane filter cartridges designed for high-volume filtration (Graver Technologies) from the ZTEC G series (0.1 μm and 0.2 μm), ZTEC B series (0.65 μm) and PMA series (2.5 μm) were used. When a sufficient volume of water had been filtered (400 l for bulk filtration and an additional 800 l for serial size filtration), filters were removed and stored on dry ice. Filters were stored in a −80 °C freezer until processed. The surface geology of each sampling site was determined from the California Department of Conservation's 2010 geological map of California (https://maps.conservation.ca.gov/cgs/gmc/), rock fragments recovered during drilling (Pr4) and by on-site geological surveys. Pumped groundwater was shipped on dry ice to the UC Davis Analytical Laboratory for water chemistry measurements of electrical conductivity (EC), sodium adsorption ratio (SAR), total organic carbon (TOC), dissolved organic carbon (DOC), NH₄-N, NO₃-N, SO₄-S (soluble S), HCO₃, CO₃, soluble Zn, Cu, Mn, Fe, Cd, Cr, Pb, Ni, K, Ca, Mg, Na, Cl and B. Water chemistry measurements are shown in Supplementary Table 5.

**DNA extraction and sequencing.** The plastic housing was removed from the filter cartridges under sterile conditions and the filters were retained for DNA extraction.

To extract DNA, either a quarter or a half of a filter was placed in PowerBead solution from the Qiagen DNeasy PowerSoil kit (no bead-beating was performed), then vortexed for 10 min with massaging to remove cells from the entire filter surface. After vortexing, the filter was removed, solution C1 (Qiagen DNeasy PowerSoil Kit) was added to the PowerBead solution and the solution was placed in a 65 °C water bath for 30 min. The rest of the DNA extraction procedure was performed according to the Qiagen DNeasy PowerSoil kit manufacturer's instructions, beginning with the addition of solution C2. Ethanol precipitation was performed to concentrate and purify the extracted DNA before sequencing. Genomic DNA was quantified using the Qubit dsDNA High Sensitivity assay and, when quantity permitted, DNA quality was assessed using agarose gel electrophoresis. Library preparation and sequencing were performed at the California Institute for Quantitative Biosciences' (QB3) genomics facility and the Chan Zuckerberg BioHub's sequencing facility. Libraries were prepared with target insert sizes of 400–600 bp. Samples were sequenced using 150 bp paired-end reads on either a HiSeq 4000 platform or a NovaSeq 6000 platform, with a read depth of ~10 Gbp per sample except for Ag March 2017 samples, which were sequenced at 150 Gbp.

**Metagenomic assembly.** BBTools (v.38.78) was used to remove Illumina adapters as well as PhiX and other Illumina trace contaminants[60]. Reads were trimmed using Sickle[61] (v.1.33) using the default quality threshold of 20 (quality type set to sanger, which is CASAVA v.1.8 or higher). Each physical filter was considered to be an independent sample, that is, metagenomic reads from a single filter were assembled together, rather than coassembing total reads from all filters/size fractions. Assembly was performed using MEGAHIT (v.1.2.9) with the default parameters[62]. Assembled contigs were then scaffolded using the scaffolding function from IDBA-UD[63] (v.1.1.3). Scaffold coverage values were calculated as the ratio of total length of mapped reads to the total length of the scaffold, using bowtie2 (v.2.3.5.1)[64] for mapping. Only scaffolds of >1 kb in length were considered for gene prediction and genome binning. Gene prediction was performed using Prodigal (v.2.6.3) using the 'meta' option[65] and genes were annotated using USEARCH[66] (v.10.0.240) against the KEGG[67,68], Uniref100 (ref. [69]) and UniProt[70] databases. 16S rRNA genes were identified using a custom HMM[2] (16SfromHMM.py, available at GitHub (https://github.com/christophertbrown/bioscripts)) and insertions of 10 bp or greater were removed. Prediction of tRNA genes was performed using tRNAscan-SE[71] (v.1.3.1).

**Genome binning, curation, dereplication and coverage calculation.** Scaffolds longer than 1 kb only were considered for protein annotation and binning. Scaffolds were binned on the basis of GC content, coverage, presence/copies of ribosomal proteins and single-copy genes, taxonomic profile, tetranucleotide frequency and patterns of coverage across samples. On ggKbase (https://ggkbase.berkeley.edu/), protein annotations were performed using USEARCH (v.10.0.240) against the KEGG, UniRef100 and UniProt databases as well as against an internal database comprised of publicly available genomes from NCBI. Scaffold taxonomic profiles were then determined on the basis of a voting scheme, whereby the winning taxonomic profile had to have more than 50% of protein 'votes' for each taxonomic rank on the basis of protein annotations. A combination of manual binning on ggKbase (https://ggkbase.berkeley.edu/) and automated binning using CONCOCT[72] (v.1.1.0), Maxbin2[73] (v.2.2.7) and Abawaca2 (v.1.07) was used to generate candidate bins for each sample. The best bins were determined using DASTool[34] (v.1.1.1) and manually checked using ggKbase to remove incorrectly assigned scaffolds according to the criteria listed above. Bacterial genomes were then filtered for completeness (>70%) using a set of 43 single copy genes previously used for the CPR[2,19], and archaeal genomes were filtered using 48 single-copy genes for DPANN. Contamination was assessed using checkM[74] (<10%; Supplementary Table 2). The program dRep[35] (v.2.5.3) was used to dereplicate genomes from each site at 99% ANI (strain level), resulting in a representative set of 2,007 genomes across all sites. The median estimated genome completeness of each site's representative set is over 90%, with 18–58% of each site's raw reads mapping back to the representative set (Supplementary Table 1). Singlefold coverage values for genomes were calculated as the ratio of the total length of mapped reads (bowtie2 v.2.3.5.1) to the total length of the genome.

**Phylogenetic classification.** Genomes with a clear taxonomic classification on the basis of the internal ggKbase database (>50% of the genome sequence had a clear scaffold-level taxonomic winner, based on best matches of protein sequences to those in genomes of a taxonomically comprehensive database) were classified according to their predicted ggKbase taxonomy. For genomes without a clear predicted ggKbase taxonomy, phylogenetic analysis was performed using several marker sets as follows: concatenated ribosomal proteins (encoded by a syntenic block of genes and selected to avoid binning error chimaeras), rpS3 proteins and 16S rRNA genes (for CPR bacteria). Reference sequences for all of the phylogenetic trees were taken from previously published studies that recovered many high-quality CPR and DPANN genomes[2,3,19,22].

The concatenated ribosomal protein set for bacteria includes 15 proteins (L2, L3, L4, L5, L6, L14, L15, L18, L22, L24, S3, S8, S10, S17 and S19), whereas the archaeal set includes 14 proteins (the bacterial set without S10, which is missing from many archaeal genomes). Ribosomal proteins were identified by searching predicted open reading frames (ORFs) against ribosomal protein databases using

USEARCH[66]. For each individual ribosomal protein, hits and reference sequences were aligned to the Pfam HMM model using hmmalign from HMMer[75] (v.3.3), alignments were converted from the Stockholm format to FASTA and insertions added by hmmalign were stripped. All individual ribosomal protein alignments were concatenated together, and concatenated sequences with an ungapped length of greater than 1,100 amino acid residues were combined with reference sequences to build a maximum-likelihood tree using IQ-Tree (v.1.6.12; iqtree -s <alignmentfile> -st AA -nt 48 -bb 1000 -m LG+G4+FO+I).

For *rpS3* gene phylogenetic analysis, *rpS3* genes were identified using a custom HMM with an HMM alignment score cut-off of 40 (ref. [36]). Identified *rpS3* genes were aligned with *rpS3* reference sequences using mafft[76] (using the default parameters) and columns with >95% gaps were removed with trimal[77]. The alignment was used to build a maximum likelihood tree using IQ-Tree (iqtree -s <alignmentfile> -st AA -nt 48 -bb 1000 -m LG+G4+FO+I).

For 16S rRNA gene phylogenetic analysis of CPR bacterial genomes, 16S rRNA genes were identified using a custom HMM[2] (using 16SfromHMM.py, available at GitHub (https://github.com/christophertbrown/bioscripts)) and insertions of 10 bp or greater were removed (using strip_masked.py from https://github.com/christophertbrown/bioscripts). Sequences with lengths of >800 bp were used for phylogenetic analysis. SSU-align was used to align 16S sequences from this study with reference sequences from the previous studies mentioned above as well as CPR bacteria sequences from SILVA database[78]. The resulting alignment was used to build a maximum-likelihood tree using RAxML-HPC BlackBox[79] (v.8.2.12) on the CIPRES Science Gateway[80] with the general time reversible model of nucleotide substitution (raxmlHPC-HYBRID -T 4 -s infile -N autoMRE -n result -f a -p 12345 -x 12345 -m GTRCAT).

Genomes forming the new phylum-level lineages 'Candidatus Genascibacteria' and 'Candidatus Montesolbacteria' were identified on the basis of the following criteria: (1) they formed a monophyletic group in the 16S rRNA gene phylogeny; (2) 16S rRNA genes shared less than 76% sequence identity to the closest representatives; (3) they were also supported by the concatenated ribosomal protein phylogeny; and (4) more than one representative draft genome was available. A list of 'Candidatus Genascibacteria' and 'Candidatus Montesolbacteria' genomes and ANI with closest 16S rRNA hits from SILVA[78] is provided in Supplementary Table 3.

**Ordination analysis.** Principal component analysis was performed on *rpS3* relative coverage values and on the metabolic capacities of whole communities (the summed relative coverage values of genomes encoding a particular metabolic transformation). Principal component analysis was performed using the FactoMineR package[81] and visualized using factoextra[82]. Relative abundance values were scaled to unit variance before the calculation of the principal components. NMDS analysis was performed on normalized read counts (reads per million total reads) for all genomes from Ag groundwater, based on read mapping with BBMap[60]. NMDS analysis was performed using the metaMDS function in the Vegan package for R[83], using the default parameters. In brief, the data were transformed using Wisconsin double standardization of the square root of the matrix, followed by construction of a Bray–Curtis dissimilarity matrix, then an NMDS with 20 random starts. Finally, the results were scaled to maximize variation to the first principal component. Results were visualized using the ggplot2 package for R[84].

**Assessing index hopping between Pr1 and Pr7.** Pr1 and Pr7 were the only pair of analysed sites that shared more than a few strains (44 pairs of genomes with >99% ANI). These two sites are separated by Putah Creek, a major hydrological feature, and so are unlikely to be fed from the same aquifer. Although DNA from Pr1 and Pr7 was sequenced on the same NovaSeq 6000 lane, we do not attribute this strain overlap to index hopping, as dual indexing was used and reads with mismatched indices were not analysed, reducing the already low incidence of index hopping (<2% of reads). Furthermore, although the 44 genome pairs share >99% ANI, they are not identical, differing in sequence by up to 10,000 bp per Mb of genome.

**Genome and community-level metabolic predictions.** To analyse the metabolic capacity of the sampled groundwater communities at both the genome and community level, the program METABOLIC[41] (v.4.0) was used to search predicted ORFs against a curated set of KEGG, TIGRfam, Pfam and custom HMM profiles corresponding to key marker genes for biogeochemical cycling. For specific sets of proteins that are often misannotated due to high sequence similarity despite divergent function (for example, *amoABC*/*pmoABC*), an additional motif-validation step was performed in which sequences were searched for conserved residue patterns indicative of either *amoABC* or *pmoABC*. On the basis of the presence/absence of this manually curated set of marker genes, the presence/absence of metabolic capacities encoded by each genome was determined, and the number and relative abundance of genomes in the community that encode a metabolic capacity were calculated. The biogeochemical cycling diagrams shown in Fig. 4 and Extended Data Fig. 4 are based off this manually curated set of key marker genes.

In addition to marker gene analysis, METABOLIC was also used to evaluate the completeness of KEGG modules for key biogeochemical cycling processes. In brief, the capacity of a genome for a broad metabolic function (for example, carbon fixation) was determined using the following steps:

1. The presence/absence of relevant genes (for example, either the large or small RuBisCo subunit, phosphoribulokinase, phosphoglycerate kinase) was determined by profiling against a custom set of HMMs, utilizing Kofam-suggested cut-off values for Kofam HMMs and custom cut-off values for TIGRfam, Pfam and custom HMMs. Custom cut-offs were chosen by adjusting noise cut-offs and trusted cut-offs to avoid potential false-positive hits[19].
2. The presence/absence of each reaction in the relevant KEGG module was determined by combinations of key genes (as defined by the KEGG database). For example, the KEGG reaction R00024 (the carboxylation of RuBP by RuBisCo) in the KEGG module M00165 (the Calvin–Benson–Bassham cycle) is considered present only if the genome contains a hit for either the large or small subunits of RuBisCo (KEGG entries K01601–K01602).
3. A given KEGG module was considered to be present if genes identified for >75% of the reactions in the module were present. This 75% cut-off was chosen to reflect the fact that MAGs, which are in most cases neither complete nor circularized (in our case, we have a 70% cut-off for genome completeness), will have incomplete metabolic pathways.
4. Finally, a genome was considered to have broad metabolic capacity (carbon fixation) if any relevant KEGG module was present (CBB pathway, 3HP cycle, 3HP/4HB cycle, Wood Ljungdahl pathway or reverse tricarboxylic acid cycle). The results from METABOLIC for each site are provided in Supplementary Tables 8–15.

**Cryo-TEM sample preparation in the field.** Cryo-TEM samples were prepared onsite at the Ag dairy farm on 5 February 2018. Approximately 30 l of pumped Ag groundwater was concentrated to a final volume of ~5 ml, using TFF (Millipore Pellicon Cassette Standard Acrylic Holder) with a 30 kDa ultrafiltration cassette (Millipore Pellicon 2 Biomax). Aliquots of 5 µl were taken directly from the suspensions and deposited onto 300 mesh lacey carbon coated Cu-grids (Ted Pella, 01895) that had been treated by glow discharge within 24 h. Grids were blotted with filter paper and plunged into liquid ethane held at liquid nitrogen temperatures using a portable, custom-built cryo-plunging device[85]. Plunged grids were stored in liquid nitrogen before transfer to the microscope and maintained at 80 K during acquisition of all datasets.

**Cryo-TEM imaging.** Imaging was performed using a JEOL–3100-FFC electron microscope (JEOL) equipped with a FEG electron source operating at 300 kV. An Omega energy filter (JEOL) attenuated electrons with energy losses that exceeded 30 eV of the zero-loss peak before detection by a Gatan K2 Summit direct electron detector. Dose-fractionated images were acquired with a pixel size of 3.41 Å px$^{-1}$ using a dose of 7.27 e$^-$ Å$^{-2}$ per frame. Data were collected using the Gatan Microscopy Suite (v.3.4.1) and SerialEM (v.3.7). Up to 30 frames per image were aligned and averaged using IMOD[86] (v.4.9) and image contrast was adjusted in ImageJ (v.2.0.0).

**Analysis of cell distribution across serial size filters.** To analyse the distribution of Ag cells across size fractions, we needed to estimate total cell counts, whereas sequencing data can generate only relative abundance values (in the absence of an internal standard). We began with the general equation: total cell count of a genome = relative abundance from sequencing × microbial load, an approach that has been discussed and tested in depth previously[87]. Our method takes the form: $c = x \times l \times m$ where $c$ is the total cell count of a genome; $x$ is the relative coverage of a genome; $l$ is the total cell counts of all community members per ng of DNA in the community; and $m$ is the ng of DNA extracted from the size fraction. The term $l \times m$ estimates microbial load, that is, the total cell count of all members in a community.

In our method, we utilize DNA yield (measured variable $m$ in our equation) as an estimate of microbial load in a sample. DNA yield is an imperfect estimate of true microbial load for a number of reasons, including potential ploidy[88] and bias in sequencing representation depending on the DNA extraction method[89]. However, there are also limitations and problems with other estimates of microbial load, such as flow cytometry-based cell counting[90]. Given that we extracted all samples in this study using the same DNA extraction kit and have fluorometry-based measurements of DNA yield (Qubit dsDNA HS Assay), we chose to use DNA yield as the best available measurement of microbial load.

Fluorometry-based quantification of DNA yield measures DNA mass (that is, the number of double-stranded DNA base pairs). Meanwhile, the relative abundance of a genome (relative coverage) is proportional to the relative fraction of total cells represented by the genome, rather than the relative fraction of total DNA represented by the genome. For example, a CPR genome with a relative abundance of 1% will constitute less than 1% of the total DNA yield from a groundwater community, owing to its smaller genome size than other members of the community. To account for genome-size-dependent DNA yield, we calculated how many microbial cells would correspond to 1 ng of DNA on the basis of the genome sizes of each genome recovered from the community (parameter $l$ in our equation). The molecular weight of each genome calculated as number of base pairs × 650 Da per base pair. The relative coverage of a genome in a given size fraction was calculated as the total length of reads mapping to the genome divided by the total length of the genome (mapping was performed with bowtie2)[64].

To find significant differences in cell counts between two given size fractions (that is, 2.5+ μm versus 0.1–0.2 μm), paired $t$-tests were performed on cell counts from each phylum with more than 5 representative genomes and with cell count distributions in each size fraction that did not deviate significantly from normality (assessed by plotting cell count distributions and performing a Shapiro–Wilks test).

**iRep analysis.** Instantaneous replication rates were calculated for Ag bacterial genomes using iRep[49] (v.1.1.14) with a tolerance of three mismatches per read. Reads from each size fraction were mapped to the bacterial genomes using bowtie2 (ref. [64]).

**Reporting Summary.** Further information on research design is available in the Nature Research Reporting Summary linked to this article.

## Data availability
NCBI accession numbers for metagenome reads and metagenome assembled genomes (BioProject: PRJNA640378) are provided in Supplementary Table 18. Metagenome assembled genomes are also available online (http://ggkbase.berkeley.edu/all_nc_groundwater_genomes; please note that it is necessary to register for an account by provision of an email address before download).

## Code availability
Identification of 16S rRNA genes and removal of insertions was performed using the custom scripts 16SfromHMM.py and strip_masked.py, which are available in the ctbBio Python package (https://github.com/christophertbrown/bioscripts/blob/master/ctbBio/16SfromHMM.py and https://github.com/christophertbrown/bioscripts/blob/master/ctbBio/strip_masked.py). Identification of rpS3 genes was performed using an HMM trained as previously described[36] on a published alignment of rpS3 sequences from across the tree of life3. The custom HMMs used to identify key genes in metabolic cycling are available in the METABOLIC program (https://github.com/AnantharamanLab/METABOLIC).

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

## Acknowledgements

We thank E. Genasci, A. Book, K. Kritikos, K. Fults, D. Fults, P. Smith and J. Smith for access to their groundwater wells; C. J. Castelle, L. Valentin, B. Al-Shayeb, K. Lane, M. Olm, N. Oberleitner, R. Méheust, A. Jaffe, J. West-Roberts, A. Probst and D. Geller-McGrath for their assistance with field work; C. J. Castelle and A. Jaffe for assistance with archaeal and CPR phylogenetic analysis, respectively; and E. Montabana for help with collection of cryo-TEM data. C.H. was supported by a Camille and Henry Dreyfus Foundation Postdoctoral Fellowship in Environmental Chemistry. Funding for groundwater sampling and sequencing was provided by the Innovative Genomics Institute, the Allen Foundation and the Chan Zuckerberg BioHub.

## Author contributions

C.H. and J.F.B. designed the study. C.H., R.K. and I.F.F. collected samples, extracted DNA and performed manual binning. C.H. and R.K. performed on-site TFF filtration. C.H. performed automated binning, bin selection, bin curation and dereplication, phylogenetic analysis, abundance and community comparison analysis, metabolic analysis, cell count distribution analysis and on-site TFF filtration. M.L.W. prepared cryo-TEM grids and collected cryo-TEM data. R.K. performed ordination analysis. C.H. and J.F.B. drafted the manuscript. All of the authors reviewed the manuscript.

## Competing interests

The authors declare no competing interests.

## Additional information

**Extended data** is available for this paper at https://doi.org/10.1038/s41564-020-00840-5.

**Correspondence and requests for materials** should be addressed to J.F.B.

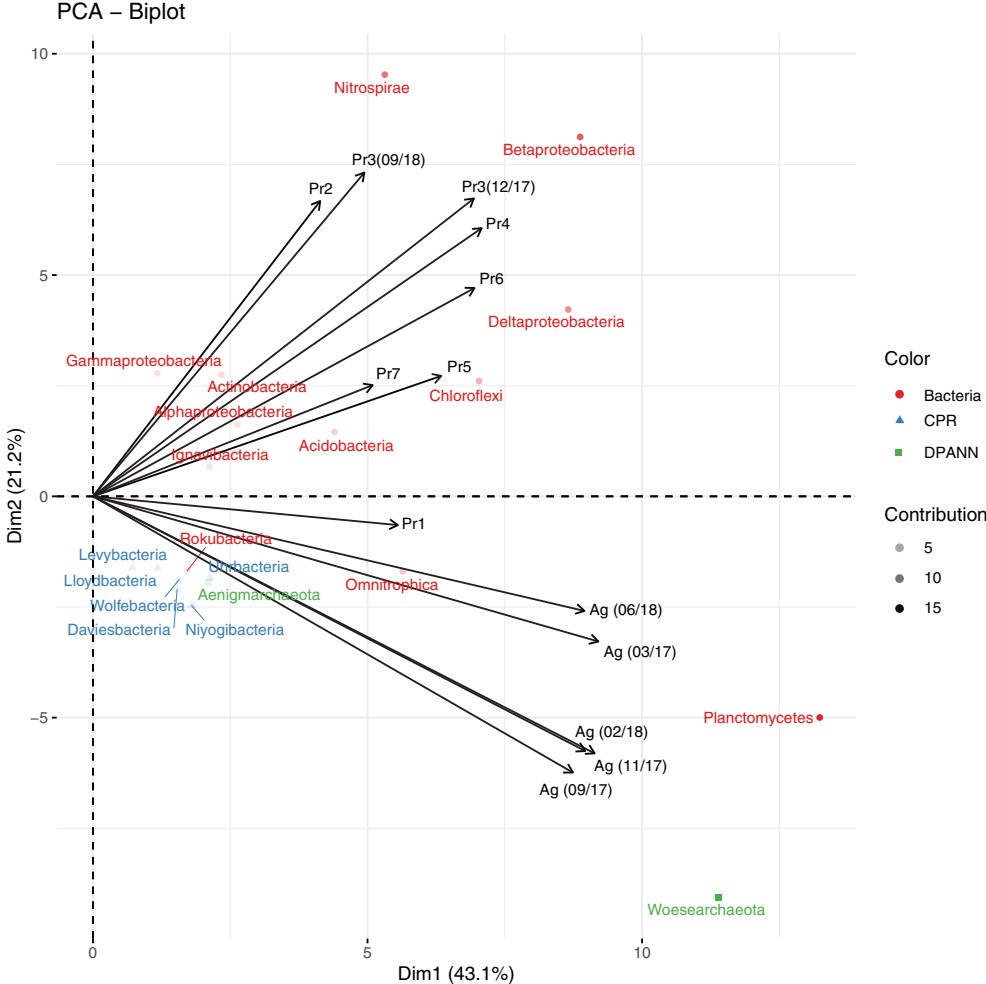

**Extended Data Fig. 1 | A biplot representation of phylum-level lineages in ordination space.** Arrows show the direction of greatest gradient change according to site. The transparency of the points reflects the contribution of the phylum to the principal components.

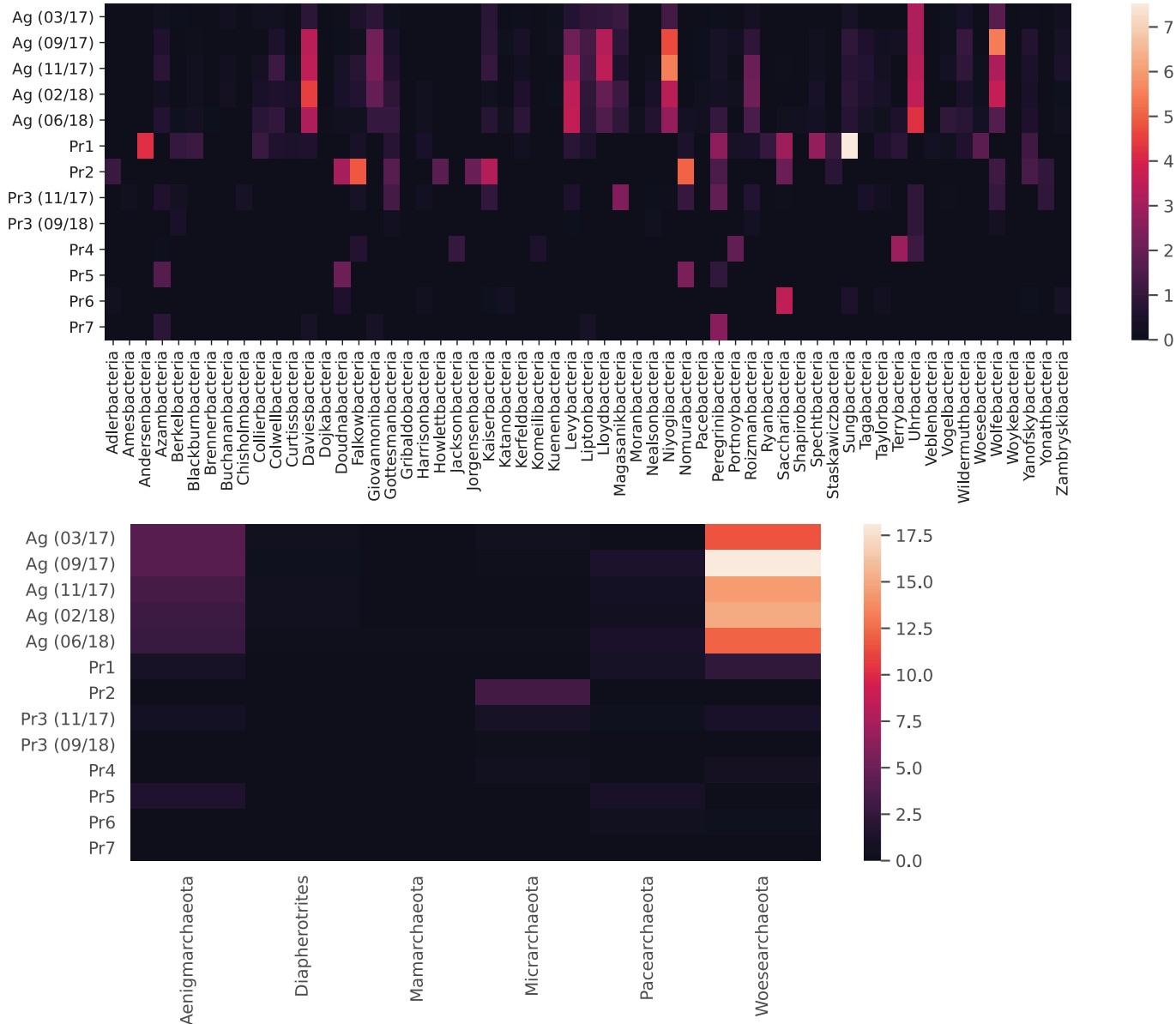

**Extended Data Fig. 2 | Distribution of CPR (top) and DPANN (bottom) phylum-level lineages across groundwater sites.** Color/legend indicate relative coverage values (percentages).

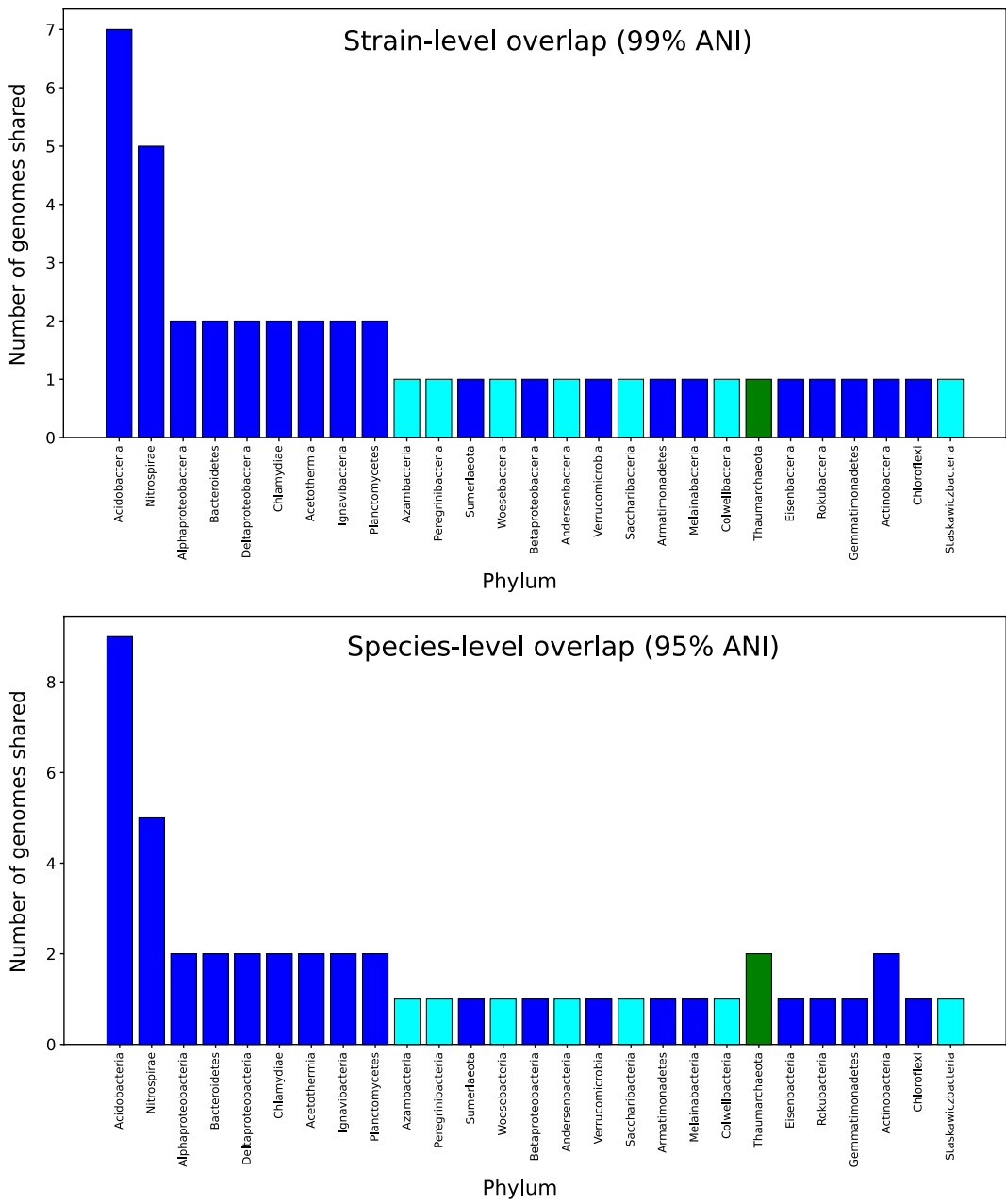

**Extended Data Fig. 3 | Genome similarity at the strain (>99% ANI) and species (>95% ANI) level between Pr1 and Pr7 genomes.** Blue bars indicate non-CPR bacteria, aqua bars indicate CPR bacteria, and green bars indicate archaea. The magnitude of the y-axis indicates the number of genomes shared according between the two sites according to the ANI threshold.

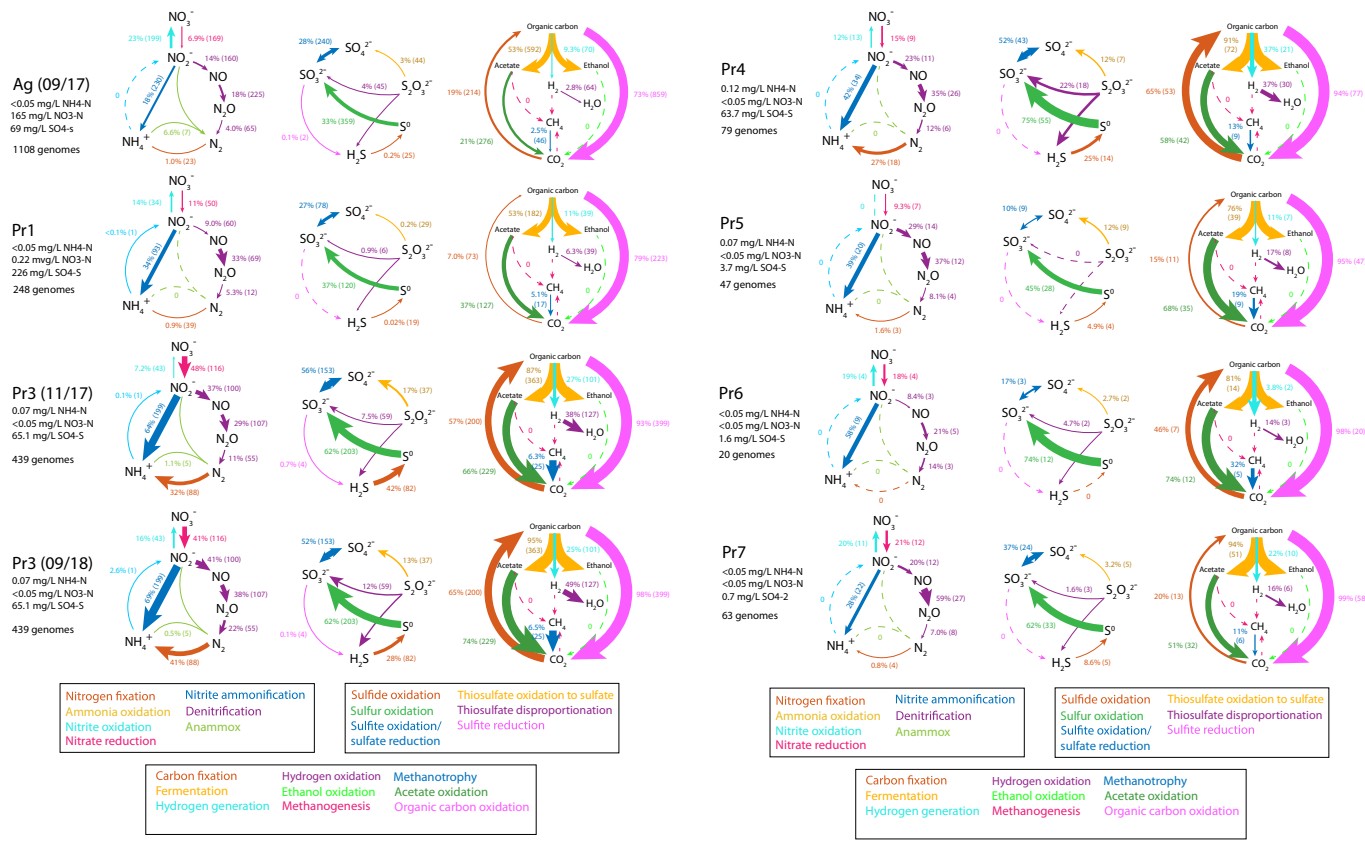

**Extended Data Fig. 4 | Community-level cycling of nitrogen, sulfur, and carbon in the eight groundwater communities sampled in this study.** Listed next to each metabolic step are the total relative abundance of all genomes capable of carrying out the step, and the number of genomes containing the capacity for that step. Arrow sizes are drawn proportional to the total relative abundance of genomes capable of carrying out the metabolic step.

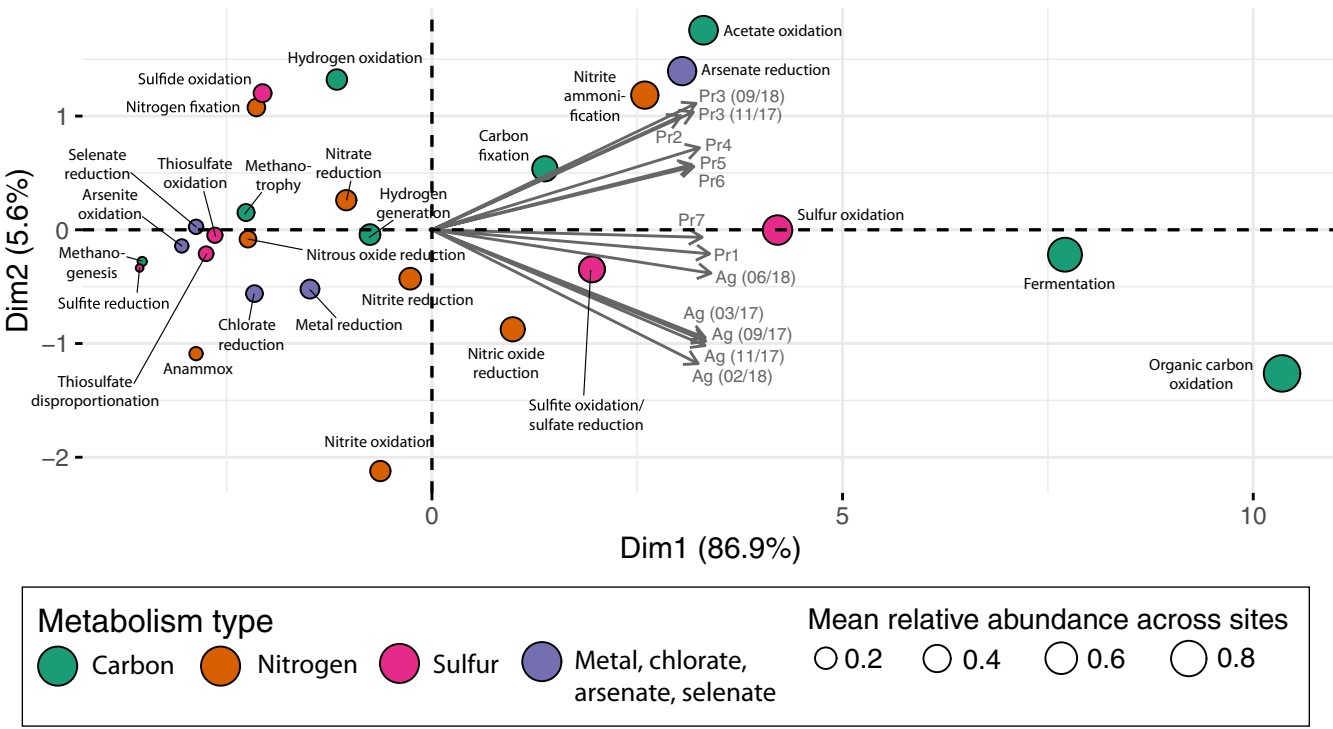

**Extended Data Fig. 5 | Depiction of total relative coverages of different metabolic capacities, in principal component space.** Principal component analysis was performed on the relative abundance of organisms with specific metabolic capacities across all sites.

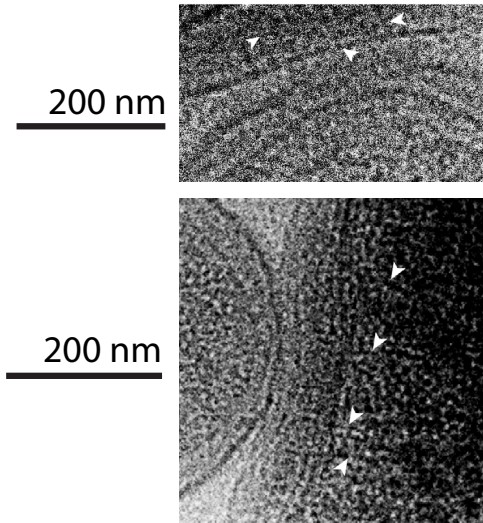

**Extended Data Fig. 6 | Zoomed in view of cryo-TEM images of ultra-small cells connected to host cells with pili-like appendages in *Ag* groundwater (Fig. 5) concentrated by tangential flow filtration.** White arrows indicate pili-like appendages extending into the host from the ultra-small cell.

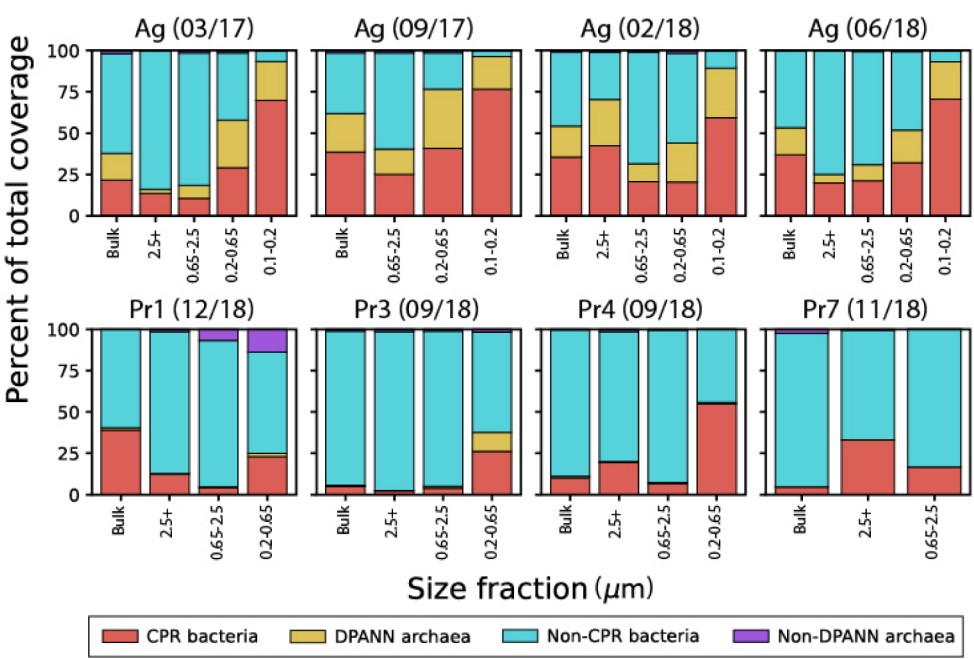

**Extended Data Fig. 7 |** Relative coverage for CPR bacteria, non-CPR bacteria, DPANN archaea, and non-DPANN archaea genomes in all size fractions sequenced in this study.

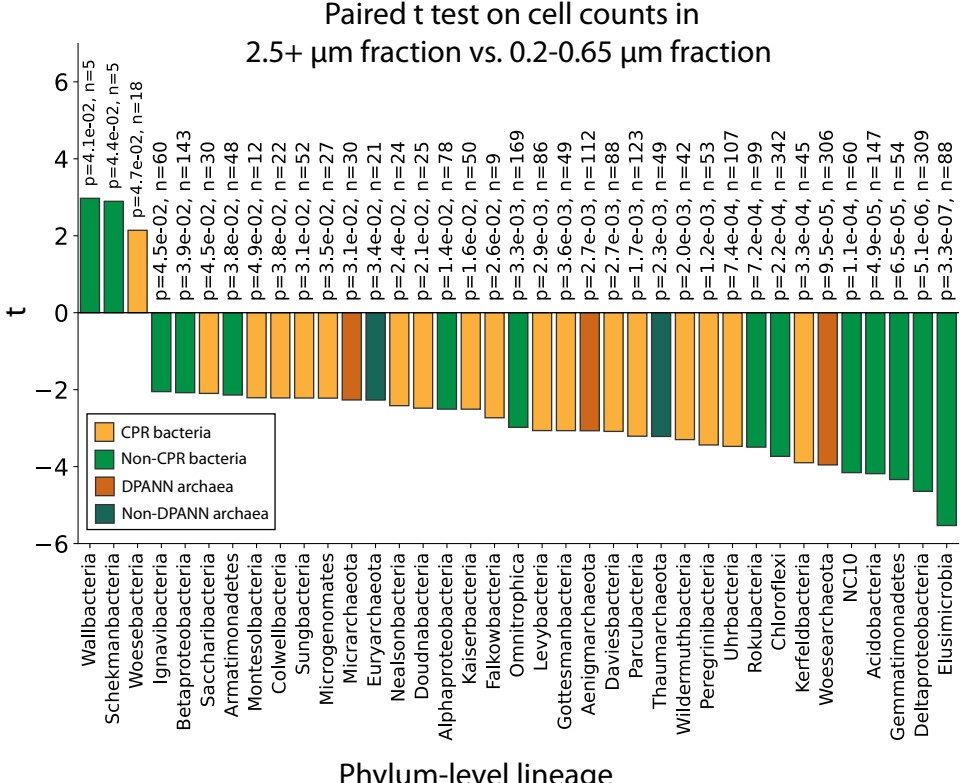

**Extended Data Fig. 8 | Results from a paired t-test (two-tailed) on estimated cell counts of genomes in the 2.5+ μm versus the 0.2–0.65 μm size fractions after serial size filtration.** A positive t statistic indicates enrichment on the 2.5+ μm compared to the 0.2–0.65 μm size fraction. Values listed by each bar are the calculated p value (top value) and sample size (bottom value) for each phylum-level lineage.

# nature research

# Reporting Summary

Nature Research wishes to improve the reproducibility of the work that we publish. This form provides structure for consistency and transparency in reporting. For further information on Nature Research policies, see our Editorial Policies and the Editorial Policy Checklist.

## Statistics

For all statistical analyses, confirm that the following items are present in the figure legend, table legend, main text, or Methods section.

| n/a | Confirmed | |
|---|---|---|
| ☐ | ☒ | The exact sample size (*n*) for each experimental group/condition, given as a discrete number and unit of measurement |
| ☐ | ☒ | A statement on whether measurements were taken from distinct samples or whether the same sample was measured repeatedly |
| ☐ | ☒ | The statistical test(s) used AND whether they are one- or two-sided<br>*Only common tests should be described solely by name; describe more complex techniques in the Methods section.* |
| ☒ | ☐ | A description of all covariates tested |
| ☐ | ☒ | A description of any assumptions or corrections, such as tests of normality and adjustment for multiple comparisons |
| ☐ | ☒ | A full description of the statistical parameters including central tendency (e.g. means) or other basic estimates (e.g. regression coefficient) AND variation (e.g. standard deviation) or associated estimates of uncertainty (e.g. confidence intervals) |
| ☐ | ☒ | For null hypothesis testing, the test statistic (e.g. *F*, *t*, *r*) with confidence intervals, effect sizes, degrees of freedom and *P* value noted<br>*Give P values as exact values whenever suitable.* |
| ☒ | ☐ | For Bayesian analysis, information on the choice of priors and Markov chain Monte Carlo settings |
| ☒ | ☐ | For hierarchical and complex designs, identification of the appropriate level for tests and full reporting of outcomes |
| ☒ | ☐ | Estimates of effect sizes (e.g. Cohen's *d*, Pearson's *r*), indicating how they were calculated |

*Our web collection on statistics for biologists contains articles on many of the points above.*

## Software and code

Policy information about availability of computer code

| | |
|---|---|
| Data collection | Gatan Microscopy Suite (v 3.4.1), SerialEM (v 3.7) |
| Data analysis | BBTools (v 38.78); Sickle (v 1.33); MEGAHIT (v 1.2.9); IDBA-UD (v 1.1.3); bowtie2 (v 2.3.5.1); Prodigal (v 2.6.3); usearch (v 10.0.240); 16SfromHMM.py (available at https://github.com/christophertbrown/bioscripts); tRNAscan-SE (v 1.3.1); CONCOCT (v 1.1.0); Maxbin2 (v 2.2.7); Abawaca (v 1.07); DASTool (v 1.1.1); ggkbase (https://ggkbase.berkeley.edu/); dRep (v 2.5.3); METABOLIC (v 1.3); IMOD (v 4.9); ImageJ (v 2.0.0); iRep (v 1.1.14); |

For manuscripts utilizing custom algorithms or software that are central to the research but not yet described in published literature, software must be made available to editors and reviewers. We strongly encourage code deposition in a community repository (e.g. GitHub). See the Nature Research guidelines for submitting code & software for further information.

## Data

Policy information about availability of data

All manuscripts must include a data availability statement. This statement should provide the following information, where applicable:

- Accession codes, unique identifiers, or web links for publicly available datasets
- A list of figures that have associated raw data
- A description of any restrictions on data availability

SRA accession numbers for metagenome reads are in SI Table 18. All metagenome-assembled genomes from this study are deposited in NCBI under Bioproject PRJNA640378. The genomes are also available at: http://ggkbase.berkeley.edu/all_nc_groundwater_genomes (please note that it is necessary to register for an account by provision of an email address prior to download).

# Field-specific reporting

Please select the one below that is the best fit for your research. If you are not sure, read the appropriate sections before making your selection.

☐ Life sciences      ☐ Behavioural & social sciences      ☒ Ecological, evolutionary & environmental sciences

For a reference copy of the document with all sections, see nature.com/documents/nr-reporting-summary-flat.pdf

# Ecological, evolutionary & environmental sciences study design

All studies must disclose on these points even when the disclosure is negative.

| | |
|---|---|
| Study description | This study performs genome-resolved metagenomics analysis and cryo-electron microscopy on bacterial and archaeal communities in 8 groundwater sites. |
| Research sample | For the metagenomics portion of the study, the research samples are metagenomes sequenced from 8 groundwater sites in northern California. For the cryo-electron microscopy, the sample is groundwater from one of these sites concentrated by tangential flow filtration. |
| Sampling strategy | At each site, 400-1200 L of groundwater (planktonic portion) was pumped onto filters from which DNA was extracted. For cryo-electron microscopy, 20 L of groundwater was pumped and concentrated to <5 mL using tangential flow filtration. |
| Data collection | Extracted DNA was sequenced on either HiSeq 4000 or NovaSeq 6000 platforms, at either the California Institute for Quantitative Biosciences' (QB3) genomics facility or the Chan Zuckerberg Biohub's sequencing facility. |
| Timing and spatial scale | Groundwater sampling dates by site: Ag (03/17, 09/17, 11/17, 02/18, 06/18), Pr1 (12/18), Pr2 (05/19), Pr3 (11/17, 09/18), Pr4 (09/18), Pr5 (05/19), Pr6 (05/19), Pr7 (11/18). |
| Data exclusions | No data were excluded from analysis. |
| Reproducibility | No explicit measures were taken to ensure reproducibility of assembled genomes from each site. Time series sampling of Ag groundwater show that similar genomes are recovered from each time point. |
| Randomization | Genomes were taxonomically classified based on a phylogenetic tree of concatenated ribosomal proteins, allowing us to categorize genomes as CPR bacteria, non-CPR bacteria, DPANN archaea, and non-DPANN archaea. |
| Blinding | Blinding was not relevant to our study. |

Did the study involve field work?      ☒ Yes      ☐ No

## Field work, collection and transport

| | |
|---|---|
| Field conditions | All groundwater was pumped from shallow wells (<100 m deep). |
| Location | Sites Pr1 through Pr7 are located in Lake/Napa County, California, while site Ag is located in Modesto, California. |
| Access & import/export | Private sites were sampled with explicit permission from the property owner. |
| Disturbance | To our knowledge, our groundwater sampling did not cause any disturbance. |

# Reporting for specific materials, systems and methods

We require information from authors about some types of materials, experimental systems and methods used in many studies. Here, indicate whether each material, system or method listed is relevant to your study. If you are not sure if a list item applies to your research, read the appropriate section before selecting a response.

## Materials & experimental systems

| n/a | Involved in the study |
|-----|----------------------|
| ☒ | Antibodies |
| ☒ | Eukaryotic cell lines |
| ☒ | Palaeontology and archaeology |
| ☒ | Animals and other organisms |
| ☒ | Human research participants |
| ☒ | Clinical data |
| ☒ | Dual use research of concern |

## Methods

| n/a | Involved in the study |
|-----|----------------------|
| ☒ | ChIP-seq |
| ☒ | Flow cytometry |
| ☒ | MRI-based neuroimaging |

