## [Peer Review file · Nature Microbiology]

Peer Review Information

Journal: Nature Microbiology

Manuscript Title: **Genome-resolved metagenomics reveals huge and site-specific diversity of episymbiotic CPR bacteria and DPANN archaea in groundwater ecosystems**

Corresponding author name(s): Jillian Banfield

Ts

Reviewer Comments & Decisions:

Decision Letter, initial version:

Dear Jill,

Thank you for your patience while your manuscript "Huge and variable diversity of episymbiotic CPR bacteria and DPANN archaea across groundwater ecosystems" was under peer-review at Nature Microbiology. It has now been seen by 3 referees, whose expertise and comments you will find at the of this email. Although they find your work of some potential interest, they have raised a number of concerns that will need to be addressed before we can consider publication of the work in Nature Microbiology.

In particular, referee #1 notes that "I did not see any evidence that diverse CPR and DPANN archaea are episymbionts in Ag groundwater, and no images were provided from other groundwater systems." This should be addressed with improved TEM images with more evidence to link these aspects. Other concerns from the referees include additional methods details and improved reporting of the experimental design, clarifying that repeated sampling rather than a time series was used, toning down of some statements, improved presentation of the genomes and metabolic potential, and additional phylogenetics analyses.

Should further experimental data allow you to address these criticisms, we would be happy to look at a revised manuscript.

Please include a data availability statement as a separate section after Methods but before references, under the heading "Data Availability". This section should inform readers about the availability of the

data used to support the conclusions of your study. This information includes accession codes to public repositories (data banks for protein, DNA or RNA sequences, microarray, proteomics data etc...), references to source data published alongside the paper, unique identifiers such as URLs to data repository entries, or data set DOIs, and any other statement about data availability. At a minimum, you should include the following statement: "The data that support the findings of this study are available from the corresponding author upon request", mentioning any restrictions on availability. If DOIs are provided, we also strongly encourage including these in the Reference list (authors, title, publisher (repository name), identifier, year). For more guidance on how to write this section please see: <http://www.nature.com/authors/policies/data/data-availability-statements-data-citations.pdf>

* If you have not done so already we suggest that you begin to revise your manuscript so that it conforms to our Article format instructions at <http://www.nature.com/nmicrobiol/info/final-submission>. Refer also to any guidelines provided in this letter.

{REDACTED}-

Note: This url links to your confidential homepage and associated information about manuscripts you may have submitted or be reviewing for us. If you wish to forward this e-mail to co-authors, please delete this link to your homepage first.

Nature Microbiology is committed to improving transparency in authorship. As part of our efforts in this direction, we are now requesting that all authors identified as 'corresponding author' on published papers create and link their Open Researcher and Contributor Identifier (ORCID) with their account on the Manuscript Tracking System (MTS), prior to acceptance. This applies to primary research papers

only. ORCID helps the scientific community achieve unambiguous attribution of all scholarly contributions. You can create and link your ORCID from the home page of the MTS by clicking on 'Modify my Springer Nature account'. For more information please visit www.springernature.com/orcid.

If you wish to submit a suitably revised manuscript we would hope to receive it within 6 months. If you cannot send it within this time, please let us know. We will be happy to consider your revision, even if a similar study has been accepted for publication at Nature Microbiology or published elsewhere (up to a maximum of 6 months).

Reviewer Expertise:

Referee #1: groundwater microbial communities, metagenomics, biogeochemistry

Referee #2: archaea, metagenomics, phylogenetics

Referee #3: subsurface microbial communities, 'omics, biogeochemical cycles

Reviewer Comments:

Reviewer #1 (Remarks to the Author):

This is another interesting paper by Christine He and her co-authors. They performed genome-resolved metagenomics of eight groundwater microbiomes in California which yielded 746 dereplicated CPR and DPANN genomes, which is awesome.

One major finding of this story is that little species-level genome overlap exists between the sites according to distinct physicochemical conditions at each site, but maybe also according to host populations. To investigate this relationship, the authors profiled all recovered genomes against KEGG and custom protein HMM databases which yielded metabolic profiles of the whole groundwater communities at each site. Surprisingly, heatmap of all CPR and DPANN genomes did not show a clear clustering of key genes required for various metabolic and biosynthetic functions. Nonetheless, many CPR and DPANN lineages appear to have a complementary or accessory role in different steps of denitrification, nitrite reduction to ammonia. A few genomes encode for sulfur dioxygenase *sdo*, and others have genes involved in sulfate reduction, suggesting a potential role of CPR and DPANN organisms in transformations to sulfite.

Based on these results, I am not convinced that the high level of differentiation between groundwater communities reflects species adaptation to different physiochemical conditions as the authors suggested (Page 16, line 419). It would be very helpful, if the authors could provide some additional information here.

One agriculturally-impacted, river sediment-hosted aquifer (site Ag) was sampled 5 times across a 15 months period. Relative abundance over time for non-CPR bacteria and CPR bacteria showed a peak of Planctomycetes which coincided with a low CPR abundance and vice versa suggesting a possible parasitic CPR-host relationship. A potential commensal or mutualistic CPR-host relationship was suggested with two Ignavibacteria and Betaproteobacteria organisms. But temporal resolution was too low to identify potential hosts.

To me, the coolest part of this paper is the cryo-TEM imaging which shows pili-mediated episymbiotic interactions between ultra-small cells and host cells. At the ultra-small cell/host contact region, the host cell envelope appears to be thickened. One host cell has multiple ultra-small bacterial cells directly attached to its cell envelope which appear to be in the process of dividing, raising the possibility that replication is correlated with host attachment. Other ultra-small cells were observed to be associated with/in close proximity to lysed cells, suggesting that some organisms scavenge resources (especially lipids or lipid building blocks) from dead cells or act parasitically from host organisms. These images also suggest that some ultra-small bacteria have cell envelopes that do not resemble those of Gram-negative bacteria, but seemingly can attach to Gram-negative hosts. Unfortunately, ultra-small cells and host cells can not be identified in TEM images. Thus, there is no final proof that ultra-small cells are indeed CPR. Thus, I would recommend to adjust the wording in the text.

The authors conclude that cryo-TEM imaging shows that some CPR bacteria in Ag groundwater are episymbionts of prokaryotic hosts (Page 12, line 301). Serial size filtration for fractionating CPR and DPANN cells according to host attachment identified two lineages, Kerfeldbacteria in the CPR and Pearchaeota within DPANN, with likely strong physical attachment to hosts. However, the title promises much more: "Huge and variable diversity of episymbiotic CPR bacteria and DPANN archaea across groundwater ecosystems". I did not see any evidence that diverse CPR and DPANN archaea are episymbionts in Ag groundwater, and no images were provided from other groundwater systems. Thus, the title should be changed to better reflect the much more restricted findings of this story. Some parts should be rewritten.

Another recent study on groundwater CPR and DPANN organisms claims that their cell-cell association experiments show lack of physical associations between most Patescibacteria or DPANN with other microorganisms (preprint <https://www.biorxiv.org/content/10.1101/2020.04.07.029462v1>). In addition, general genome features of Patescibacteria and DPANN do not seem to provide convincing evidence of an obligate symbiotic lifestyle. These authors state that the unusual genomic features of these organisms and prevalent auxotrophies may be a result of minimal cellular energy transduction mechanisms that potentially precede the evolution of respiration, thus relying solely on fermentation for energy conservation.

As the findings of both studies are somehow contradictory, and cell attachment could only be shown for some ultra-small cells from site Ag, it would be nice to comment on potential alternative explanations.

In general, integration of other findings would also improve the discussion part which is rather short and a little bit disappointing, as many arguments were already discussed in the results section. I would also love to see more findings from other groups studying the groundwater microbiome. The fraction of self-citations is really very high.

Specific comments:

Page 3, line 73, see also line 91: When you used the word "time-series", I expected a much higher temporal resolution over 15 months than just 5 time points. Samplings were also not done following a

regular sampling campaign, as samples were taken in March 2017, September 2017, November 2017, February 2018, June 2018. What you show is a temporal patterns over 15 months, which is great as you used a genome-centric approach. But please avoid the term time-series throughout the manuscript when analyzing only 5 time points.

Any idea why the first sample shows the lowest abundance of CPR?

Page 7, Fig. 2: 10. How did you calculate relative abundance? Just % of total reads mapping to a specific MAG?

Page 8, Fig. 3: Due to the lack of electron transport systems, fermentation in remains the main process for energy conservation, which is consistent with your results on metabolic profile Heatmap in Fig3B. But separating metabolic profile results for MAGs recovered from Pr and Ag sites may give more detailed insights. Can you add some more information here?

Page 11, line 271: You try to link the higher increase in metabolic capacities during the 2016-17 compared to the 2017-18 rainy season with a major difference in rainfall. However, groundwater table fluctuations always show a time lag to precipitation events. Please give more details on the dynamic changes of groundwater level in the groundwater well over the investigation period. Otherwise, this comment is highly speculative.

Page 11, line 288: How representative are the figures presented? How many TEM samples were analyzed and how many ultra-small cells did not show attachment? According to Methods (Page 20, line 529) you analyzed one groundwater sample from site Ag obtained at February 2018 which has a high relative abundance of CPR and DPANN. As you claim in the title that diverse CPR and DPANN are episymbionts, you should provide some statistical analyses. Please provide more details.

Page 14, line 357: I do not think that the title of the figure legend is correct. The authors assume that some CPR are attached to their hosts. CPR lineage Kerfeldbacteria and DPANN lineage Pacearchaeota are enriched on the 2.5+ μm size fraction relative to the 0.1-0.2 μm size fraction indicating that a high fraction of these populations is host attached. Please rephrase.

Page 15, line 388: Tracking iRep values of CPR genomes across size fractions. Why was the time point 11/17 omitted in Fig. 6B, but shown in 6C?

Fig. 6 C. The differences in iRep are small. You just analyzed 5 time points. Is it not a little bit too exaggerating to infer that CPR cell replication, stimulated by host attachment, is more prevalent during the rainy season compared to the dry season? Please also rephrase this part in the discussion (Page 17, line 446) (See also comment above).

Page 16, line 401. The first sentences are more a summary and are not needed. Please shorten this paragraph.

Page 17, line 443: Please provide some information about the heavy input of carbon and nitrogen from agricultural waste into Ag groundwater. How does this input vary over time?

Page 19, line 505: It would be nice to provide summary of internal ggKbase database used for taxonomic classifications. It seems that it is not accessible.

Page 19, line 493: The authors don't give the quality scores they used to trim the reads with. Not sure if sickle has some defaults, but that info is usually included.

In addition, they should explain more about the scaffolding with IDBA & MEGAHIT. Usually you use one or the other. They should also clarify that whether they assembled each sample independently? Were there replicates, how were these handled? Mapping scaffolds back to reads by bowtie2 does not by default gives scaffold coverage. How these values were calculated?

Page 19, line 498: Functional annotations of predicted ORFs was done by aligning with different databases eg. KEGG, Uniref100, Uniprot etc. but what was the criteria or cutoff for predicting good hit versus bad alignment hit?

Page 20, line 549: "The coverage of a genome is proportional to cell counts, so we assumed that a genome's relative coverage is the fraction of total cells in the community represented by that genome. Therefore, the cell count of a genome was calculated as the product of the genome relative coverage and estimated total cells in the community." Do you have any citation to support this method ?

Fig. 1A: The image from Google maps is of really low quality and hard to read. Show another map or move it to Supplemental material with a higher quality.

Fig. 3: Please convert units from mg/L to μM . In addition, yellow on white is hard to read. Please improve the color contrast.

SI Fig. 4: Please convert units from mg/L to μM . In addition, yellow on white is hard to read. Please improve the color contrast.

Reviewer #2 (Remarks to the Author):

In the manuscript by He et al., entitled "Huge and variable diversity of episymbiotic CPR bacteria and DPANN archaea across groundwater ecosystems", the authors reconstruct 746 CPR and DPANN genomes from one agricultural and several pristine groundwater samples and infer the metabolic potential of the symbionts as well as metabolic processes on community level. Furthermore, the authors compare community structure and chemistry variation across sites, try to identify potential host groups and use Cryo-TEM to study the potential of host-symbiont interactions. Finally, they also assess the level of host attachment across various DPANN and CPR lineages by estimating cell abundances based on genomic abundance data.

The study is very comprehensive, represents an impressive amount of work and is well written. While the authors have previously reported on large amounts of DPANN and CPR bins from other sites, the presence and variation of these lineages in groundwater ecosystems, which in part provide drinking water, have so far not been analysed in detail. Finding a diversity of CPR and DPANN in these systems could potentially be interesting with regard to human disease because certain members of these groups seem to be part of the human microbiome.

However, I have a number of comments regarding methodology and their effect on results/conclusions, which the authors should take into consideration.

Major comments:

Current flow of the manuscript

My impression is that the extensive analyses presented in this study, while very interesting, are sometimes a bit disconnected. Would it be possible to more clearly phrase the major questions/aims and use these as a thread throughout the manuscript? Further, the method section in the main text is very short and I would highly recommend that the method section of the Supplementary material is being moved to the main document.

Metabolic inferences:

In general, it is currently unclear, which exact enzymes the authors looked at to infer presence/absence and abundances. To make this analysis more reproducible and transparent, I would recommend that the authors provide supplementary tables, reporting the presence/absence pattern of all investigated marker proteins (for each functional category) and their COGs/KOs/PFAMs/Interpro-domains (e.g. those used to draw the metabolic profiles, e.g. in Figure 3A and B) across the metagenomes and MAGs analysed.

Further, it would be useful to list the read coverage of these marker proteins that were used to draw the figures (Fig. 3 (A and B) and SI Fig. 4). Or perhaps, I missed these files?

Did you do any manual inspections to assess presence/absence of metabolic modules and verify inferences? The rule of counting a KEGG module as present whenever 75% of its genes were present could result in "false positives" leading to incorrect inferences. Furthermore, for certain pathways, such as the 3-Hydroxypropionate and related carbon fixation pathways this may be difficult, because enzymes belong to broad protein families which may rather participate in other pathways. In turn, I am wondering whether you took into account or give higher credit to the presence of key enzymes specific to a certain pathway? For instance, even if more than 75% of the enzymes of the Wood-Ljungdahl

pathway are present, this pathway may be absent if the key enzyme Acetyl-CoA synthase/ Carbon monoxide dehydrogenase is lacking and may have functions other than carbon fixation (see for instance Zhuang et al. 2014, PNAS, Incomplete Wood–Ljungdahl pathway facilitates one-carbon metabolism in organohalide-respiring *Dehalococcoides mccartyi*).

Finally, I had one question regarding ammonia oxidation: Figure 3 and SI Figure 4 suggest that you could not detect any genes coding for proteins involved in ammonia oxidation. I found this a bit surprising/interesting because according to SI Table 2 (SI Table 2: Completeness and contamination estimates for all 2,007 dereplicated genomes in this study), your samples contain several MAGs assigned to putative ammonia oxidizing archaea such as Nitrosoarchaea/ Thaumarchaeota. Can you verify, that these MAGs are indeed Thaumarchaeota and if so, check whether they encode amo genes?

Cell distribution across filters

I had two concerns regarding the calculation of total cell counts:

First of all, I am not sure about the validity of the assumption that genome coverage is proportional to cell counts because microorganisms are known to have various levels of ploidy that makes it hard to compare cell numbers based on nucleic acid content, especially in complex microbial communities, see for instance Soppa, 2014: "Polyploidy in Archaea and Bacteria: About Desiccation Resistance, Giant Cell Size, Long-Term Survival, Enforcement by a Eukaryotic Host and Additional Aspects".

Furthermore, regarding the calculation of cells counts based on the product of genome relative coverage and estimated total cells in the community: I was wondering to what extent cells that did not lyse bias this analysis.

Did you try to estimate the amount of lysed versus non-lysed cells across samples?

Phylogenetic analyses

Protein trees: Are fasta files (including reference sequences), alignments files and single gene trees for each of these ribosomal proteins available, i.e. did you perform single protein trees to confirm that the identified proteins indeed represent ribosomal proteins and not distant homologs? How were the sequences of the new MAGs aligned with the reference sequences? Was any trimming performed? To avoid misalignments, it is recommended to first align all single protein homologs of a certain ribosomal protein (from both MAGs and reference species), eventually trim the single protein alignments (and perform single protein trees to identify potential distant paralogs, which, if present, should be removed before realigning/trimming) and then concatenate all single protein alignments for phylogenetic analyses. Considering the large diversity of organisms in these phylogenetic analyses, it may also be useful to test models and use a model that best fits the data. E.g. it is highly likely that mixture models (implemented in IQ-tree) would yield more likely phylogenetic trees which would help to evaluate the support for monophyly of the new phyla suggested in this study. To this end, I would also recommend to estimate branch support using single branch test as fast bootstraps can overestimate support.

It would also be helpful, if the authors could provide additional phylogenetic trees that include MAGs of the other communities members outside the DPANN/CPR, as these are important with regard to potential hosts as well as for the metabolic inferences of total communities.

16S rRNA: I do not understand why 16S rRNA trees were inferred with a protein model "LG protein substitution matrix". Can you check whether this is correct and if so, redo the analyses using a nucleic acid model?

Proposal of new phyla

I was wondering whether the criteria you used for determining the status of a new phylum are sufficient: e.g. It would be great if you mention the ANI thresholds used to support the status of a new phylum in the methods section and mention how this compares to suggested thresholds? Furthermore, please evaluate the phylum-level status of these lineages based on results of the suggested phylogenetic inferences (see above). In particular, the protein phylogeny should be taken into account because concatenated protein trees are usually more reliable to accurately place symbionts in phylogenetic trees (perhaps, this phylogeny would be more robust, if more proteins were concatenated). Further, I am wondering what you mean by point 4 (line 910): one representative draft genome? Was this a draft

genome of high quality? I would recommend that you mention the genome completeness/quality in Table S3 and also report ANI and support values from the two phylogenetic analyses.

Pili

I was surprised reading that the authors infer that all cells with pili-like structures belong to CPR but not to DPANN. In Comolli, 2009, the authors concluded that appendages on ARMAN cells likely represent pili. Furthermore, Hamm identified putative T4 pili proteins in Nanohaloarchaeota (Hamm et al., 2018) and St John, 2019 identified flagella-like structures in Nanoclepta.

In turn, I was wondering, whether you may have missed archaeal pili proteins? Which protein families did you look for? Please note that archaeal T4 pili encoding proteins are hard to identify as they differ from bacterial ones. A nice review on this can be found in Makarova et al., 2017, Diversity and Evolution of Type IV Pili Systems in Archaea. Some candidates for archaeal pilin proteins such as VirB11 seem to be present in DPANN based on pfam domains and perhaps the authors could check those and others to determine, whether pili may be encoded by DPANN.

- ATPase involved in archaeal pili biosynthesis, VirB11, COG00630, pf00437

- Pilus assembly protein, ATPase of CpaF family, COG04962, pfam00437, TIGR03819

<https://www.uniprot.org/uniprot/?query=pf00437&sort=score#orgViewBy>

DPANN group (660 results)

- FlaK/PulO Peptidase A24A, prepilin type IV/ Type II secretory pathway, prepilin signal peptidase, COG01989, pfam01478

<https://www.uniprot.org/uniprot/?query=pf01478&sort=score#orgViewBy>

DPANN group (163 results)

- Pilin/Flagellin, FlaG/FlaF family, COG03430, pfam07790

<https://www.uniprot.org/uniprot/?query=taxonomy:1783276%20pf07790>

DPANN group (36 results)

- Pilin

<https://www.uniprot.org/uniprot/?query=taxonomy:1783276%20pilin>

- Pilus assembly protein TadC, COG02064, PF00482

<https://www.uniprot.org/uniprot/?query=PF00482&sort=score#orgViewBy>

DPANN group (661 results)

Figure 4:

I did not find any information on how the plots in D, E and F were generated and how they allow to depict potential host groups. Currently, the inference of potential host-symbiont relationships (lines 261-267 and discussion) seems a bit speculative to me. E.g. Planctomycetes could as well just outcompete the host of CPR or have any other indirect effect rather than being partners, etc. Can you be more specific as to the methods as well as consider additional analyses to support inferences?

Minor comments:

line 22: not all CPR/DPANN are uncultivated

line 23: sentence is not completely clear, consider rephrasing "however, the influence of groundwater chemistry on variation in ..."

line 47: consider additional references:

Krause, 2017: Characterisation of a stable laboratory co-culture of acidophilic nanoorganisms

St John, 2019: A new symbiotic nanoarchaeote (*Candidatus Nanoclepta minutus*) and its host

(*Zestosphaera tikiterensis* gen. nov., sp. nov.) from a New Zealand hot spring

Figure 1Aa-b: the colored stars are hard to see on the colored maps, can this be modified?

Figure 1B: where are the hatched bars representing unbinned rps4? (line 112)

Figure 1C: very hard to see which phylum is represented by which color

Figure 2: would it be possible to use the same color code for the box plots and the phylogenies?

Figure 2B: what is the lineage between Nanohaloarchaeota and Parvarchaeota?

Who are the Mamarchaeota?

line 182: consider rephrasing title as the following paragraph is more about community function that roles of CPR and DPANN

115: does the percentage of DPANN/CPR abundance take into account metagenomes from all filter size steps? Perhaps, this should be explained in more detail in the methods. How are these abundances comparable?

Figure 3B: it would be easier to see presence of genes if plotted onto a wait background

line 215: which genes were considered to approximate carbon fixation capacity?

line 222: please check whether NosD necessarily is involved in nitrous oxide metabolism or represents an accessory protein that could be involved in another functional context

Figure 4A: change green color? (color blindness)

line 448: anoxic instead of anaerobic?

line 451-453: this should be phrased a bit more careful

is the ggKbase pipeline for taxonomic classification published/accessible to allow reproducibility?

how were insertions in ribosomal RNAs removed?

line 575: specify "SI Table xx"

Cryo-TEM: it is hard to see the pili connecting symbionts with hosts, i.e. pili that extent into the host cell. I am wondering whether this would be easier to see, without the white boxes surrounding it (getting too noisy). A simple arrow should be sufficient.

Reviewer #3 (Remarks to the Author):

in their manuscript "Huge and variable diversity of episymbiotic CPR bacteria and DPANN archaea across groundwater ecosystems", He and colleagues describe the genome resolved metagenomic analysis of samples from a timeseries of groundwater samples from a site influenced by agricultural runoff, as well as single time points from 7 sites that are considered pristine. The analysis recovers 2007 genomes, of which 746 are affiliated with the CPR and DPANN clades. the manuscript focuses on the characteristics of the microorganisms represented by those 746 genomes.

The authors frame their analysis in light of the possibility that CPR and DPANN organisms in groundwater could be a source for organisms of these clades detected in the human microbiome, but do not show any comparisons of the organisms detected in their groundwater samples to organisms detected in the human microbiome. Rather, the differences between the various sites suggest that environmental conditions play a major role in shaping the CPR/DPANN community. In light of this, it seems less likely that groundwater is a seedbank for the CPR/DPANN component of the human microbiome. The authors should either include a comparison of the CPR/DPANN community described here with those detected in humans, or alternatively ground this hypothesis by providing additional references on groundwater as a source for human microbiota in general.

The current statement on data availability shows a link to ggKbase, and placeholders for bioproject ID and the supplemental table where the list will be available. Even after signing up for ggKbase, and using the link while logged in, I get an "You do not have permission to do that. Please email ggkbase-ticket@berkeley.edu for help." error. Please make sure the data is available before the manuscript is accepted/published.

The successful binning of the major representatives of the microbial communities in all but one site is quite impressive, as is the fact that the genomes represent a majority of reads on average. This said,

can the authors discuss/ speculate (here and in the manuscript) why the binning of abundant lineages in pr2 is so much less effective than it is for the other sites?

The overlap between sites pr1 & pr7 is interesting, especially since it comprises the majority of genomes (44 of 63) recovered from pr7. The authors provide reasons why the aquifers are unlikely to be connected, and why index hopping is unlikely, but provide no explanation why the community overlap would exist. Since the geochemistry doesn't appear to be strongly similar and the sites are located further apart than some of the other sites are from pr1, it is a mystery why the similarity would be so high. Is there any chance of sample switching?

For figure 3, while I like that general aspects of community metabolic potential are included, figure 3B shows that the contribution of the CPR/DPANN genomes to these overviews is minimal. Thus, the connection to the central point of the manuscript seems limited.

For the metabolic genes present in the CPR, did the authors do any phylogenetic verification of the annotation? NirK and nosZ can be very sensitive to misannotation.

If I understand figure 4B correctly, the lower section only shows centroid genomes of the same analysis represented in the top section. If that interpretation is correct, how is it possible that the centroid for some of the 0.1 micron fraction is outside of the range of the data shown in the top section

The cryoSEM shown in figure 5 is not very clear. While the pili-like structures are clear, it is very difficult to identify the pili entering the host cell in the white boxes (panel H, line 350). It is also hard to see the higher density layer at point of host cell attachment in 5d and 5e and the distinction single membrane versus double membrane. 5i should have an orange arrow according to line 295.

line 329: the authors should be careful in interpreting phylogeny from the cryo-EM data. There is an assumption that the small cells observed here are from the CPR lineage, but no direct evidence of this. I agree it is a likely interpretation, but the language describing the interaction between ultra-small putative CPR bacteria and host cells should reflect this is an assumption.

comparisons of different size fractions in the different panels (>2.5 vs 0.2-0.1 in A, 2.5-0.65 vs 0.2-0.1 in B, and >2.5 vs 0.65-0.2 in SI fig 7) make it seem cherry picked. While I think showing the strongest effects in the main figure is fine, it would be good to show all comparisons in the supplement.

The colors/symbols on SI fig 2 do not fit with the legend.

Author Rebuttal to Initial comments

Reviewer #1 (Remarks to the Author):

1. *This is another interesting paper by Christine He and her co-authors. They performed genome-resolved metagenomics of eight groundwater microbiomes in California which yielded 746 dereplicated CPR and DPANN genomes, which is awesome.*

We thank the reviewer for their appreciation of the metagenomics work.

2. *One major finding of this story is that little species-level genome overlap exists between the*

sites according to distinct physicochemical conditions at each site, but maybe also according to host populations. To investigate this relationship, the authors profiled all recovered genomes against KEGG and custom protein HMM databases which yielded metabolic profiles of the whole groundwater communities at each site. Surprisingly, heatmap of all CPR and DPANN genomes did not show a clear clustering of key genes required for various metabolic and biosynthetic functions.

The reviewer brings up an interesting point. In prior work, our lab has conducted an extensive analysis of protein families within the CPR and found that lineages within the CPR do cluster based on protein family content (<https://doi.org/10.1038/s41467-019-12171-z>). In the current manuscript, the gene set used was quite limited (a curated set of key genes related to biogeochemical function) and so it is perhaps not surprising that the same pattern was not detected. In fact, these observations offer the insight that biogeochemically relevant functions are more likely to be organism specific and do not closely follow the pattern observed when the comprehensive proteome is used. We have added a brief mention of these points to the manuscript.

- 3. Nonetheless, many CPR and DPANN lineages appear to have a complementary or accessory role in different steps of denitrification, nitrite reduction to ammonia. A few genomes encode for sulfur dioxygenase *sdo*, and others have genes involved in sulfate reduction, suggesting a potential role of CPR and DPANN organisms in transformations to sulfite.*

Based on these results, I am not convinced that the high level of differentiation between groundwater communities reflects species adaptation to different physicochemical conditions as the authors suggested (Page 16, line 419). It would be very helpful, if the authors could provide some additional information here.

We agree that this statement is speculative. We have modified the text to state that the most parsimonious explanation is a mix of adaptation to biogeochemical conditions, coupled with bottlenecks and/or founder effects.

4. *One agriculturally-impacted, river sediment-hosted aquifer (site Ag) was sampled 5 times across a 15 months period. Relative abundance over time for non-CPR bacteria and CPR bacteria showed a peak of Planctomycetes which coincided with a low CPR abundance and vice versa suggesting a possible parasitic CPR-host relationship. A potential commensal or mutualistic CPR-host relationship was suggested with two Ignavibacteria and Betaproteobacteria organisms. But temporal resolution was too low to identify potential hosts.*

We have adjusted the wording to indicate that the temporal abundance trends may reflect some sort of relationship (direct or indirect) that merits further investigation.

5. *To me, the coolest part of this paper is the cryo-TEM imaging which shows pili-mediated episympiotic interactions between ultra-small cells and host cells.*

We thank the reviewer for their appreciation of the cryo-TEM images.

6. *At the ultra-small cell/host contact region, the host cell envelope appears to be thickened. One host cell has multiple ultra-small bacterial cells directly attached to its cell envelope which appear to be in the process of dividing, raising the possibility that replication is correlated with host attachment. Other ultra-small cells were observed to be associated with/in close proximity to lysed cells, suggesting that some organisms scavenge resources (especially lipids or lipid building blocks) from dead cells or act parasitically from host organisms. These images also suggest that some ultra-small bacteria have cell envelopes that do not resemble those of Gram-negative bacteria, but seemingly can attach to Gram-negative hosts. Unfortunately, ultra-small cells and host cells can not be identified in TEM images. Thus, there is no final proof that ultra-small cells are indeed CPR. Thus, I would recommend to adjust the wording in the text.*

The reviewer is correct that there is no definitive proof that ultra-small cells are CPR bacteria, although the connection between small genomes and small cell size has been well established in our prior work [Luef et al.] and that of others. We have adjusted the text to indicate that it is likely rather than certain that these ultra-small cells are CPR and DPANN organisms.

7. *The authors conclude that cryo-TEM imaging shows that some CPR bacteria in Ag groundwater are episympionts of prokaryotic hosts (Page 12, line 301). Serial size filtration for fractionating CPR and DPANN cells according to host attachment identified two lineages, Kerfeldbacteria in the CPR and Pacearchaeota within DPANN, with likely strong physical attachment to hosts. However, the title promises much more: "Huge and variable diversity of episympiotic CPR bacteria and DPANN archaea across groundwater"*

ecosystems". I did not see any evidence that diverse CPR and DPANN archaea are episymbionts in Ag groundwater, and no images were provided from other groundwater systems.

We thank the reviewer for this point. We concur that we present a single set of observations so it is technically incorrect to say that the data in the original manuscript applied to a diversity of CPR and DPANN (we are delighted that this reviewer found the data that we did provide of great interest!).

We attempted to address this comment by conducting tangential flow filtration at three additional sites. Despite a heroic effort (especially due to the COVID-19 restrictions), we were unable to acquire additional images. Further analysis demonstrated that it would take > 8 days of on-site filtration to get cell densities sufficient to yield cryo-TEM data. Even if this was feasible in such dilute groundwater systems, the perturbation the cells would experience over this time period would render the experiment of no value.

To provide a different but supporting line of evidence, we turned to the data from the size filtrates. We hypothesized that if CPR and DPANN attach to other organisms via physical interactions (as shown in the cryo-TEM), they should occur in size larger size fractions larger than predicted based on their cell sizes (median CPR cell diameter of ~0.2 μm : <https://www.biorxiv.org/content/10.1101/2020.04.07.029462v1>). In our serial size filtration data across groundwater sites, we see that a significant fraction of the organisms in the 2.5+ and 0.65-2.5 μm size fractions are CPR and DPANN organisms (newly added Fig 6A). Since a large proportion of total filtered biomass is in the 0.65-2.5 μm fraction (in some sites, this fraction contains the majority of total biomass), this indicates that a significant fraction of total CPR and DPANN organisms in the population are likely attached to hosts. Additionally, the CPR and DPANN organisms found in the 2.5+ and 0.65-2.5 μm size fractions are from diverse lineages (SI Table 7). We have added a note to the text that while we have highlighted Kerfeldbacteria and Pacearchaeota as prominent examples, it is likely that CPR and DPANN organisms from diverse lineages are host-attached in groundwater sites. The presence of these small organisms in large size fractions is not likely due to clogging of filters, as visual inspection of filters after filtration showed no visible buildup, and we did not observe any slowdown in the pumping rate that would occur if filters clogged.

8. Thus, the title should be changed to better reflect the much more restricted findings of this story. Some parts should be rewritten.

Given the combination of filtration, cell distribution and iRep results (Fig 6), cryo-TEM imaging of Ag groundwater (Fig 5), and metabolic analyses indicating minimal biosynthetic abilities across all groundwater sites (Fig 4), we feel confident that these organisms are symbionts and some evidence supports them to be episymbionts. Thus, we have retained episymbionts in the title, but agree to change this to symbionts if the editor requests this.

9. *Another recent study on groundwater CPR and DPANN organisms claims that their cell-cell association experiments show lack of physical associations between most Patescibacteria or DPANN with other microorganisms (preprint <https://www.biorxiv.org/content/10.1101/2020.04.07.029462v1>). In addition, general genome features of Patescibacteria and DPANN do not seem to provide convincing evidence of an obligate symbiotic lifestyle. These authors state that the unusual genomic features of these organisms and prevalent auxotrophies may be a result of minimal cellular energy transduction mechanisms that potentially precede the evolution of respiration, thus relying solely on fermentation for energy conservation. As the findings of both studies are somehow contradictory, and cell attachment could only be shown for some ultra-small cells from site Ag, it would be nice to comment on potential alternative explanations.*

We thank the reviewer for mentioning this preprint that offers a conflicting view, thus giving us the chance to address it. These authors used FACS on cells from multiple sites, obtained SAGs, and found little co-occurrence of CPR/DPANN genomic signatures with non-CPR/non-DPANN organisms (based on 16S rRNA genes and SAG contamination levels) within sorted droplets. In our opinion, the study's conclusion is based upon relatively little evidence of uncertain quality, as well as some flawed reasoning.

First, there is no guarantee that their droplet sorting method is robust. The authors themselves acknowledge that disruption of associations could have occurred during sampling, and addressed this by using "gentle mixing." In fact, the authors observed few cell-cell interactions of *any* type across 46 globally distributed sites, whereas multiple previous studies have observed specific CPR-host and DPANN-host associations through similar cell sorting techniques and microscopy. We find it premature of the authors to conclude that CPR and DPANN organisms do not widely form host associations, suggesting that previous studies observing these interactions were flukes. A clear alternative possibility is that the authors' droplet methodology needs improvement.

Another problem arises from the incompleteness of SAGs obtained from cell sorting. Based on the preprint's Supplementary Table 1, the mean completeness of the obtained SAGs is 31%. To us, drawing conclusions based on *absence* of genomic signatures within genomes that are only 31% complete is a fundamentally flawed approach. Additionally, cell sorting and SAGs are low throughput techniques - an average of ~100 SAGs were obtained per site in this preprint study. It is quite possible when sampling only 100 cells from an entire community that CPR and DPANN organisms are missed.

Beyond this, it is possible that some CPR do not intimately attach to host cells and some may only have transient attachments. Although not provided in this paper, we have used cryo-TEM to image Saccharibacteria (TM7) aligned along the surfaces of their Actinobacterial hosts (oral samples from the Forsyth Institute) and detect no pili-like structures or cell surface modifications that would ensure robust attachments..

Finally, the authors suggest that CPR/DPANN are “free-living”, focusing on electron transport, but ignoring biosynthesis. The authors do not provide a plausible hypothesis for how these organisms obtain building blocks or alternatives for missing critical biosynthetic pathways. Of course, this would be impossible for them to do with their highly incomplete, fragmented SAGs.

We hope that these issues we found with the preprint are addressed in the peer review process. We believe this preprint illustrates just how little is known about CPR/DPANN symbiosis, how important (and rarely used) cryo-TEM imaging is for directly observing these interactions, and the importance of proper sampling and sequencing methods to detect CPR/DPANN organisms.

10. In general, integration of other findings would also improve the discussion part which is rather short and a little bit disappointing, as many arguments were already discussed in the results section. I would also love to see more findings from other groups studying the groundwater microbiome. The fraction of self-citations is really very high.

We recognize that many of our citations of pristine or agriculturally impacted groundwater microbiome studies come from our lab (most prior surveys target contaminated groundwater). Prior to the Banfield lab’s genome-resolved metagenomic studies of Crystal Geyser and Rifle aquifers, knowledge about groundwater microbiology was very minimal and incomplete. Even to date, ours are among the few studies that have utilized sequential filtration and collected the post-0.2 µm filtrate (given that CPR/DPANN can pass through 0.2 µm filters). Through use of these methods, we encountered innumerable new groups of organisms that had been almost completely overlooked, many of them CPR and DPANN. Other metagenomics studies were not often genome-resolved or relied on genomes from single cells, which are of comparatively low quality and prone to contamination (see <https://doi.org/10.1038/s41564-017-0098-y>). Since then, there have been some high quality publications/reviews on groundwater microbial ecology from other groups, and we have added citations.

Specific comments:

11. Page 3, line 73, see also line 91: When you used the word “time-series”, I expected a much higher temporal resolution over 15 months than just 5 time points. Samplings were also not done following a regular sampling campaign, as samples were taken in March 2017, September 2017, November 2017, February 2018, June 2018. What you show is a temporal patterns over 15 months, which is great as you used a genome-centric approach. But please avoid the term time-series throughout the manuscript when analyzing only 5 time points.

We have replaced “time series sampling” with “temporal sampling” in the text.

12. Any idea why the first sample shows the lowest abundance of CPR?

One potential reason is that the 03/17 Ag time point was mistakenly sequenced at a much greater depth (~10x deeper) than the subsequent time points. Given that CPR organisms are so abundant, we considered that deeper sequencing may have resulted in significantly greater

detection of non-CPR organisms at the tail end of the rank abundance curve. However, subsampling and assembly of 10% of the bulk filtration reads from 03/17 did not result in a higher observed CPR and DPANN organism abundance than in the full assembly.

An alternative possible contributor to the lower observed CPR abundance on 03/17 is the record rainfall experienced in northern California during the 2016-17 rainy season (<https://www.climate.gov/news-features/featured-images/very-wet-2017-water-year-ends-california#:~:text=Across%20the%20Northern%20Sierra%20Mountains,record%20set%20in%201982%2D83.>). In fact, this was one of the wettest seasons in the history of the region. We mention this in the manuscript.

13. Page 7, Fig. 2: 10. How did you calculate relative abundance? Just % of total reads mapping to a specific MAG?

Relative abundance for each OTU represented by an rpS3 marker gene was calculated as ratio of the coverage of the scaffold containing the rpS3 marker gene to the total coverage values of all rpS3 gene-containing scaffolds. This has been clarified in the methods.

14. Page 8, Fig. 3: Due to the lack of electron transport systems, fermentation in remains the main process for energy conservation, which is consistent with your results on metabolic profile Heatmap in Fig3B. But separating metabolic profile results for MAGs recovered from Pr and Ag sites may give more detailed insights. Can you add some more information here?

We found that Pr sites clustered separately from Ag in PCA analysis of broad metabolic capacities (SI Fig 5). We have added Supplementary Tables 8 through 15 with detailed HMM hits and KEGG module completion estimates for each site separately, allowing a reader to focus on specific genes or pathways of interest.

15. Page 11, line 271: You try to link the higher increase in metabolic capacities during the 2016-17 compared to the 2017-18 rainy season with a major difference in rainfall. However, groundwater table fluctuations always show a time lag to precipitation events. Please give more details on the dynamic changes of groundwater level in the groundwater well over the investigation period. Otherwise, this comment is highly speculative.

Importantly, in both years the peak of the rainy season occurred in January (based on precipitation data from NOAA), while we sampled in March of 2017 and February of 2018. The overall rainfall in the five months prior to our sampling was four times higher in 2017 compared to 2018. The Ag site is a shallow groundwater aquifer (sampled at 6 m) so we do not expect groundwater response times to exceed several months and consider it unlikely that lags complicated our interpretation.

16. Page 11, line 288: *How representative are the figures presented? How many TEM samples were analyzed and how many ultra-small cells did not show attachment?*

The figures presented are essentially all the focused images we obtained of host-attached ultra-small cells. See our response below for why the constraints of our methods make it exceedingly difficult to obtain statistics from cryo-TEM images of TFF-filtered groundwater communities. We did not see any ultra-small cells that were not in close physical proximity to a larger host organism.

17. *According to Methods (Page 20, line 529) you analyzed one groundwater sample from site Ag obtained at February 2018 which has a high relative abundance of CPR and DPANN. As you claim in the title that diverse CPR and DPANN are episyntons, you should provide some statistical analyses. Please provide more details.*

To obtain cryo-TEM images is exceedingly difficult, costly and time consuming. The on-site biomass concentration step is challenging, and then ~1 μL of solution has to be appropriately loaded onto a grid, cells must adhere in sufficient quantity, and the grids must be cryo-plunged. Sometimes the ice is too thick, other times too thin, often with large variance even when the same person does grid preparation with the same techniques. Many TEM sessions we tried failed completely for these reasons.

Notably, the larger size of host organisms results in a halo of thick ice that often obscures the edges where CPR and DPANN organisms would be attached, preventing a focused image. As such, we may have been unable to image many of the attached ultra-small cells that were present in the sample. The fact that we were able to obtain focused images of several episyntotic interactions between ultra-small cells and hosts is an indication that their actual occurrence is likely more common than we observed through cryo-TEM.

To find even a few cells can take an entire TEM session. Finding one such association is important and informative. It is notable that only one prior study (from our lab) acquired images of CPR from an environmental sample, and in that study, no images provided any insights into host attachment. Thus, even the one dataset that we provide is quite novel. In the absence of completely new methods, acquiring a large dataset that could be analyzed statistically would take decades.

18. *Page 14, line 357: I do not think that the title of the figure legend is correct. The authors assume that some CPR are attached to their hosts. CPR lineage Kerfeldbacteria and DPANN lineage Pacearchaeota are enriched on the 2.5+ μm size fraction relative to the 0.1-0.2 μm size fraction indicating that a high fraction of these populations is host attached. Please rephrase.*

We have amended the figure legend and introduced new data supporting attachment of CPR/DPANN organisms to hosts across groundwater sites (Fig 6A).

19. Page 15, line 388: Tracking iRep values of CPR genomes across size fractions. Why was the time point 11/17 omitted in Fig. 6B, but shown in 6C?

For time point 11/17, we performed bulk filtering (entire community on a 0.1 μm filter) but not size fractionation. Fig 6D (formerly 6C) shows iRep values obtained from mapping bulk filtration reads, while Fig 6C (formerly Fig 6B) shows iRep values obtained from mapping size fraction reads. We thank the reviewer for pointing out that this difference lacks an explanation, and have added explicit clarification to the text and caption for Fig 6.

20. Fig. 6 C. The differences in iRep are small. You just analyzed 5 time points. Is it not a little bit too exaggerating to infer that CPR cell replication, stimulated by host attachment, is more prevalent during the rainy season compared to the dry season? Please also rephrase this part in the discussion (Page 17, line 446) (See also comment above).

We disagree that the observed differences in iRep values are “small”. First and foremost, the statistical testing we performed (independent t test) indicates statistically significant differences between iRep values of CPR genomes ($n = 500+$ genomes; quite a large sample size) across certain time point pairs (03/17 versus 06/18 and 09/17 versus 06/18). While the differences may look small to the eye in the Fig 6D boxplot (it is impossible with 500+ genomes to make a plot showing iRep differences at the individual genome level, as in Fig 6C), we are confident stating that iRep differences between these specific sets of time points are statistically significant.

Second, even ignoring the results of our statistical tests, we argue that, based on how the iRep metric is defined, an iRep difference of 0.1-0.2 (as observed in Fig 6D) is large enough to be quite meaningful in terms of population behavior. A genome’s iRep value measures the average replication state of the cell population represented by that genome. An iRep value of 1.0 would be produced by a population where no cells are actively replicating. An iRep value of 2.0 would be produced by a population where all cells are making a single copy of their genome (Brown et al. 2016). Therefore, a difference of 0.1-0.2 in iRep values indicates a change in replication state of 10-20% of the population.

We understand that the meaning of iRep values are not immediately intuitive and have added two markers at iRep=1.0 and iRep=2.0 in Fig 6C and 6D to provide benchmarks for interpreting iRep values. We have also added text as well as a note to the caption of Fig 6 to aid the reader in interpreting iRep values.

21. Page 16, line 401. The first sentences are more a summary and are not needed. Please shorten this paragraph.

We agree that the discussion is improved by shortening this summary paragraph, and have done so.

22. Page 17, line 443: Please provide some information about the heavy input of carbon and nitrogen from agricultural waste into Ag groundwater. How does this input vary over time?

We have added a sentence at this point that describes several avenues for how agricultural waste (cow manure) may seep into groundwater: manure lagoons, dried manure piles, and the adjacent corn field which is fertilized by the manure. Unfortunately the dairy farm does not measure how much manure is produced, runs off into the fields, or is used to fertilize the fields over time.

23. Page 19, line 505: It would be nice to provide summary of internal ggKbase database used for taxonomic classifications. It seems that it is not accessible.

We have added additional text in the Methods clarifying how proteins were annotated and scaffold taxonomic profiles determined. The internal ggKbase database is comprised of publicly available genomes (NCBI), including many genomes for candidate phyla that originated in our lab. Given that these genomes are publicly available, anyone wanting to do similar analyzes can construct their own databases, and would presumably give analogous results. Additionally, at this time, our database is several years old and needs to be updated, so it would not actually be a service to provide it.

It is important to note that final genome taxonomy was determined not by Usearch to public databases or the internal ggKbase database, but by phylogenetic analysis of concatenated ribosomal protein sequences.

24. Page 19, line 493: The authors don't give the quality scores they used to trim the reads with. Not sure if sickle has some defaults, but that info is usually included.

This information has been updated in the text.

25. In addition, they should explain more about the scaffolding with IDBA & MEGAHIT. Usually you use one or the other. They should also clarify that whether they assembled each sample independently? Were there replicates, how were these handled? Mapping scaffolds back to reads by bowtie2 does not by default gives scaffold coverage. How these values were calculated?

We found that MEGAHIT achieved better assemblies than IDBA-UD of the large (>100 Gbp per sample) datasets taken from Ag groundwater on 03/17. However, MEGAHIT is only a de novo assembler and does not perform scaffolding of assembled contigs, as IDBA-UD does.

Therefore, we assembled contigs with MEGAHIT and then used the scaffolding function of IDBA-UD. We have added a note about this choice into the Methods. We also added clarification in the Methods that reads from each filter were independently assembled, as well as the explicit formula used for single-fold coverage calculations of both scaffolds and genomes.

26. Page 19, line 498: *Functional annotations of predicted ORFs was done by aligning with different databases eg. KEGG, Uniref100, Uniprot etc. but what was the criteria or cutoff for predicting good hit versus bad alignment hit?*

We have added details about choosing cutoffs for HMMs as well as criteria for determining the presence of KEGG reactions, KEGG modules, and broad metabolic capacities to the Methods section. Additionally, we have added 8 SI tables that detail the METABOLIC results (at the gene, reaction, module, and metabolic capacity level) for each groundwater site sampled in this study.

27. Page 20, line 549: *"The coverage of a genome is proportional to cell counts, so we assumed that a genome's relative coverage is the fraction of total cells in the community represented by that genome. Therefore, the cell count of a genome was calculated as the product of the genome relative coverage and estimated total cells in the community." Do you have any citation to support this method?*

To analyze the distribution of cell numbers across size fractions, we need to estimate absolute/true cell counts, while sequencing data can only generate relative abundance values. The method we use to estimate cell counts takes the general form of: (True Sample) = (Relative abundance from sequencing) * (Microbial load), an approach discussed and tested in depth here: <https://doi.org/10.1038/s41467-019-10656-5>. As outlined in the manuscript, our method takes the form: $c = x * l * m$, where c = "True Sample", i.e. total cell count of a genome; x = relative coverage of a genome; l = cell counts per nanogram of DNA in the community; and m = nanograms of DNA extracted from the size fraction. The term $l * m$ estimates "Microbial load", i.e. total cell count of a community.

In our method, we utilize DNA yield (measured variable m in our equation) as an estimate of microbial load in a sample. DNA yield is an imperfect estimate of true microbial load for a number of reasons, including bias depending on the DNA extraction method (<https://doi.org/10.1128/mSystems.00095-16>). However, there are also limitations and problems with other estimates of microbial load, such as flow cytometry-based cell counting (<https://doi.org/10.1128/MMBR.00009-08>). Given that we extracted all samples in this study using the same DNA extraction kit, we have chosen to employ DNA yield as the best available measurement of microbial load.

Fluorometry-based quantification of DNA yield measures DNA weight (i.e. the number of dsDNA base pairs). Meanwhile, the relative abundance of a genome (relative coverage) is proportional to the relative fraction of total cells represented by the genome, rather than the relative fraction of total DNA represented by the genome. For example, a CPR genome with a relative abundance of 1% will constitute less than 1% of the total DNA yield from a groundwater community, because of its significantly smaller genome size than other members of the community. To account for genome size-dependent DNA yield, we calculated how many microbial cells would correspond to 1 ng of DNA based upon the genome sizes of each member of the community (parameter l in our equation).

We acknowledge that our method only estimates true cell counts for a genome, and depends on several assumptions that may or may not hold true for our samples. However, we believe that we have made rational and justifiable choices to utilize the measurements available to us in order to make meaningful conclusions. We have rewritten and added additional details to this section of the Methods, in order to make our choices and assumptions as clear as possible.

28. Fig. 1A: The image from Google maps is of really low quality and hard to read. Show another map or move it to Supplemental material with a higher quality.

We have replaced the original Google maps image with one of higher resolution.

29. Fig. 3: Please convert units from mg/L to μM . In addition, yellow on white is hard to read. Please improve the color contrast.

SI Fig. 4: Please convert units from mg/L to μM . In addition, yellow on white is hard to read. Please improve the color contrast.

We don't see why unit conversion is necessary, as mg/L is a standard way of reporting such chemical measurements. We have darkened the yellow text in Fig 3 and SI Fig 4 to improve contrast.

Reviewer #2 (Remarks to the Author):

1. *In the manuscript by He et al., entitled “Huge and variable diversity of episymbiotic CPR bacteria and DPANN archaea across groundwater ecosystems”, the authors reconstruct 746 CPR and DPANN genomes from one agricultural and several pristine groundwater samples and infer the metabolic potential of the symbionts as well as metabolic processes on community level. Furthermore, the authors compare community structure and chemistry variation across sites, try to identify potential host groups and use Cryo- TEM to study the potential of host-symbiont interactions. Finally, they also assess the level of host attachment across various DPANN and CPR lineages by estimating cell abundances based on genomic abundance data.*

The study is very comprehensive, represents an impressive amount of work and is well written. While the authors have previously reported on large amounts of DPANN and CPR bins from other sites, the presence and variation of these lineages in groundwater ecosystems, which in part provide drinking water, have so far not been analysed in detail. Finding a diversity of CPR and DPANN in these systems could potentially be interesting with regard to human disease because certain members of these groups seem to be part of the human microbiome.

We thank the reviewer for their careful reading and positive comments.

However, I have a number of comments regarding methodology and their effect on results/conclusions, which the authors should take into consideration.

Major comments:

Current flow of the manuscript

2. *My impression is that the extensive analyses presented in this study, while very interesting, are sometimes a bit disconnected. Would it be possible to more clearly phrase the major questions/aims and use these as a thread throughout the manuscript?*

We agree that the paper has rather broad scope, but we feel that the series of investigations address one large question: What is the role of CPR and DPANN organisms in groundwater communities? The subtopics are perhaps best summed up by this sentence in the Introduction: “Despite their detection in groundwater, the variation in abundance and distribution of CPR and DPANN organisms across groundwater environments, their roles within groundwater ecosystems, and their relationships with host organisms are not well characterized.”

To tie the narrative back to these questions throughout the text, we have added/amended the

following text:

Introduction, final paragraph: “Here, we use genome-resolved metagenomics to analyze eight groundwater communities in Northern California, with the goal of understanding what role CPR and DPANN organisms play within groundwater ecosystems.”

Pg 4: “First, we sought to characterize the composition of each groundwater community and compare community compositions, with a particular focus on the prevalence and diversity of CPR and DPANN organisms.”

Pg 8: “Next, given the prevalence of CPR and DPANN organisms in these eight groundwater communities, we sought to investigate the potential metabolic roles these organisms play and how these roles may vary between communities.”

Pg 10: “After establishing the prevalence and metabolic roles of CPR and DPANN organisms in groundwater communities, we performed temporal and size filtration sampling of Ag groundwater (Fig 4A) in order to investigate how these characteristics change with time and environmental factors.”

Pg 11: “Fundamental to understanding the wider role of CPR and DPANN organisms in groundwater communities is understanding their specific relationships with other organisms (hosts). Little is known about these relationships and only a handful of studies have performed high resolution microscopy to directly image physical associations between CPR/DPANN organisms and hosts in natural environments^{14,43,44}.”

Pg 14: “Imaging of direct attachment of CPR/DPANN organisms to host cells led us to ask how widespread physical attachment is across the diversity of both radiations. To answer this question, we analyzed how CPR/DPANN organisms were distributed among size fractions, which should reflect two factors: cell size and attachment to larger host cells.”

3. *Further, the method section in the main text is very short and I would highly recommend that the method section of the Supplementary material is being moved to the main document.*

In response to this reviewer and other reviewers, we have substantially expanded the Methods and Supplementary Methods sections.

Metabolic inferences:

4. *In general, it is currently unclear, which exact enzymes the authors looked at to infer presence/absence and abundances. To make this analysis more reproducible and transparent, I would recommend that the authors provide supplementary tables, reporting the presence/absence pattern of all investigated marker proteins (for each functional category) and their COGs/KOs/PFAMs/Interpro-domains (e.g. those used to draw the metabolic profiles, e.g. in Figure 3A and B) across the metagenomes and MAGs analysed. Further, it would be useful to list the read coverage of these marker*

proteins that were used to draw the figures (Fig. 3 (A and B) and SI Fig. 4). Or perhaps, I missed these files?

We agree that these details are critical and were not provided in the original manuscript. We have added details about which genes we searched for, HMM IDs and cutoff choices, as well as criteria for determining the presence of KEGG reactions, KEGG modules, and broad metabolic capacities to the Methods section. Additionally, we have added 8 SI tables that detail the METABOLIC results (at the gene, reaction, module, and metabolic capacity level) for all genomes from each groundwater site.

The specific marker genes that were used to infer the presence/absence of biogeochemical functions are listed in the “FunctionHit” tab of SI Tables 8-15. Coverage values used in Fig 3A and SI Fig 4 correspond to average coverage values across the entirety of a genome containing the corresponding key marker genes.

5. *Did you do any manual inspections to assess presence/absence of metabolic modules and verify inferences? The rule of counting a KEGG module as present whenever 75% of its genes were present could result in “false positives” leading to incorrect inferences. Furthermore, for certain pathways, such as the 3-Hydroxypropionate and related carbon fixation pathways this may be difficult, because enzymes belong to broad protein families which may rather participate in other pathways. In turn, I am wondering whether you took into account or give higher credit to the presence of key enzymes specific to a certain pathway? For instance, even if more than 75% of the enzymes of the Wood- Ljungdahl pathway are present, this pathway may be absent if the key enzyme Acetyl- CoA synthase/ Carbon monoxide dehydrogenase is lacking and may have functions other than carbon fixation (see for instance Zhuang et al. 2014, PNAS, Incomplete Wood–Ljungdahl pathway facilitates one-carbon metabolism in organohalide-respiring *Dehalococcoides mccartyi*).*

First, we note that the 75% threshold is for the reactions, not genes, that comprise a module. In turn, a reaction is considered present only when 100% of the genes necessary for that reaction are present. This is a subtle but important distinction, as the 75% reaction threshold used is more stringent than a 75% gene threshold. We have added more detail to the Methods section explaining these criteria.

The reviewer is correct that a 75% threshold for reactions in a given module could still result in false positives (inference that a pathway is present when a critical gene is missing). We also note that there was no explicit weighting of enzymes or reactions that are critical or unique to pathways, which may also result in incorrect pathway determinations. However, manual inspection of select pathways shows that the 75% reaction threshold rarely considers a pathway present when such a critical enzyme is missing. For example, the Wood-Ljungdahl pathway (M00377) consists of 6 reactions, so according to the 75% reaction threshold, 5 out of the 6 reactions must be present for the pathway to be considered present. Manual inspection

shows that there are no genomes for which the only missing reaction is that corresponding to Acetyl-CoA synthase/carbon monoxide dehydrogenase (K14138+K00197+K00194).

Another pathway we manually inspected is the Calvin-Benson-Bassham cycle (M00377), which is considered present if 9 out of 11 total reactions are present. In *Ag*, out of 33 genomes passing this 75% reaction threshold, 10 genomes (GD18-4_B1_Deltaproteobacteria_Maxbin2_440, GD2017-2_S2_QB3_180125_Deltaproteobacteria_58_12, GD2017-2_S65_QB3_180125_Deltaproteobacteria_58_11, GD2017-3_B1_QB3_180125_Deltaproteobacteria_57_12, GD2018-4_B1_QB3_180703_Alphaproteobacteria_59_8, GD2018-4_B1_QB3_180703_Alphaproteobacteria_70_17, GD2018-4_B1_QB3_180703_Armatimonadetes_31_7, GD2018-4_B1_QB3_180703_Deltaproteobacteria_58_8_b, GD2018-4_B1_QB3_180703_NC10_67_16, and S_p1_S3_coassembly_Alphaproteobacteria_56_24) encode for all reactions except for the two reactions corresponding to PRK and RuBisCO. Although we could choose to be conservative and decide that the CBB pathway is absent in these 10 organisms, they all together make up less than 0.1% of the population, and their presence/absence does little to change our conclusions about metabolism about the community level.

We note that we chose a 75% threshold to balance between false positives and false negatives (a gene is actually encoded for by an organism, but did not assemble or get binned). Only with fully complete, circularized genomes can one be confident that an undetected gene is truly missing from an organism's genome, rather than simply not observed in an incomplete genome assembly. Completion and curation of a MAG is possible only when the genome is composed of a few scaffolds and is incredibly time-consuming, so not an option for 2,007 MAGs. We recognize that any metabolic pathway prediction from draft MAGs is going to result in mistakes, but believe that our threshold of 75% is a reasonable balance that minimizes both false negative and false positive detection.

Finally, we note that the numbers presented in Fig 3A are assessments of the genomic *potential* for these metabolic pathways, but do not have transcriptomic information to say which pathways are expressed and at what level. In the big picture, the difference between genomically encoded pathway potential and the level at which pathways are actually expressed is likely more significant than differences due to inclusion or exclusion of false positive genomes.

6. *Finally, I had one question regarding ammonia oxidation: Figure 3 and SI Figure 4 suggest that you could not detect any genes coding for proteins involved in ammonia oxidation. I found this a bit surprising/interesting because according to SI Table 2 (SI Table 2: Completeness and contamination estimates for all 2,007 dereplicated genomes in this study), your samples contain several MAGs assigned to putative ammonia oxidizing archaea such as Nitrosoarchaea/Thaumarchaeota. Can you verify, that these MAGs are indeed Thaumarchaeota and if so, check whether they encode amogenes?*

In our original results, we did not detect any genomes which contained *amoABC*. However, it is difficult to distinguish hits for *amoABC* with those for *pmoABC* (thrown together in one protein cluster in many databases; K10944+K10945+K10946 in KEGG). Since we submitted the manuscript, a new version of the METABOLIC software has been released that includes a manually curated motif validation step to distinguish between these two proteins with high sequence similarity but divergent function. After inclusion of this new motif validation step (SI Tables 8-15 include results from the newest version of METABOLIC), we find that two genomes contain positive hits for *amoABC*: Montesol18_Sp65_coassembly_Nitrospirae_56_10 in *Pr1*, which has a relative coverage of $\sim 10^{-6}$; and SR2-18-B1_coassembly_Nitrospirae_56_60, which has a maximum relative coverage of 2.6% at the 2018 time point. We have updated the figures to reflect these new findings, but even these updated results do not indicate much genomically encoded potential for aerobic ammonia oxidation. This is perhaps not surprising given that the groundwater sites we sampled were anoxic.

Cell distribution across filters

7. *I had two concerns regarding the calculation of total cell counts:*

First of all, I am not sure about the validity of the assumption that genome coverage is proportional to cell counts because microorganisms are known to have various levels of ploidy that makes it hard to compare cell numbers based on nucleic acid content, especially in complex microbial communities, see for instance Soppa, 2014: "Polyploidy in Archaea and Bacteria: About Desiccation Resistance, Giant Cell Size, Long-Term Survival, Enforcement by a Eukaryotic Host and Additional Aspects".

Furthermore, regarding the calculation of cells counts based on the product of genome relative coverage and estimated total cells in the community: I was wondering to what extent cells that did not lyse bias this analysis.

Did you try to estimate the amount of lysed versus non-lysed cells across samples?

We concur that polyploidy, if it occurs in these organisms, complicates our analyses but we have no data to address this and, to our knowledge, no large-scale analysis of polyploidy across bacteria and archaea exists (the Soppa, 2014 reference discusses only 8 species). This is a complication for nearly any microbial ecology study attempting to quantify organism abundances, with no solution in sight.

We did not estimate lysed versus non-lysed cells. No DNA extraction method is 100% efficient for 100% of a community's members, but our DNA extraction utilized physical (vortexing), chemical (surfactant), and thermal (heating) steps to treat cells, so we expect to have lysed as many cells as possible. Additionally, the same DNA extraction method was used for all samples in this study.

We have added to the manuscript a comment that we recognize that our biomass estimate is flawed, due to potential polyploidy and DNA extraction bias. However, we maintain that using

extracted DNA to estimate biomass provides substantially more insight than no attempt to estimate biomass.

Phylogenetic analyses

8. *Protein trees: Are fasta files (including references sequences), alignments files and single gene trees for each of these ribosomal proteins available? i.e. did you perform single protein trees to confirm that the identified proteins indeed represent ribosomal proteins and not distant homologs?*

We have added tree and alignment files as Supplementary Data 1-4. We did not construct trees for individual ribosomal proteins, besides rpS3. However, the HMMs used to identify the ribosomal proteins are well tested, and for the concatenated ribosomal protein sequences, we required that the ribosomal proteins are co-located in the genome. Thus, the probability that any non-ribosomal proteins were used is very small. Additionally, each gene was aligned separately prior to concatenation (see response below).

9. *How were the sequences of the new MAGs aligned with the reference sequences? Was any trimming performed? To avoid misalignments, it is recommended to first align all single protein homologs of a certain ribosomal protein (from both MAGs and reference species), eventually trim the single protein alignments (and perform single protein trees to identify potential distant paralogs, which, if present, should be removed before realigning/trimming) and then concatenate all single protein alignments for phylogenetic analyses.*

We created ribosomal protein trees as the reviewer describes: first by concatenating reference sequences and HMM hits, then aligning to the Pfam HMM model, and removing insertions added by hmalign (see “Phylogenetic classification” in the Methods section, which was previously in the Supplementary Methods section).

10. *Considering the large diversity of organisms in these phylogenetic analyses, it may also be useful to test models and use a model that best fits the data. E.g. it is highly likely that mixture models (implemented in IQ-tree) would yield more likely phylogenetic trees which would help to evaluate the support for monophyly of the new phyla suggested in this study. To this end, I would also recommend to estimate branch support using single branch test as fast bootstraps can overestimate support.*

We thank the reviewer for these important comments. While we agree that mixture models could yield more likely phylogenetic trees, we believe our current LG + gamma model of evolution provides sufficient support for the monophyly of the two new phyla suggested, and is consistent with the use of LG plus gamma models of evolution used in previous studies that have taxonomically profiled the CPR and DPANN radiations (Brown et al, Nature, 2015; Hug et al, Nature Microbiology, 2016; Jaffe et al, BMC Biology, 2020). We did attempt to run the PMSF

approximation in IQTree, but this job has been running for over 3 weeks on our cluster (48 CPUs) with no end in sight - we can only guess it will take an incredibly long time to run.

11. It would also be helpful, if the authors could provide additional phylogenetic trees that include MAGs of the other communities members outside the DPANN/CPR, as these are important with regard to potential hosts as well as for the metabolic inferences of total communities.

We have added the full, unpruned tree containing the CPR from Fig 2B as Supplementary Data 3-4.

12. 16S rRNA: I do not understand why 16S rRNA trees were inferred with a protein model “LG protein substitution matrix”. Can you check whether this is correct and if so, redo the analyses using a nucleic acid model?

We thank the reviewer for catching this mistake. We did utilize the General Time Reversible model of nucleotide substitution to construct our 16S rRNA tree (GTRCAT model option in RAxML; the full RAxML command is now listed in the Methods).

Proposal of new phyla

13. I was wondering whether the criteria you used for determining the status of a new phylum are sufficient: e.g. It would be great if you mention the ANI thresholds used to support the status of a new phylum in the methods section and mention how this compares to suggested thresholds?

We apologize that we don't understand this question, as we cannot obtain ANI thresholds for genomes that are so divergent at the nucleotide level (i.e. ANI thresholds are only defined for species to strain level; (Olm et al. 2020)). As we note in the “Phylogenetic classification” section of our Methods, our classification of novel phyla is based upon the same set of criteria originally used to delineate phylum-level lineages within the CPR: genomes are monophyletic in 16S rRNA gene and concatenated ribosomal protein phylogenies, 16S rRNA gene sequences share $\leq 24\%$ identity with closest representatives, and more than one genome is present in the novel lineage. These criteria are generally consistent with those that have been used by others to define novel lineages.

14. Furthermore, please evaluate the phylum-level status of these lineages based on results of the suggested phylogenetic inferences (see above). In particular, the protein phylogeny should be taken into account because concatenated protein trees are usually more reliable to accurately place symbionts in phylogenetic trees (perhaps, this phylogeny would be more robust, if more

proteins were concatenated).

As we note in the above response and in our Methods section, we considered both 16S rRNA gene and concatenated ribosomal protein phylogenies in determining a novel phylum-level lineage.

15. *Further, I am wondering what you mean by point 4 (line 910): one representative draft genome? Was this a draft genome of high quality? I would recommend that you mention the genome completeness/quality in Table S3 and also report ANI and support values from the two phylogenetic analyses.*

By “representative”, we meant a genome from the dereplicated set of 2,007 genomes (dereplicated at 99% ANI). As we mention in our Results and Methods, all genomes used for analysis in this study were at least 70% complete and had less than 10% contamination (as measured by copies of single copy genes). The quality of all genomes is shown in Table S2 (completeness, contamination, and copies of all single copy genes). As we note in above responses, there are no accepted ANI thresholds for genomes as divergent as at the phylum level. Support values are available in the supplementary tree files.

Pili

16. *I as surprised reading that the authors infer that all cells with pili-like structures belong to CPR but not to DPANN. In Comolli, 2009, the authors concluded that appendages on ARMAN cells likely represent pili. Furthermore, Hamm identified putative T4 pili proteins in Nanohaloarchaeota (Hamm et al., 2018) and St John, 2019 identified flagella-like structures in Nanoclepta.*

The reviewer is correct that the Comolli, 2009 study does infer that the 9-10 nm thick appendages are likely pili. In terms of pilin structural proteins (K02657 - K02661, K06596, K02667 - K02670), we found a single protein in a single DPANN genome among all *Ag* genomes. The *Ag* CPR genomes contain more hits to pili genes but none contain a complete set - at most, some contain both *pilT* (K02669) and *pilU* (K02670). Given the incomplete genomic encoding for pili in CPR and DPANN genomes, we agree that our original statement (“The presence of pili identifies these ultra-small cells as CPR bacteria, since components of type IV pili systems are present in the majority of CPR bacterial genomes and missing in DPANN genomes”) is too strong. We have amended the manuscript to remove the identification of the ultra-small cells as CPR bacteria based upon pili-like appendages.

17. *In turn, I was wondering, whether you may have missed archaeal pili proteins? Which protein families did you look for? Please note that archaeal T4 pili encoding proteins are hard to identify as they differ from bacterial ones. A nice review on this can be found in Makarova et al., 2017, Diversity and Evolution of Type IV Pili Systems in Archaea. Some candidates for archaeal pilin proteins such as VirB11 seem to be present in DPANN based on pfam domains and perhaps the authors could check those and others to determine, whether pili may be encoded by DPANN.*

The results of searching for the suggested genes are listed below. As we note in our previous response to point 17, some pili-related genes can be found in both CPR and DPANN genomes, but no genome contains close to a complete set of necessary genes for assembly and operation of pili, making it difficult to assess which organisms actually have pili from genomic information alone.

- *ATPase involved in archaeellum/pili biosynthesis, VirB11, COG00630, pf00437*

There are three DPANN genomes in *Ag* containing this gene based (KEGG K03196):

GD2017-1_S_p2_S4_Biohub_170907_DPANN_55_27

S_p2_S4_coassembly_Micrarchaeota_46_17

S_p2_S4_coassembly_DPANN_51_18

- *Pilus assembly protein, ATPase of CpaF family, COG04962, pfam00437, TIGR03819*

<https://www.uniprot.org/uniprot/?query=pf00437&sort=score#orgViewBy>

DPANN group (660 results)

We found 167 hits to *cpaF* based on KEGG K02283. Among these hits, there is only a single DPANN genome (no other archaea).

- *FlaK/PulO Peptidase A24A, prepilin type IV/ Type II secretory pathway, prepilin signal peptidase, COG01989, pfam01478*

<https://www.uniprot.org/uniprot/?query=pf01478&sort=score#orgViewBy>

DPANN group (163 results)

- *Pilin/Flagellin, FlaG/FlaF family, COG03430, pfam07790*

<https://www.uniprot.org/uniprot/?query=taxonomy:1783276%20pf07790>

DPANN group (36 results)

- *Pilin/Flagellin, FlaG/FlaF family, COG03430, pfam07790*

<https://www.uniprot.org/uniprot/?query=taxonomy:1783276%20pf07790>

DPANN group (36 results)

Three *fla* genes are missing entirely from *Ag* genomes (*flaA*, *flaE*, and *flaG*). We found 106 DPANN genomes with hits to *flaK* (KEGG K07991), 12 DPANN genomes with hits to *flaF*. There are 20 DPANN genomes with 4 of the *fla* genes.

- *Pilin*

<https://www.uniprot.org/uniprot/?query=taxonomy:1783276%20pilin>

- *Pilus assembly protein TadC, COG02064, PF00482*

<https://www.uniprot.org/uniprot/?query=PF00482&sort=score#orgViewBy>

DPANN group (661 results)

This gene (KEGG K12511) was found in 5 DPANN genomes and 3 CPR genomes.

18. Figure 4:

I did not find any information on how the plots in D, E and F were generated and how they allow to depict potential host groups. Currently, the inference of potential host- symbiont relationships (lines 261-267 and discussion) seems a bit speculative to me.

E.g. Planctomycetes could as well just outcompete the host of CPR or have any other indirect effect rather than being partners, etc. Can you be more specific as to the methods as well as consider additional analyses to support inferences?

Fig 4D and 4E depict relative abundance values (percent of total coverage) for individual genomes, whose calculation is now described in the Methods section: “Single-fold coverage values for genomes were calculated as the ratio of total length of mapped reads (bowtie2 version 2.3.5.1) to the total length of the genome.”

Fig 4F depicts total relative abundance values of genomes capable of a specific metabolic function (calculation described in the “Genome and community-level metabolic predictions” section of the Methods and the same as utilized in Fig 3).

We agree that co-occurrence patterns are certainly not sufficient evidence for a symbiotic relationship. We have modified our language to emphasize that a symbiotic relationship based on these observed patterns is speculative, merits more investigation, and is only one of many potential reasons for these abundance patterns, as the reviewer correctly points out.

Minor comments:

19. line 22: not all CPR/DPANN are uncultivated

We have changed “uncultivated” to “unisolated”.

20. line 23: sentence is not completely clear, consider rephrasing “however, the influence of groundwater chemistry on variation in ...”

We have restructured the sentence as suggested.

21. line 47: consider additional references:

Krause, 2017: Characterisation of a stable laboratory co-culture of acidophilic nanoorganisms Sr John, 2019: A new symbiotic nanoarchaeote (Candidatus Nanoclepta minutus) and its host (Zestosphaera tikiterensis gen. nov., sp. nov.) from a New Zealand hot spring

We thank the reviewer for the suggestion and have added this reference.

22. Figure 1Aa-b: the colored stars are hard to see on the colored maps, can this be modified?

We have replaced the colored stars with black shapes.

23. *Figure 1B: where are the hatched bars representing unbinned rpS4? (line 112)*

The hatched bars are intended to indicate individual rpS3 genes that were unbinned (Fig 1C), whereas Fig 1B shows the total relative coverage of all rpS3 genes which comprise each phylum. This information is also provided in the figure caption.

24. Figure 1C: very hard to see which phylum is represented by which color

This is a difficult problem to solve as there are 65 phyla/groups represented in Fig 1C. For further clarification, we have added SI Table 3 which lists the relative coverage and taxonomic classification of all rpS3 genes for all sites.

25. Figure 2: would it be possible to use the same color code for the box plots and the phylogenies?

Because there are 60 different colors/phyla in the box plot, we believe that coloring the tree by phylum would be more visually distracting than helpful. Alternatively, coloring the boxplot according to the tree would result in dozens of phyla sharing a single color. We regret that the user cannot match colors between the boxplot and tree, but believe that this is the best presentation of the data.

26. Figure 2B: what is the lineage between Nanohaloarchaeota and Parvarchaeota? Who are the Mamarchaeota?

The lineage between Nanohaloarchaeota and Parvarchaeota consists of 6 genomes for which we were not confident in assigning classification:

BrettBaker Meg 22 1618 bin 99

BrettBaker Meg19 1012 Bin 507

GCA 002763265 1 unclassified Archaea miscellaneous

BrettBaker Meg22 46 Bin 205

BrettBaker Meg22 1214 Bin 175

BrettBaker Meg22 1214 Bin 73

Mamarchaeota were defined in “Major New Microbial Groups Expand Diversity and Alter our Understanding of the Tree of Life”, Cell, 2018.

27. line 182: consider rephrasing title as the following paragraph is more about community function that roles of CPR and DPANN

We have changed the section title to “Potential roles of CPR and DPANN organisms in biogeochemical cycling by groundwater communities”.

28. 115: does the percentage of DPANN/CPR abundance take into account metagenomes from all filter size steps? Perhaps, this should be explained in more detail in the methods. How are these abundances comparable?

The reviewer raises a very valid question. All assessment of community-level abundances are based on bulk filtration (the entire community onto a 0.1 μm filter), samples that were explicitly collected for this purpose. This has been clarified further in the text. Comparison of bulk relative abundances versus relative abundances in each size fraction are now shown in SI Fig 8.

29. Figure 3B: it would be easier to see presence of genes if plotted onto a white background

We considered this suggestion and found that the detected genes stand out better with the current color scheme (warm colors on a black background).

30. line 215: which genes were considered to approximate carbon fixation capacity?

Please see Supplementary Tables 8-15 that we added to further clarify the details of metabolism analysis.

31. line 222: please check whether NosD necessarily is involved in nitrous oxide metabolism or represents an accessory protein that could be involved in another functional context

To our knowledge, NosD's only role is to insert copper into the NosZ protein (Holloway et al. 1996).

32. Figure 4A: change green color? (color blindness)

We thank the reviewer for pointing this out and have changed colors accordingly.

33. line 448: anoxic instead of anaerobic?

We have changed "more anaerobic" to "more anoxic."

34. line 451-453: this should be phrased a bit more careful

We agree, as noted in our response to point 20. We rephrased this sentence to emphasize that the proposed symbiotic relationships are only one plausible explanation for the observed abundance patterns.

35. is the ggKbase pipeline for taxonomic classification published/accessible to allow reproducibility?

The internal ggKbase database is comprised of publicly available genomes (NCBI), including many genomes for candidate phyla that originated in our lab. Given that these genomes are now publicly available, anyone wanting to do similar analyses can construct their own databases, and their databases would presumably give analogous results. At this time, our

database is several years old and needs to be updated, so it would not be a service to provide it.

36. how were insertions in ribosomal RNAs removed?

Insertions were identified and removed as described in Brown et al, "Unusual biology across a group comprising more than 15% of domain Bacteria," Nature, 2015. Briefly, all assembled 16S rRNA genes were searched against the manually curated structural alignment of the 16S rRNA provided with SSU-Align, and large gaps in the alignment between each sequence and the model revealed the boundaries of insertions. Scripts 16SfromHMM.py and strip_masked.py from ctbBio (<https://github.com/christophertbrown/bioscripts>) were used to identify and remove insertions >10 bp, respectively. This information has been added to the Methods.

37. ine 575: specify "SI Table xx"

NCBI accession info is now in SI Table 18.

38. Cryo-TEM: it is hard to see the pili connecting symbionts with hosts, i.e. pili that extent into the host cell. I am wondering whether this would be easier to see, without the white boxes surrounding it (getting too noisy). A simple arrow should besufficient.

We agree that removing the white boxes reduces the visual clutter, but the pili are difficult to see for a reader who is not accustomed to looking at whole cells in cryo-TEM images. For example, another reviewer commented that the pili within the white box are difficult to identify. To address this concern we have kept the white boxes in the main Fig 5, and have included zoomed in views of the pili, without the bounding white boxes to reduce visual clutter, in SI Figure 6.

Reviewer #3 (Remarks to the Author):

in their manuscript "Huge and variable diversity of episymbiotic CPR bacteria and DPANN archaea across groundwater ecosystems", He and colleagues describe the genome resolved metagenomic analysis of samples from a timeseries of groundwater samples from a site influenced by agricultural runoff, as well as single time points from 7 sites that are considered pristine. The analysis recovers 2007 genomes, of which 746 are affiliated with the CPR and DPANN clades. the manuscript focuses on the characteristics of the microorganisms represented by those 746 genomes.

1. *The authors frame their analysis in light of the possibility that CPR and DPANN organisms in groundwater could be a source for organisms of these clades detected in the human microbiome, but do not show any comparisons of the organisms detected in their groundwater samples to organisms detected in the human microbiome. Rather, the differences between the various sites suggest that environmental conditions play a major role in shaping the CPR/DPANN community. In light of this, it seems less likely that groundwater is a seedbank for the CPR/DPANN component of the human microbiome. The authors should either include a comparison of the CPR/DPANN community described here with those detected in humans, or alternatively ground this hypothesis by providing additional references on groundwater as a source for human microbiota in general.*

CPR bacteria are often detected in human microbiomes, but draft quality genomes of human-associated organisms are rare, making genomic comparisons difficult. Saccharibacteria is the one CPR phylum for which more than a few draft quality genomes from human microbiomes exist. A recent study (<https://doi.org/10.1016/j.celrep.2020.107939>) found that low genetic variation and high genomic synteny are maintained across human oral-associated and groundwater Saccharibacteria genomes, and propose different evolutionary models that explain this. We have added this reference to the text to give more context, but it is difficult for us to perform such a comparison ourselves as human microbiome datasets from humans drinking from these groundwater wells (as opposed to treated drinking water) have not been sequenced. This remains an objective for a future study. The current study lays the groundwork for such an investigation, underlining the importance of sampling the exact groundwater source that is relevant for each human studied. To address this concern, we have added these thoughts to the Discussion section.

2. *The current statement on data availability shows a link to ggKbase, and placeholders for bioproject ID and the supplemental table where the list will be available. Even after signing up for ggKbase, and using the link while logged in, I get an "You do not have permission to do that. Please email ggkbase-ticket@berkeley.edu for help." error. Please make sure the data is available before the manuscript is accepted/published.*

We apologize for this oversight. It is simple to open the data to the public, and we have done so. We are also in the process of submitting the data to NCBI, and have added Bioproject and Biosample numbers for all genomes (SI Table 18).

3. *The successful binning of the major representatives of the microbial communities in all but one site is quite impressive, as is the fact that the genomes represent a majority of reads on average. This said, can the authors discuss/speculate (here and in the manuscript) why the binning of abundant lineages in pr2 is so much less effective than it is for the other sites?*

We thank the reviewer for asking about this. The explanation is that a larger than normal proportion of the *Pr2* sequencing dataset comprised sequences from phage (and the sample contains some eukaryote sequences). These phage scaffolds (identified as so by lack of a bacterial or archaeal winner taxonomy at the kingdom level, and presence of phage-associated genes) comprise much of the longer scaffolds in the unbinned *Pr2* data.

4. *The overlap between sites pr1 & pr7 is interesting, especially since it comprises the majority of genomes (44 of 63) recovered from pr7. The authors provide reasons why the aquifers are unlikely to be connected, and why index hopping is unlikely, but provide no explanation why the community overlap would exist. Since the geochemistry doesn't appear to be strongly similar and the sites are located further apart than some of the other sites are from pr1, it is a mystery why the similarity would be so high. Is there any chance of sample switching?*

We note that a significant fraction of identified organisms (19 out of 63 in *Pr7*, and 204 out of 248 in *Pr1*) are distinct at the species level. We consider sample switching to be unlikely given that *Pr1* and *Pr7* were sampled at different times than any other site, and therefore underwent separate DNA extraction, library prep, and were sequenced separately from other filters/sites.

5. *For figure 3, while I like that general aspects of community metabolic potential are included, figure 3B shows that the contribution of the CPR/DPANN genomes to these overviews is minimal. Thus, the connection to the central point of the manuscript seems limited.*

We consider this analysis to be critical to the argument for symbiotic lifestyles. As has come to light recently, TM7/Saccharibacteria organisms are generally minimal in terms of capacities, but gene content can vary significantly with environment (e.g., human-associated versus groundwater versus soil).

6. *For the metabolic genes present in the CPR, did the authors do any phylogenetic verification of the annotation? NirK and nosZ can be very sensitive to misannotation.*

We built a *nirK* tree and found that *nirK* genes from this study's CPR and DPANN genomes form their own clade, with the closest clade including known nitrogen cyclers such as Candidatus

Brocadia. We did not find *nosZ* genes in our CPR and DPANN genomes but did find *nosD* and *nirD* genes.

7. *If I understand figure 4B correctly, the lower section only shows centroid genomes of the same analysis represented in the top section. If that interpretation is correct, how is it possible that the centroid for some of the 0.1 micron fraction is outside of the range of the data shown in the top section*

We have corrected our terminology and thank the reviewer for pointing this out. Each point in the bottom panel is calculated based on the distances between the genomes between the sites. NMDS arranges the points on the plot so that the distances among each pair of points correlates as best as possible to the dissimilarity between those two samples. The values on the two axes tell you nothing about the variables for a given sample – the plot is just a two dimensional space to arrange the points.

8. *The cryoSEM shown in figure 5 is not very clear. While the pili-like structures are clear, it is very difficult to identify the pili entering the host cell in the white boxes (panel H, line 350). It is also hard to see the higher density layer at point of host cell attachment in 5d and 5e and the distinction single membrane versus double membrane. 5i should have an orange arrow according to line 295.*

We have improved these images to the extent possible using state of the art image enhancement methods, but unfortunately these images are difficult to decipher, especially to people who are not used to looking at whole cells in cryo-TEM images. To try to address this, we have included zoomed in views of the pili, without the bounding white boxes to reduce visual clutter, in SI Figure 6.

9. *line 329: the authors should be careful in interpreting phylogeny from the cryo-EM data. There is an assumption that the small cells observed here are from the CPR lineage, but no direct evidence of this. I agree it is a likely interpretation, but the language describing the interaction between ultra-small putative CPR bacteria and host cells should reflect this is an assumption.*

In response to this comment and one from Reviewer 2 we acknowledge that it isn't possible to be certain (either way, CPR or DPANN, the results demonstrate episybiosis).

10. *comparisons of different size fractions in the different panels (>2.5 vs 0.2-0.1 in A, 2.5-0.65 vs 0.2-0.1 in B, and >2.5 vs 0.65-0.2 in SI fig 7) make it seem cherry picked. While I think showing the strongest effects in the main figure is fine, it would be good to show all comparisons in the supplement.*

The results of all size fraction comparisons performed are now available in SI Table 17.

11. *The colors/symbols on SI fig 2 do not fit with the legend.*

We have fixed the colors/symbols in SI Fig 2.

Brown, Christopher T., Matthew R. Olm, Brian C. Thomas, and Jillian F. Banfield. 2016.

“Measurement of Bacterial Replication Rates in Microbial Communities.” *Nature Biotechnology* 34 (12): 1256–63.

Holloway, P., W. McCormick, R. J. Watson, and Y. K. Chan. 1996. “Identification and Analysis of the Dissimilatory Nitrous Oxide Reduction Genes, nosRZDFY, of *Rhizobium Meliloti*.” *Journal of Bacteriology* 178 (6): 1505–14.

Olm, Matthew R., Alexander Crits-Christoph, Spencer Diamond, Adi Lavy, Paula B. Matheus Carnevali, and Jillian F. Banfield. 2020. “Consistent Metagenome-Derived Metrics Verify and Delineate Bacterial Species Boundaries.” *mSystems* 5 (1).
<https://doi.org/10.1128/mSystems.00731-19>.

Decision Letter, first revision:

Dear Professor Banfield,

Thank you for your patience while your manuscript "Huge and variable diversity of episymbiotic CPR bacteria and DPANN archaea across groundwater ecosystems" was under peer review at Nature Microbiology. It has now been seen by our referees, and in the light of their advice I am delighted to say that we can in principle offer to publish it. First, however, we would like you to revise your paper to address the points made by the reviewers, and to ensure that it is in Nature Microbiology format.

The referees' remaining comments are clear, and should not be difficult to implement. Editorially, we will need you to make some changes so that the paper complies with our Guide to Authors at <http://www.nature.com/nmicrobiol/info/gta>.

Specific points:

In particular, while checking through the manuscript and associated files, we noticed the following specific points which we will need you to address:

1. Main text is 5056 and must be reduced to 4000 words, see my suggestions. See the attached and lightly edited version of your manuscript and please use this as the basis for the final revision. Please see my suggested title and some alterations to the abstract, introduction and discussion. I've shortened
2. Please convert 9 Supplementary figures to 9 extended data figures, and revise all the callouts in the manuscript. Each piece of Supplementary information or extended data must be called out at least once in the manuscript.
3. Please collate your Supplementary tables into one combined Supplementary information pdf. You can also upload each Supplementary table separately in the original Excel format.
4. Please upload each figure separately, in an editable format. The main and extended data figures need to be uploaded individually and without captions; they should be provided in eps, tiff or jpeg format. The main text file needs to be submitted in Word or LaTeX format, and all (main and extended) figure legends need to be included at the end of this file. Do NOT include any figures in the main text file. Choosing the right electronic format for your figures at this stage will speed up the processing of your paper. We would like the figures to be supplied as vector files - EPS, PDF, AI or postscript (PS) file formats (not raster or bitmap files), preferably generated with vector-graphics software (Adobe Illustrator for example). Please try to ensure that all figures are non-flattened and fully editable. All images should be at least 300 dpi resolution (when figures are scaled to approximately the size that they are to be printed at) and in RGB colour format. Please do not submit Jpeg or flattened TIFF files. Please

see also 'Guidelines for Electronic Submission of Figures' at the end of this letter for further detail. Please view http://www.nature.com/authors/editorial_policies/image.html for more detailed guidelines.

We will edit your figures/tables electronically so they conform to Nature Microbiology style. If necessary, we will re-size figures to fit single or double column width. If your figures contain several parts, the parts should be labelled lower case a, b, and so on, and form a neat rectangle when assembled.

5. We do not use Supplementary Methods. Please move as much of the methodology as you can into the ONLINE METHODS section and put any other needed text into a Supplementary Note. Let me know if this will be more than 10,000 words.

6. Please ensure you address all the points raised by our partner SNTPS in the attached files. Please also use those comments to finalize the reporting summary for resubmission.

7. Please add a data availability statement to the manuscript and include deposition of the short reads, long reads and the MAGs as needed, as all of this must be available to readers.

8. Your article will be published Open access, per the inclusion of new phyla representatives in your MAGs.

9. Figure legends need to be less than 375 words long. Figure legend titles should be on one line where possible so please shorten to fit.

10. A competing interests statement needs to be included in the manuscript text (before or after the Acknowledgements).

11. A data availability statement must be included before the methods. This must contain accession codes for the reads and the assemblies. This section should inform readers about the availability of the data used to support the conclusions of your study. This information includes accession codes to public repositories (data banks for protein, DNA or RNA sequences, microarray, proteomics data etc...), references to source data published alongside the paper, unique identifiers such as URLs to data repository entries, or data set DOIs, and any other statement about data availability. If DOIs are provided, we also strongly encourage including these in the Reference list (authors, title, publisher (repository name), identifier, year). For more guidance on how to write this section please see: <http://www.nature.com/authors/policies/data/data-availability-statements-data-citations.pdf>

12. Please use the checks made by our partner SNTPS to improve reporting throughout and to amend and revise the reporting summary. A finalized reporting summary must be uploaded with the final submission.

13. Please check the PDF of the whole paper and figures (on our manuscript tracking system) VERY CAREFULLY when you submit the revised manuscript. This will be used as the 'reference copy' to make sure no details (such as Greek letters or symbols) have gone missing during file-transfer/conversion and re-drawing.

14. All Supplementary Information must be submitted in accordance with the instructions in the attached Inventory of Supporting Information, and should fit into one of three categories: PLEASE NOTE you should convert all of your Supp info figures into Extended data figures.

1. EXTENDED DATA: Extended Data are an integral part of the paper and only data that directly contribute to the main message should be presented. These figures will be integrated into the full-text

HTML version of your paper and will be appended to the online PDF. There is a limit of 10 Extended Data figures, and each must be referred to in the main text. Each Extended Data figure should be of the same quality as the main figures, and should be supplied at a size that will allow both the figure and legend to be presented on a single legal-sized page. Each figure should be submitted as an individual .jpg, .tif or .eps file with a maximum size of 10 MB each. All Extended Data figure legends must be provided in the attached Inventory of Accessory Information, not in the figure files themselves.

2. SUPPLEMENTARY INFORMATION: Supplementary Information is material that is essential background to the study but which is not practical to include in the printed version of the paper (for example, video files, large data sets and calculations). Each item must be referred to in the main manuscript and detailed in the attached Inventory of Accessory Information. Tables containing large data sets should be in Excel format, with the table number and title included within the body of the table. All textual information and any additional Supplementary Figures (which should be presented with the legends directly below each figure) should be provided as a single, combined PDF. Please note that we cannot accept resupplies of Supplementary Information after the paper has been formally accepted unless there has been a critical scientific error.

All Extended Data must be called out in your manuscript and cited as Extended Data 1, Extended Data 2, etc. Additional Supplementary Figures (if permitted) and other items are not required to be called out in your manuscript text, but should be numerically numbered, starting at one, as Supplementary Figure 1, not SI1, etc.

3. SOURCE DATA: We encourage you to provide source data for your figures whenever possible. Full-length, unprocessed gels and blots must be provided as source data for any relevant figures, and should be provided as individual PDF files for each figure containing all supporting blots and/or gels with the linked figure noted directly in the file. Statistics source data should be provided in Excel format, one file for each relevant figure, with the linked figure noted directly in the file. For imaging source data, we encourage deposition to a relevant repository, such as figshare (<https://figshare.com/>) or the Image Data Resource (<https://idr.openmicroscopy.org>).

15. Nature Research journals [encourage authors to share their step-by-step experimental protocols](https://www.nature.com/nature-research/editorial-policies/reporting-standards#protocols) on a protocol sharing platform of their choice. Nature Research's Protocol Exchange is a free-to-use and open resource for protocols; protocols deposited in Protocol Exchange are citable and can be linked from the published article. More details can found at www.nature.com/protocolexchange/about.

Please note that after the paper has been formally accepted you can only provide amended Supplementary Information files for critical changes to the scientific content, not for style. You should clearly explain what changes have been made if you do resupply any such files.

Figure legends must provide a brief description of the figure and the symbols used, within 350 words. This must include definitions of any error bars employed in the figures.

It is a condition of publication that you include a statement before the acknowledgements naming the author to whom correspondence and requests for materials should be addressed.

Finally, we require authors to include a statement of their individual contributions to the paper -- such as experimental work, project planning, data analysis, etc. -- immediately after the acknowledgements. The statement should be short, and refer to authors by their initials. For details please see the Authorship section of our joint Editorial policies at http://www.nature.com/authors/editorial_policies/authorship.html

We will not send your revised paper for further review if, in the editors' judgement, the referees'

comments on the present version have been addressed. If the revised paper is in Nature Microbiology format, in accessible style and of appropriate length, we shall accept it for publication immediately.

Please resubmit electronically

- * the final version of the text (not including the figures) in either Word or Latex.
- * publication-quality figures. For more details, please refer to our Figure Guidelines, which is available here: https://mts-nmicrobiol.nature.com/letters/Figure_guidelines.pdf
- * Extended Data & Supplementary Information, as instructed
- * a point-by-point response to any issues raised by our referees and to any editorial suggestions.
- * any suggestions for cover illustrations, which should be provided at high resolution as electronic files. Please note that such pictures should be selected more for their aesthetic appeal than for their scientific content. I am sure you will understand that we cannot make any promise as to whether any of your suggestions might be selected for the cover of Nature Microbiology.

Please use the following link to access your home page:

{REDACTED}

- * This url links to your confidential homepage and associated information about manuscripts you may have submitted or be reviewing for us. If you wish to forward this e-mail to co-authors, please delete this link to your homepage first.

Please also send the following forms as a PDF by email to microbiology@nature.com.

- * Please sign and return the <http://www.nature.com/documents/snl-ltp.docx> target="_blank">Licence to Publish form .

- * Or, if the corresponding author is either a Crown government employee (including Great Britain and Northern Ireland, Canada and Australia), or a US Government employee, please sign and return the <http://www.nature.com/documents/snl-ltp-crown.docx> target="_blank"> Licence to Publish form for Crown government employees, or a <http://www.nature.com/documents/snl-ltp-govus.docx> target="_blank"> Licence to Publish form for US government employees.

- * Should your Article contain any items (figures, tables, images, videos or text boxes) that are the same as (or are adaptations of) items that have previously been published elsewhere and/or are owned by a third party, please note that it is your responsibility to obtain the right to use such items and to give proper attribution to the copyright holder. This includes pictures taken by professional photographers and images downloaded from the internet. If you do not hold the copyright for any such item (in whole or part) that is included in your paper, please complete and return this <http://www.nature.com/documents/thirdpartyrights-origres.doc> target="_blank">Third Party Rights Table, and attach any grant of rights that you have collected.

For more information on our licence policy, please consult <http://npg.nature.com/authors>.

ORCID

Nature Microbiology is committed to improving transparency in authorship. As part of our efforts in this₂

direction, we are now requesting that all authors identified as 'corresponding author' create and link their Open Researcher and Contributor Identifier (ORCID) with their account on the Manuscript Tracking System (MTS) prior to acceptance. ORCID helps the scientific community achieve unambiguous attribution of all scholarly contributions. For more information please visit <http://www.springernature.com/orcid>

For all corresponding authors listed on the manuscript, please follow the instructions in the link below to link your ORCID to your account on our MTS before submitting the final version of the manuscript. If you do not yet have an ORCID you will be able to create one in minutes. <https://www.springernature.com/gp/researchers/orcid/orcid-for-nature-research>

IMPORTANT: All authors identified as 'corresponding author' on the manuscript must follow these instructions. Non-corresponding authors do not have to link their ORCIDs but are encouraged to do so. Please note that it will not be possible to add/modify ORCIDs at proof. Thus, if they wish to have their ORCID added to the paper they must also follow the above procedure prior to acceptance.

To support ORCID's aims, we only allow a single ORCID identifier to be attached to one account. If you have any issues attaching an ORCID identifier to your MTS account, please contact the [Platform Support Helpdesk](http://platformsupport.nature.com/).

Nature Research journals [encourage authors to share their step-by-step experimental protocols](https://www.nature.com/nature-research/editorial-policies/reporting-standards#protocols) on a protocol sharing platform of their choice. Nature Research's Protocol Exchange is a free-to-use and open resource for protocols; protocols deposited in Protocol Exchange are citable and can be linked from the published article. More details can found at www.nature.com/protocolexchange/about.

We hope that you will support this initiative and supply the required information. Should you have any query or comments, please do not hesitate to contact me.

We hope to hear from you within two weeks; please let us know if the revision process is likely to take longer.

Reviewer Comments:

Reviewer #1 (Remarks to the Author):

The authors addressed all my concerns of the first review. Thanks.

Reviewer #2 (Remarks to the Author):

Review He et al., 2020

I have enjoyed reading the revised version of the article entitled "Huge and variable diversity of episymbiotic CPR bacteria and DPANN archaea across groundwater ecosystems". The authors have addressed my major concerns and I think that the addition of introductory sentences to each paragraph help to better build up the story. Also, the more detailed method section is very helpful. I have just a few remaining comments, see below.

General comment:

the diversity of DPANN (especially in Ag samples) is very impressive considering that there are very few other archaeal bins. How do you explain this? Do you suspect these DPANN to interact with Bacteria? Maybe worth to mention in the discussion/results.

Minor comments:

line 23: maybe remove "However" (not connected to first sentence)

line 94: what do you mean by "All sites were sampled at <100 m". Perhaps delete here and move to methods section?

line 126: regarding the definition of phylum-level lineages. I apologize for asking for ANI in my previous review. That was a typo and should have been AAI (average amino acid identity). This latter measure could be used as an additional mean to validate the phylum-level distinction.

line 116 and Fig 1C: I noticed that, while in most sites, bins have been obtained for the most abundant representatives, in one site (Pr2), the abundant organisms are not binned and very few reads mapped to bins. Just out of curiosity: do you have any thoughts what may explain this? Maybe worth mentioning.

line 147: sentence starting "Principal component analysis...", maybe add an explanation to justify why this has been done: "in order to..."

line 148: should this be "separate" instead of "separately"?

line 232: are you sure, these are bona fide NirD and not just small iron-sulfur cluster proteins? Please check carefully.

line 235: as far as I know, the enzymes encoded by these genes (cysC, cysN) also function in sulfur assimilation, please check and eventually tone down the conclusion drawn from the presence of these genes.

line 297 and 392: are you sure, these are CPR? or could they be DPANN? (see also general comment)

line 355: remove "that"?

line 364: presence of DPANN

line 400: this sentence seems to contradict the paragraph above, especially because it starts with "as expected". Based on the paragraph above, this is rather "unexpected"?

line 482: absence does not need to be in italic? Argument is sufficiently convincing.

line 590: word missing in front of "Useach"?

line 591: replace "concatenated with" by "added to" (to avoid confusion with the phylogenetic meaning of concatenation: i.e. combining different aligned proteins).

line 593: replace "all ribosomal proteins..." by "all individual ribosomal protein alignments..."

Tables: its great that the authors now provide the tables with the information of absence/presence of pathways and enzymes. Unfortunately, the tables/data files were not numbered, such that it was a bit difficult to link them to the legend. it would be helpful, if the first sheet per excel data file, could provide the header and legend for the file.

References:

line 51: perhaps cite: Brandt and Albertsen, 2018. Investigation of Detection Limits and the Influence of DNA Extraction and Primer Choice on the Observed Microbial Communities in Drinking Water Samples Using 16S rRNA Gene Amplicon Sequencing

line 460 and following part in Discussion: maybe worth integrating following manuscript: McLean et al., 2020. Acquisition and Adaptation of Ultra-small Parasitic Reduced Genome Bacteria to Mammalian Hosts.

You mention in your response to reviewer 3 that this reference was added but I could not see it... Not sure, whether I missed this or whether it has not been added. Can you check this?

Author Rebuttal, first revision:

Reviewer #1 (Remarks to the Author):

The authors addressed all my concerns of the first review.

Thanks. Reviewer #2 (Remarks to the Author):

I have enjoyed reading the revised version of the article entitled “Huge and variable diversity of episyntrophic CPR bacteria and DPANN archaea across groundwater ecosystems”. The authors have addressed my major concerns and I think that the addition of introductory sentences to each paragraph help to better build up the story. Also, the more detailed method section is very helpful. I have just a few remaining comments, see below.

We thank the reviewer for their careful reading of our manuscript and their positive reception to our changes.

General comment:

the diversity of DPANN (especially in Ag samples) is very impressive considering that there are very few other archaeal bins. How do you explain this? Do you suspect these DPANN to interact with Bacteria? Maybe worth to mention in the discussion/results.

This is certainly an intriguing question that we’ve asked ourselves. We have added a sentence posing this question in the discussion.

Minor comments:

line 23: maybe remove “However” (not connected to first sentence)

We have made this change.

line 94: what do you mean by “All sites were sampled at <100 m”. Perhaps delete here and move to methods section?

We have removed this sentence and added sampling depth information to the methods.

line 126: regarding the definition of phylum-level lineages. I apologize for asking for ANI in my previous review. That was a typo and should have been AAI (average amino acid identity). This latter measure could be used as an additional mean to validate the phylum-level distinction.

We thank the reviewer for their clarification.

line 116 and Fig 1C: I noticed that, while in most sites, bins have been obtained for the most abundant representatives, in one site (Pr2), the abundant organisms are not binned and very few reads mapped to bins. Just out of curiosity: do you have any thoughts what may explain this? Maybe worth mentioning.

We can't say for sure why Pr2 stands out in this respect, but it's likely due in part to a combination of community complexity and sequencing depth. We have decided not to add to the text as this is speculative and does not alter the results or conclusions.

line 147: sentence starting "Principal component analysis...", maybe add an explanation to justify why this has been done: "in order to..."

We have heavily revised this entire section of text, making it more clear that the PCA is done to evaluate differences between communities, and moving the principal component analysis to the SI.

line 148: should this be "separate" instead of "separately"?

We have made this change.

line 232: are you sure, these are bona fide NirD and not just small iron-sulfur cluster proteins? Please check carefully.

We are reasonably confident that these are NirD genes.

line 235: as far as I know, the enzymes encoded by these genes (cysC, cysN) also function in sulfur assimilation, please check and eventually tone down the conclusion drawn from the presence of these genes.

We have toned down the sulfur cycling conclusion and amended the relevant statements to be more general.

line 297 and 392: are you sure, these are CPR? or could they be DPANN? (see also general comment)

We have amended the text to express this ambiguity.

line 355: remove "that"?

We have done this.

line 364: presence of

DPANN

We have amended the text to express this ambiguity.

line 400: this sentence seems to contradict the paragraph above, especially because it starts with “as expected”. Based on the paragraph above, this is rather “unexpected”?

We thank the reviewer for pointing this out, and have removed “as expected.”

line 482: absence does not need to be in italic? Argument is sufficiently convincing.

We have removed the italics.

line 590: word missing in front of “Usearch”?

We have amended the sentence.

line 591: replace “concatenated with” by “added to” (to avoid confusion with the phylogenetic meaning of concatenation: i.e. combining different aligned proteins).

We have made the suggested change.

line 593: replace “all ribosomal proteins...” by “all individual ribosomal protein alignments...”

We have made the suggested change.

Tables: its great that the authors now provide the tables with the information of absence/presence of pathways and enzymes. Unfortunately, the tables/data files were not numbered, such that it was a bit difficult to link them to the legend. it would be helpful, if the first sheet per excel data file, could provide the header and legend for the file.

I’m not sure if the table/data file numbering was lost in the previous submission, but all of our supplementary data and files are numbered.

References:

line 51: perhaps cite: Brandt and Albertsen, 2018. Investigation of Detection Limits and the Influence of DNA Extraction and Primer Choice on the Observed Microbial Communities in Drinking Water Samples Using 16S rRNA Gene Amplicon Sequencing

We thank the reviewer for the suggestion. However, we already have 3 references to support the point and 90 total references, so have decided not to add a 4th reference.

line 460 and following part in Discussion: maybe worth integrating following manuscript: McLean et al., 2020. Acquisition and Adaptation of Ultra-small Parasitic Reduced Genome Bacteria to Mammalian Hosts. You mention in your response to reviewer 3 that this reference was added but I could not see it... Not sure, whether I missed this or whether it has not been added. Can you check this?

We thank the reviewer for catching this missing reference. We have added to this reference to the text.

Final Decision Letter:

Dear Professor Banfield,

I am pleased to accept your Article "Genome-resolved metagenomics reveals site-specific diversity of episymbiotic CPR bacteria and DPANN archaea in groundwater ecosystems" for publication in Nature Microbiology. Thank you for having chosen to submit your work to us.

Before your manuscript is typeset, we will edit the text and look particularly carefully at the titles of all papers to ensure that they are relatively brief and understandable.

The subeditor may send you the edited text for your approval. Once your manuscript is typeset you will receive a link to your electronic proof via email within 20 working days, with a request to make any corrections within 48 hours. If you have queries at any point during the production process then please contact the production team at rjsproduction@springernature.com. Once your paper has been scheduled for online publication, the Nature press office will be in touch to confirm the details.

Acceptance of your manuscript is conditional on all authors' agreement with our publication policies (see www.nature.com/nmicrobiolate/authors/gta/content-type/index.html). In particular your manuscript must not be published elsewhere and there must be no announcement of the work to any media outlet until the publication date (the day on which it is uploaded onto our website).

The Author's Accepted Manuscript (the accepted version of the manuscript as submitted by the author) may only be posted 6 months after the paper is published, consistent with our [self-archiving embargo](http://www.nature.com/authors/policies/license.html). Please note that the Author's Accepted Manuscript may not be released under a Creative Commons license. For Nature Research Terms of Reuse of archived manuscripts please see: <http://www.nature.com/authors/policies/license.html#terms>

P.S. Click on the following link if you would like to recommend Nature Microbiology to your librarian <http://www.nature.com/subscriptions/recommend.html#forms>

** Visit the Springer Nature Editorial and Publishing website at http://editorial-jobs.springernature.com?utm_source=ejP_NMicro_email&utm_medium=ejP_NMicro_email&utm_campaign=ejP_NMicro for more information about our career opportunities. If you have any questions please click [here](mailto:editorial.publishing.jobs@springernature.com). **